# Semi-automated assembly of high-quality diploid human reference genomes

The current human reference genome, GRCh38, represents over 20 years of effort to generate a high-quality assembly, which has benefitted society[1,2]. However, it still has many gaps and errors, and does not represent a biological genome as it is a blend of multiple individuals[3,4]. Recently, a high-quality telomere-to-telomere reference, CHM13, was generated with the latest long-read technologies, but it was derived from a hydatidiform mole cell line with a nearly homozygous genome[5]. To address these limitations, the Human Pangenome Reference Consortium formed with the goal of creating high-quality, cost-effective, diploid genome assemblies for a pangenome reference that represents human genetic diversity[6]. Here, in our first scientific report, we determined which combination of current genome sequencing and assembly approaches yield the most complete and accurate diploid genome assembly with minimal manual curation. Approaches that used highly accurate long reads and parent–child data with graph-based haplotype phasing during assembly outperformed those that did not. Developing a combination of the top-performing methods, we generated our first high-quality diploid reference assembly, containing only approximately four gaps per chromosome on average, with most chromosomes within ±1% of the length of CHM13. Nearly 48% of protein-coding genes have non-synonymous amino acid changes between haplotypes, and centromeric regions showed the highest diversity. Our findings serve as a foundation for assembling near-complete diploid human genomes at scale for a pangenome reference to capture global genetic variation from single nucleotides to structural rearrangements.

The initial draft of the human reference genome was the outcome of over a decade of effort by the Human Genome Project (HGP), with cost exceeding US$2.7 billion (over US$5 billion at today's value)[1,2,7]. Its current build, GRCh38, reflects another decade of additional effort by the Genome Reference Consortium and others to correct the primary assembly. It was created from physical maps of thousands of individually sequenced 40–2,000-kb bacterial artificial chromosomes (BACs), yeast artificial chromosomes (YACs) and fosmid clones, supplemented with whole-genome sequence data[1,2]. It is a combination of DNA sequences from 20 anonymous volunteers, with one individual representing approximately 70% of the sequence[2]. Over the years, the primary assembly was improved from having over 150,000 gaps to just 995 gaps in the current GRCh38 assembly[2,3]. Therefore, despite being one of the most complete human reference genomes available, GRCh38 represents an incomplete composite and does not adequately capture the spectrum of human global genomic variation[8].

In the years following the HGP, several technological limitations prevented the generation of new human reference genomes of similar or higher quality at scale. Sequence duplications much larger than the sequence read lengths are particularly challenging to assemble. Although resequencing efforts using less expensive short reads contributed to revealing more single-nucleotide variation (SNV), these SNVs, and more so structural variations (SVs), are not fully captured[9,10]. The sequencing enzymes used often have difficulty reading through regions with complex structures, such as GC-rich regions found in promoters that regulate gene expression[11,12]. It is also now clear that merging diverse haplotypes into a single haploid assembly, even from the same individual, introduces multiple types of errors[9,11], including: switch errors in which variants from each haplotype are assembled into the same pseudo-haplotype; false duplications and associated gaps in which more divergent haplotype homologues are assembled as separate false paralogues; and nucleotide consensus errors due to collapses between haplotypes. One also needs diploid assemblies to: separately assemble the X and Y sex chromosomes; determine maternal and paternal gene expression imprinting, which can lead to haplotype-specific diseases[13]; and determine functional consequences of allele combinations that co-segregate on the same haplotype[14,15].

Major improvements have since been made in sequence read lengths[4,16], long read nucleotide accuracy[17], contig algorithms, scaffolding contigs into chromosomes[11,18–20], haplotype phasing[21–23] and technologies with reduced sequencing cost. These advances include those made by the Vertebrate Genomes Project (VGP)[11], the Human Genome Structural Variation Consortium (HGSVC)[10] and the Telomere-to-Telomere (T2T) consortium, which produced the first complete human reference genome, of the CHM13 cell line[5]. CHM13 originated from a hydatidiform mole, in which an ovum without maternal chromosomes was fertilized by one sperm, which then duplicated its DNA, leading to two nearly identical paternal haploid complements with an X chromosome (46,XX), eliminating the need to separate haplotypes and purge associated diploid assembly errors. Completing the

T2T-CHM13 assembly also required a substantial amount of manual curation by dozens of people over many months, with different groups focused on each chromosome. Thus, despite improvements, additional developments are needed to assemble diploid genomes at high quality and at scale, which we believe to be critical for clinically relevant samples and understanding human genetic variation.

To help overcome these limitations, in 2019 the National Human Genome Research Institute (NHGRI) invested in an international Human Pangenome Reference Consortium (HPRC), with an aspired goal of producing a high-quality pangenome reference representing over 99% of human genetic diversity for minor alleles of at least 1% or higher frequency in the human population[6]. We estimate that one could start to approach this goal with complete de novo assemblies of approximately 450 individuals (for example, 900 haplotypes) from the world population (Supplementary Note 1). That is, a primary goal of the HPRC is to build high-quality diploid assemblies from multiple individuals and then merge them to build a pangenome graph[6]. Starting in 2020, we tested the current best practices in sequencing technologies and automated assembly algorithms on one human sample, HG002, an openly consented Ashkenazi individual from the Personal Genome Project[24]. We included parental samples (HG003-father and HG004-mother) for trio-based assemblies, in which parental sequence data were used to sort haplotypes in the offspring sequence data[11,22]. Extensive evaluation of the resulting assemblies alongside GRCh38 and CMH13 led to new approaches that yielded the best values in over 60 metrics and new biological discoveries, including uncovering more genetic variation between haplotypes. We also identified areas of needed improvement to achieve automated complete and error-free diploid genome assemblies.

## Data types and algorithms

We chose HG002 because of available previous extensive public data[25] and variant benchmarks[26] generated by the Genome in a Bottle (GIAB) consortium. As a male sample, it enables the assembly and evaluation of both X and Y chromosomes. We obtained or generated additional state-of-the-art sequence data types, including PacBio HiFi long reads and Oxford Nanopore (ONT) long reads (more than 10 kb) for generating contigs, and long-range link information (for example, 10X linked reads, Hi-C linked reads, optical maps and Strand-seq) for scaffolding the contigs (Supplementary Table 1). These choices were made on the basis of lessons learned for producing high-quality assemblies from other consortia (for example, VGP[11], T2T[5] and HGSVC[10]) or individual laboratories[27–29]. In particular, long-read-based assemblies are more contiguous and more structurally accurate than short-read-based assemblies, long-range linking information can place contigs into chromosome-level scaffolds, and haplotype phasing and high base accuracy help to prevent false duplications and other common assembly errors.

We generated the high-molecular-weight DNA from an early passage (#4-10) HG002 immortalized lymphoblastoid cell line (LCL) derived from B lymphocytes, because cell lines are easier to isolate high-quality DNA, can be returned to without new blood collections and are useful for future functional gene experiments in a given genetic background. We analysed chromosome status in mitotic chromosome spreads of the LCL and found most spreads maintained a diploid 46,XY karyotype, with a small proportion being tetraploid (Supplementary Fig. 1a,b). We also did not observe large-scale within and between structural chromosomal abnormalities. A minor frequency of tetraploid karyotypes should not present a major concern for assembly as it is an exact genome doubling event.

We made an open call to the international genome community for an assembly bakeoff (that is, assemblathon) to produce the most complete and highest-quality, automated genome assembly possible of HG002 with the data provided (https://humanpangenome.org/hg002/).

We generated high sequence coverage for all technologies, so that different coverage levels could be tested, but asked that all assemblers test at least the same downsampled manufacturer recommended levels to prevent coverage as a variable when comparing different assembly algorithms. We received 23 assembly combinations, from 14 groups, including HPRC members, that used different data types and algorithms for contiging, scaffolding and/or haplotype phasing when attempted (Table 1); we named them asm1 to asm23, with suffixes a/b for haplotypes. Among these 23, 12 assembly algorithms were used: Canu and HiCanu[17], CrossStitch, DipAsm[29], FALCON Unzip[21], Flye[30], hifiasm[31], Maryland Super-Read Celera Assembler (MaSuRCA)[32], NECAT[33], Peregrine[34], Shasta[28] and wtdbg[35] (Table 1). We classified the assemblies into four categories: (1) diploid scaffolded assemblies, which attempted to assemble comparable contigs and scaffolds of both haplotypes or two pseudohaplotypes (mixed paternal and maternal-derived sequences); (2) diploid contig-only assemblies, which attempted to assemble only contigs of both haplotypes and/or pseudohaplotypes or a more complete assembly representing one pseudohaplotype; (3) haploid scaffolded assemblies, in which contigs and scaffolds were merged into one pseudohaplotype; and (4) haploid contig-only assemblies, in which only contigs were generated and merged into one pseudohaplotype (Table 1 and Supplementary Table 2a,b). Cross-Stitch and MaSuRCA are reference-based (to GRCh38 in this study), in which MaSuRCA used GRCh38 to order and orient assembled HG002 contigs into chromosome-level scaffolds, followed by gap filling with the GRCh38 sequence. Although these assemblies (asm1, asm15 and asm17) are not 'pure' de novo, they are included to establish a baseline for capturing variation guided by a reference assembly. Following the VGP model[11], we assessed over 60 metrics under 14 categories (Supplementary Table 2). About one-third of these metrics were calculated with the Merqury k-mer analysis tool[36], which we automated. Rather than having a ground-truth, most of these metrics measured the level of consistency of data types relative to the assemblies.

## Contamination and organelle genomes

We screened for non-human DNA and found that all de novo assemblies had between 1 and 25 contigs or scaffolds with library adaptor sequence contamination, which were not successfully removed during read preprocessing (Extended Data Fig. 1a and Supplementary Table 2c). The presence of adaptor sequences on reads with human sequences introduced gaps between the human-based contigs; reads with adaptor alone were concatenated to make adaptor-only contigs (Supplementary Note 2). We also found instances of assembled bacterial (*Escherichia coli*) and yeast (*Saccharomyces cerevisiae*) genomes, either as standalone contigs or scaffolds (three assemblies), chimeric with human genomic DNA (four assemblies), or both (four assemblies; Extended Data Fig. 1b and Supplementary Table 2c). There were typically 0–6 copies of these microbial genomes per assembly, except in the wtdgb2 assembly with 35 *E. coli* and 46 *S. cerevisiae* contigs. For the other assemblies, microbial contamination was inadvertently removed before submission due to: (1) not matching the GRCh38 reference for the reference-based assemblies; (2) filtering out scaffolds below a specific size; or (3) moving from the primary to the alternate assembly.

There were also from 1 to approximately 40 assembled human mitochondrial (MT) contigs in approximately 74% (17 out of 23) of the assemblies (Extended Data Fig. 1c and Supplementary Table 2c). In the trio-based assemblies, the MT genomes were all associated with the maternal haplotype, indicating that the MT reads were correctly sorted during haplotype phasing before generating contigs (in the VGP Trio assembly, the MT genome was purposely included in both haplotypes to avoid NUMT overpolishing[11]). Most MT contigs were full-length genomes, further demonstrating[37] that with long reads most new assembly algorithms can assemble a MT genome in one contig. Part of the reason for the differential presence of MT genomes in the

**Table 1 | Summary of sequencing and assembly approaches tested**

| ID | Pipeline | Technologies | Contigs | Scaffolders | Team |
|---|---|---|---|---|---|
| *Diploid contig and scaffold assemblies* | | | | | |
| asm23a,b | Trio VGP | CLR, 10X, BN and Hi-C | Trio Canu | Trio based: Scaff10x, Bionano solve and Salsa | Rockefeller |
| asm10a,b | DipAsm | HiFi and HiC | Peregrine | DipAsm, 3D-DNA, HapCUT2 and Whatshap | UCPH |
| asm2a,b | DipAsm HiRise | HiFi and HiC | Peregrine | HiRise and HapCUT2 | Dovetail |
| asm22a,b | DipAsm Salsa | HiFi and HiC | Peregrine | Salsa and HapCUT2 | Dovetail |
| asm14a,b | PGAS | HiFi and Strand-seq | Peregrine | SaaRclust | HHU + UW |
| asm17a,b | CrossStitch | HiFi, ONT-UL and HiC | CrossStitch | Ref-based to GRCh38 and HapCUT2 | JHU |
| *Diploid contig assemblies* | | | | | |
| asm6a,b | Trio Flye ONT std | ONT | Trio Flye | NA | NHGRI |
| asm7a,b | Trio Flye ONT-UL | ONT-UL more than 100 kb | Trio Flye | NA | NHGRI |
| asm19a,b | Trio HiCanu | HiFi | Trio HiCanu | NA | NHGRI |
| asm20a,b | Trio HiPeregrine | HiFi | Trio Peregrine | NA | NHGRI |
| asm9a,b | Trio hifiasm | HiFi | Trio hifiasm | NA | DFCI Harvard |
| asm11a,b | DipAsm HiRise | HiFi and HiC | Peregrine | NA | UCPH |
| asm3a,b | Peregrine HiFi 25 kb | HiFi long | Peregrine | NA | FBDS |
| asm4a,b | Peregrine HiFi 20 kb | HiFi | Peregrine | NA | FBDS |
| asm16a,b | FALCON Unzip | HiFi | FALCON unzip | NA | PacBio |
| asm8a,b | HiCanu | HiFi | HiCanu and Purge_dups | NA | NHGRI |
| *Merged haploid contig and scaffold assemblies* | | | | | |
| asm5 | Flye ONT | ONT and HiFi | Flye | Flye | UCSD |
| asm18 | Shasta ONT HiRise | ONT-UL and Hi-C | Shasta | HiRise | UCSC-CZI |
| asm21 | Shasta ONT Salsa | ONT-UL and Hi-C | Shasta | Salsa2 | UCSC-CZI |
| asm15 | MaSuRCA Flye ONT | ONT-UL more than 120 kb and HiFi | Flye | Reference based to GRCh38 and MaSuRCA | JHU |
| asm1 | MaSuRCA Combo | Old ONT, Ill and HiFi | MaSuRCA | Reference based to GRCh38 and MaSuRCA | JHU |
| *Merged haploid contig assemblies* | | | | | |
| asm3a | Peregrine HiFi 25K | HiFi long | Peregrine | NA | FBDS |
| asm4a | Peregrine HiFi | HiFi | Peregrine | NA | FBDS |
| asm13 | wtdbg2 HiFi | HiFi and Ill | wtdbg2 | NA | CAAS-AGIS |
| asm12 | NECAT ONT | ONT (no UL) | NECAT | NA | Clemson |
| *Final diploid* | | | | | |
| HPRC mat,pat | Trio HPRC v1.0 | HiFi, ONT-UL, BN and Hi-C | Trio hifiasm | Trio based: Bionano Solve, Salsa, gap fill and curated | HPRC |

Listed are the 23 assemblies generated, categorized into four broad types based on whether there were diploid or merged haploid, and scaffolded or contigs only. Details on sequencing technologies are in Supplementary Table 1. Details on assemblers are in Supplementary Table 2a,b. NA, not applicable.

assemblies is presumably due to differential read length thresholds used for initial contig assembly; the higher the size threshold, the less likely MT reads will be included[37].

## Highly contiguous phased assemblies

Our assembly targets were an expected maternal genome size of approximately 3.06 Gb (22 autosomes + X) and paternal size of approximately 2.96 Gb (22 autosomes + Y), given the expected X (155.3 Mb) and Y (approximately 60 Mb) difference of about 96 Mb[38]. Almost all assemblies, including the diploid assemblies, were close to the expected sizes of a human genome (approximately 3.0 Gb; range 2.8–3.1 Gb; Extended Data Fig. 2a–c and Supplementary Table 2d–f). Only the diploid pair asm19a and asm19b were bigger, by approximately 3%. In the trio-based assemblies, the maternal (mat) haplotypes were all longer than the paternal (pat) haplotypes, consistent with sex chromosome differences. In the non-trio diploid assemblies, each haplotype was more similar in length, skewed towards the expected size of the maternal haplotype, consistently finding either X and part of Y in both haplotypes or missing Y altogether (Supplementary Table 2d, assessed for the diploid scaffolded assemblies). The assemblies that came closest to the theoretical size (98–100%) for both maternal and paternal haplotypes were the Trio VGP scaffolded (asm23a,b) and the Trio hifiasm (asm9a,b) assemblies (Extended Data Fig. 2a). The scaffolded assemblies had quite a range, approximately 40 kb to 50 Mb, of missing sequence (total Ns), in the gaps between contigs and trailing Ns at scaffold ends (Extended Data Fig. 2c and Supplementary Table 2f). In comparison, GRCh38 has approximately 151 Mb of N bases. With the exception of Bionano optical maps, most scaffolding tools place arbitrary gap sizes. Most assemblies also had between 0.3% and 2.3% false duplications (according to *k*-mer counts; Extended Data Fig. 2d and Supplementary Table 2g), the highest in asm19a and asm19b, which could explain why they were bigger than expected[11]. GRCh38 also still contains false duplications[5,39], although difficult to estimate precisely due to the complex mixture of haplotypes.

In terms of continuity, our goal was to minimize the number of gaps for a theoretical maximum gapless contig NG50 that equals chromosome NG50 of approximately 155 Mb for human (in which half of the

assembled contigs are this size and bigger)[5]. Most assemblies had contig NG50 sizes in the range of 20–50 Mb (approximately 13–32% of the theoretical maximum), including for both haplotypes of some of the diploid assemblies (Fig. 1a and Supplementary Table 2e), indicating partial chromosomal length contigs. Exceptions well below NG50 of 20 Mb were: the alternative (alt) haplotypes from the FALCON Unzip or HiCanu approaches that generate a partial diploid assembly by design (asm16b and asm8b, respectively), with the primary pseudohaplotype being more contiguous (asm16a and asm8a); both haplotypes of the Dovetail implementation of the DipAsm assembler (asm2 and asm22), in which Hi-C data were used to phase the haplotypes. By contrast, the original implementation of DipAsm created two assemblies with contig NG50s greater than 20 Mb (asm10a,b). Not surprisingly, the assembly (asm7a,b) that used the ONT ultralong (ONT-UL) reads (more than 100 kb) had the highest contig NG50s (48.6 Mb maternal and 39.8 Mb paternal). The trio-based ONT and hifiasm (asm9a,b) HiFi assemblies had the fewest contigs (approximately 600–900) of all diploid assemblies (Extended Data Fig. 3a). All scaffolded assemblies had scaffold NG50 values ranging from 80 to 155 Mb (Fig. 1b; 52–100% of the theoretical maximum). All non-trio diploid scaffolded assemblies had 23–30 scaffolds, at or close to the expected 23 chromosomes per haplotype (Supplementary Table 2f). However, this particular metric comparison is made less informative as DipAsm inherently filters out scaffolds less than 10 kb, Phased Genome Assembly using Strand-seq (PGAS) excludes contigs less than 500 kb as the Strand-seq signal is too sparse to scaffold small contigs, and CrossStitch only includes contigs or scaffolds that align to the GRCh38 reference. The Trio VGP scaffolded assembly (asm23a,b) that did not exclude scaffolds on the basis of size or alignment to a reference, had, not surprisingly, a much higher number of scaffolds (over 2,000 each) but fewer gaps among those scaffolds (673 maternal and 917 paternal) relative to DipAsm and PGAS assemblies (900–4,000 within scaffold gaps; Extended Data Fig. 3b,c). The size of the largest scaffold (max) for most assemblies approached the size of chromosome 1 (248 Mb; range of 132–242 Mb; Supplementary Table 2f). Together, these findings demonstrate an important shift in recent assembly tools to generate two separate chromosome-level assemblies per individual, representing the two haplotypes or pseudohaplotypes, albeit with gaps.

Despite the high levels of contiguity among the assemblies, manual curation using gEVAL alignments[40], Bionano maps and Hi-C interaction plots (Extended Data Fig. 4a) revealed a handful to several hundred scaffolding errors per assembly, including: missed joins, contigs that should have been brought together as neighbours in the same scaffold; misjoins, colocalized contigs within scaffolds that do not belong together; and erroneous inversions or false duplications classified as other errors (Supplementary Table 2h and Supplementary Fig. 2a–c). There were also within contigs errors: chimeric joins without a gap; sequence expansions; and sequence collapses (Supplementary Table 2h and Supplementary Fig. 2d). There was no one approach, without using a highly curated reference (that is, CrossStitch or MaSuRCA; asm1, asm15 and asm17), that was free of one or more scaffold or contig errors in an automated process. For a complementary, quantitative measure of structural accuracy, we used Strand-seq data, generated by a method that selectively sequences the plus (Crick) and minus (Watson) strands of genomic DNA from cultured cells[41,42]. Nearly all assemblies had 1–25 (average of 6.5) misorientation errors (inversions or reverse complements), totalling from 1 to approximately 746 Mb (Extended Data Fig. 5a and Supplementary Table 2i). An exception was asm14, which used Strand-seq for scaffolding. The non-Strand-seq assembly with the least misorientation errors was Trio hifiasm (asm9a,b), with only one to two small inversions. Over half of the assemblies had 1–9 chimeric contig errors (average of 2.6), with the Trio hifiasm paternal (asm9a) assembly having the most (Extended Data Fig. 5a and Supplementary Table 2i). Overall, each approach avoided at least one type of error that others did not.

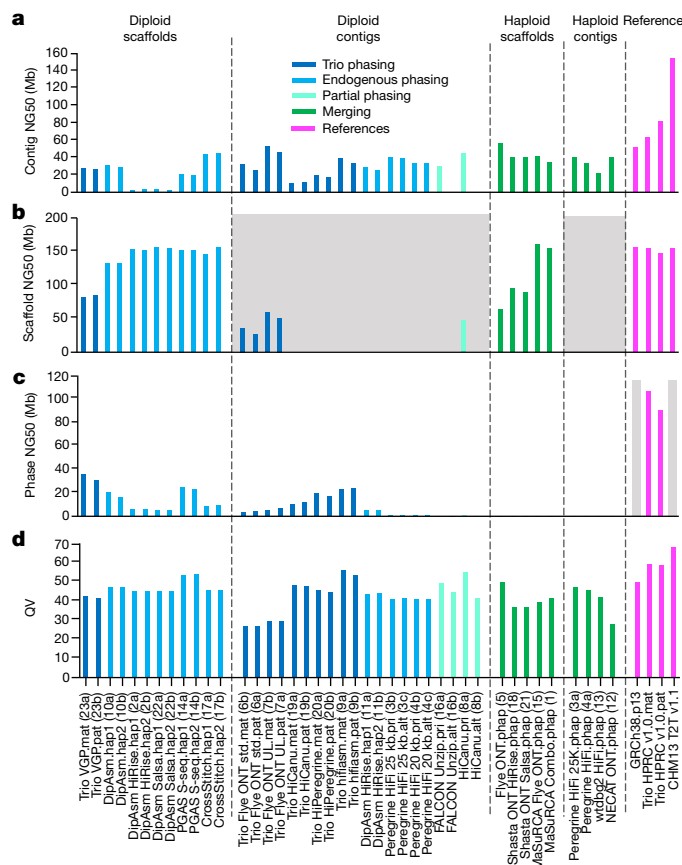

**Fig. 1 | Assembly continuity, phasing and base call accuracy metrics.**
**a**, Contig NG50 values. **b**, Scaffold NG50 values. **c**, Haplotype phase block NG50 values. **d**, QV base call accuracy; as an example, QV60 is about one error per megabase. The dashed lines separate the assemblies into the four major categories as described in Table 1. The colours designate the type of haplotype phasing performed: Trio phasing using parental data, endogenous phasing using self-data, partial endogenous phasing, merging of haplotypes, and final references with various phasing approaches. The grey shaded regions in **b** are not applicable for scaffold metrics, as these are contig-only assemblies; however, the Flye assembler inserts gaps into contigs where there is uncertainty of a repeat sequence, and the purge_dups function applied to the HiCanu contigs removes false duplications within contigs and creates a gap in the removed location. The grey shading in **c** indicates not applicable for phase blocks, because GRCh38 has many haplotypes and CHM13 is from a haploid (hap) cell line. The numbers in parentheses along the *x* axis are the assembly numbers. alt, alternate; mat, maternal; pat, paternal; phap, psuedo-haplotype; pri, primary; std., standard ONT read length; S-seq., Strand-Seq; UL., ultra-long ONT read length.

## Consensus base accuracy

Assembly base accuracy is critical for subsequent annotation of protein-coding genes and non-coding regulatory DNA, as well as for the characterization of genetic variation. To estimate base accuracy, we compared *k*-mer frequencies between unassembled Illumina sequencing reads and each assembly. PGAS Strand-seq (asm14a,b) achieved the highest consensus base accuracy (QV) among scaffolded diploid assemblies, whereas Trio hifiasm (asm9a,b) and HiCanu (asm8a) achieved the highest among the contig-only diploid assemblies (QV or 50 or higher, or no more than 1 base call error per 100,000 bp; Fig. 1d and Supplementary Table 2j). Among the merged haploid assemblies, Fly ONT.phap (asm5) performed best, with two rounds of base call polishing each with ONT and HiFi reads. What these four assemblies share in common is the use of HiFi reads, either for high-level read or contig filtering (asm14a,b and asm8a), polishing (asm5), and/or phasing

of haplotypes (asm9a,b). Obtaining such a high degree of base accuracy (QV of 50 or higher) with long reads has only been a recent advance, due to the higher accuracy of HiFi reads[17].

## Variant benchmarking

To determine how well each assembly correctly reveals haplotype variation, we developed a benchmark variant calling pipeline. We aligned each assembly to GRCh38, used dipcall[41] to call variants and compared them to a manually validated ground truth, the v4.2.1 small variant HG002 benchmark from GIAB[26], following the Global Alliance for Genomics and Health (GA4GH) benchmarking best practices[42]. For the haploid assemblies, we developed separate performance metrics that ignore genotype errors (when only one haplotype has to match the benchmark variant). We found that all diploid-based assemblies had high true-positive rates above 90% for SNVs, whereas the haploid assemblies were all around 40%, due to merging of haplotypes that exclude many heterozygous variants (Extended Data Fig. 6a and Supplementary Table 3a). As expected, the haploid assembly values were higher (65–74%) when ignoring genotype (Supplementary Table 3a). The Trio hifiasm diploid assembly (asm9) had the highest true-positive rate (99.47%). When examining variants in the harder-to-assemble segmental duplications, most of the diploid assembler performances dropped by 9–32%, whereas the Trio hifiasm and Trio HiCanu dropped by only 5–6% (Supplementary Table 3a). When we assessed the accuracy of small insertions or deletions (indels; less than 50 bp) between haplotypes, which are particularly problematic and highly variable due to their association with short tandem repeats, all HiFi-based diploid assemblies outperformed (true positive of approximately 92–98%) the haploid assemblies (approximately 38–59%), as well as the ONT diploid assemblies (about 52–58%; Extended Data Fig. 6b and Supplementary Table 3b); the latter was due to the high indel error rate in ONT reads. The Trio hifiasm (asm9a,b) assembly had the highest combination of true-positive rates for both SNVs and small indels.

As a result of these findings, the Trio hifiasm assembly was used to further improve the GIAB benchmark for SNVs, small indels and larger SVs (indels, inversions and translocations) in 273 challenging, medically relevant genes that were not well represented in the GIAB v4.2.1 benchmark or the GIAB v0.6 SV benchmark. Extensive curation by GIAB found that the Trio hifiasm assembly produced more accurate variant calls across SNVs, small indels and SVs in these challenging regions, and the primary error type fixed was inaccurate genotypes in highly homozygous regions, particularly for indels in long homopolymers[43]. These results demonstrate that diploid assemblies are not only highly concordant but exceed existing variant benchmarks in regions resolved by mapping-based methods. Thus, they show the greatest promise for resolving more challenging regions and variants not included in current benchmarks.

## Annotation

We performed annotation for each assembly by aligning the human NCBI RefSeq transcriptome dataset of 78,492 transcripts from 27,225 autosomal genes to them, and measured mapping statistics, using GRCh38 and CHM13 assemblies as controls. Most of the HG002 assemblies had 100–400 genes with no transcript alignment (over 1,600 for the haploid wtdbg2 asm13 assembly; Extended Data Fig. 7a and Supplementary Table 2k). Exceptions were the Trio VGP (asm23a,b), Trio HiCanu (asm19a,b), Trio hifiasm (asm9a,b) and reference-based assemblies (asm1, asm15 and asm17) with only approximately 60–70 unaligned genes for each haplotype, twice the missing number of 36 for GRCh38 but similar to 66 missing genes for CHM13. There were about a dozen genes present in GRCh38 and asm17 that used it as a reference, but not in any of the other HG002 assemblies or CHM13, showing a bias of false gene presence (presumably gap filled from GRCh38) for reference-based assembly methods. Most of the contig-only assemblies had more genes (approximately 100–500) split between contigs than the scaffolded assemblies (Extended Data Fig. 7a and Supplementary Table 2k), consistent with scaffolding bringing separate parts of more genes together. The Trio VGP (asm23a,b) scaffolded assembly and Trio hifiasm (asm9a,b) contig-only assembly had the fewest split genes (approximately 30–40) among the de novo assemblies, the reference-based assemblies had even fewer (1–9) and even less than GRCh38 (10 genes). Most assemblies had 100–700 genes (over 4,000 in the alts of asm16b and asm8b) that were less than 95% complete, except for the Trio VGP, Trio hifiasm and reference-based assemblies with only 32–89 incomplete genes (Extended Data Fig. 7b). For almost all assemblies, there were 200–600 genes apparently collapsed as assessed by overlapping transcript mapping, with those that used HiFi having the least collapses (Extended Data Fig. 7c). Similarly, the number of genes that required frameshift error corrections were approximately 1,000 for assemblies that used continuous long reads (CLRs; Trio VGP, asm23a,b), about 1,500 that used the 25-kb longer but less accurate HiFi reads (asm3 and asm4), approximately 6,000–16,000 (more than half of the genes) that used unpolished ONT reads, but only about 100–200 genes with the shorter (15 kb) but more accurate HiFi reads (Extended Data Fig. 7a and Supplementary Table 2k). These findings demonstrate that a critical combination of read length, base accuracy, structural accuracy and haplotype phasing are necessary to obtain the most complete and accurate annotation possible.

## Trios and higher phasing accuracy

The original Trio assembly approach of binning long reads into their respective maternal and paternal haplotypes before generating contigs was implemented with the Canu contig assembler, as TrioCanu[22]; but this approach had not yet been tested in a head-to-head comparison with different assemblers and data types. Here we tested haplotype-binned reads with different contig assembly algorithms (Flye, HiCanu, hifiasm and Peregrine), different long-read data types (HiFi, CLR and ONT) and with trio-sorted scaffolding data types (10X-linked reads, optical maps and Hi-C). We found that all trio-based approaches yielded higher phasing of the same haplotype than their non-trio counterparts. Trios that used HiFi or CLR data had the largest NG50 haplotype phase blocks (approximately 10–30 Mb versus less than roughly 0.2–5.0 Mb; Fig. 1c), the lowest haplotype switch errors within contigs or scaffolds (about 0.01–0.02% versus 0.20–7.3%; Extended Data Fig. 8a), the highest number of phased bp (Extended Data Fig. 8b) and the most complete separation (approximately 99%) of paternal and maternal haplotype k-mers when using HiFi reads (Extended Data Fig. 8c and Supplementary Table 2l,m). Several of the trio approaches (Trio HiCanu and Trio hifiasm) yielded the least collapsed sequence (Extended Data Fig. 9a–c and Supplementary Table 2n). The only non-trio method that approached the phasing accuracy for maternal and paternal alleles of a trio method used Strand-seq for phasing and scaffolding (asm14; Fig. 1c and Extended Data Fig. 8a), but it suffered from having the highest within-scaffold errors (Supplementary Fig. 2c). The trio-based ONT contig assemblies had lower haplotype phase blocks (NG50s of approximately 3–6 Mb; Fig. 1c) and higher haplotype switch errors (approximately 0.3–0.5%; Extended Data Fig. 8a), presumably owing to their higher sequence error rates. In contrast to previous findings[11], the VGP trio assemblies did not have the lowest haplotype false duplication rates, as assessed by either k-mers or BUSCO duplicate gene copies (Extended Data Fig. 2d and Supplementary Table 2g). This appears to be due to improvements in the higher read accuracy of PacBio HiFi versus CLR; the latter was used for the VGP trio assembly.

## Graph phasing is more complete and accurate

The trio-based approaches fell into two principal categories: (1) those that use parental reads to haplotype bin the reads of the child before assembly

(for example, Trio VGP, Trio Flye, Trio HiCanu and Trio Peregrine); or (2) those that generate an assembly graph of the genome of the child first and then label haplotypes in the graph using the parental reads (for example, Trio hifiasm). As presented in a complementary study conducted simultaneously[31] and further advanced here, we found that the graph-based phasing approach generally outperformed the two-step binning trio approach when high-accuracy long reads were used to build the initial assembly graph. In particular, among the diploid assemblies, the Trio hifiasm maternal (asm9a) and paternal (asm9b) assemblies had the highest combination of high-quality metric values, including the highest QV (Fig. 1d), the third highest NG50 haplotype phase blocks (Fig. 1c; Trio VGP was the highest), the highest genome completeness (Supplementary Table 2k), among the least false duplications (Extended Data Fig. 2d), the fewest contigs (Extended Data Fig. 3a), among the lowest haplotype switch errors (Extended Data Fig. 8a) and the least collapsed repeats (Extended Data Fig. 9a,b). These findings indicate that graph-based phasing of the assembly is more accurate and complete as the combination of the graph with haplotype information can correct errors made by either method alone. A prerequisite to highly accurate graph-based haplotype phasing is a well-resolved diploid assembly graph, as generated from high-accuracy long reads (for example, HiFi).

## Pan-assembly alignment

To identify both shared and distinct features of the assemblies, we utilized a pangenomic approach, performing an all-versus-all alignment for 45 assemblies (both haplotypes; Extended Data Fig. 10a), excluding the alternate contigs or unitigs of pseudohaplotype assemblies as they were highly fragmented. We annotated the alignment according to chromosomes in GRCh38 and CHM13. Pairwise Jaccard similarity analyses on the autosomes (chromosomes 1–22) clustered the Trio hifiasm and Trio HiCanu assemblies as more similar to each other and distinct from the other assemblies (Fig. 2a); at one branch higher, these trio assemblies clustered with the other trios (except Trio HiPeregrine) and with the MaSuRCA and CrossStitch reference-based assemblies. The remaining assemblies subclustered mostly by assembly pipeline, indicating that assembly approach drives their similarities the most. More pronounced than the autosomes, Jaccard similarity analyses on the XY sex chromosomes grouped all trio-based paternal assemblies into one cluster, with distinctions among themselves, relative to all of the remaining assemblies into a sister supercluster with the trio-based maternal assemblies (Fig. 2b). This finding is consistent with chromosome X and part or none of chromosome Y being present in both haplotypes with non-trio assemblers (Supplementary Table 2d). Two exceptions were the haploid Flye ONT.phap assembly (asm5) and the reference-based CrossStitch hap1 assembly (asm17a), which grouped with the trio paternal assemblies and had a more complete Y chromosome (asm17a) due to using the GRCh38 Y chromosome as a reference. Principal component analysis (PCA) on Euclidean distances between assemblies supported these conclusions, in which the trio-based autosomes (concatenated 1 through 22) clustered by parental haplotype without the presence of the sex chromosomes in the fourth dimension (Fig. 2c,d and Supplementary Fig. 3a,b with reduced labels). The Trio hifiasm and Trio HiCanu autosome assemblies were the most distinctly clustered by parental haplotype. Clustering on each autosome alone and then performing a machine learning algorithm (support vector classifier) to find whether a dimension with a hyperplane that distinctly and maximally separates the trio-based maternal and paternal haplotypes exists, revealed such a dimension (first to ninth, most often the second), explaining 3–12% of the clustering variance (Supplementary Table 4). The degree of separation (that is, PCA % variance) negatively correlated with the relative size of the centromere for each autosome (Fig. 2e). These findings indicate that the trio-based assemblies have the maximal separation of parental haplotypes, the centromeres contribute less to this signal, and this serves as a benchmark for further developing tools for better separation of haplotypes in non-trio assemblies.

## High-quality HPRC-HG002 diploid reference

On the basis of our findings, we developed a pipeline that combines the best practices of all approaches and used it to generate a higher-quality diploid de novo assembly (Extended Data Fig. 10b). We first removed the remaining HiFi reads with unremoved vectors (adaptors) using HiFiAdapterFilt (Supplementary Note 2). We then generated HiFi maternal and paternal contigs with the graph-based haplotype phasing of Trio hifiasm v0.14.1. This updated version incorporates bug fixes that we found after generating the initial HG002 assemblies, including: (1) enhancing contig QV by constructing the contig golden path through high-quality portions of error corrected reads; (2) resolving more segmental duplications by selecting high-occurrence seeds at the overlapping stage; and (3) improving contig N50 by rescuing contained reads that break contigs on one haplotype when the read actually comes from the other haplotype[44] (Supplementary Fig. 4). In addition, we titrated child and parental coverages with hifiasm and found a level (approximately 130× child HiFi; approximately 300× parent Illumina) given the data that yielded an optimal contiguity and the lowest haplotype switch error (Supplementary Fig. 4). We then separately scaffolded the maternal and paternal HiFi-based contigs with maternal and paternal Bionano optical maps. Conflicts between the HiFi contigs and Bionano optical maps were manually evaluated (curation 1), in which we accepted 5 of 15 maternal and 3 of 13 paternal joins or breaks indicated by the Bionano maps (Supplementary Table 5a). The majority of these conflicts (25 of 28) were in segmental duplications and centromeres, particularly of the acrocentric chromosomes (chromosomes 15, 21 and 22), and included haplotype SV differences in HG002; the remaining three were in known tandemly repeated genes (*IgK*, *IgH* and *TSP*), where the first two were processed by programmatic structural variation associated with B lymphocytes. We then further scaffolded the paternal and maternal assemblies with haplotype-filtered (Meryl) Hi-C (Dovetail OmniC) data and the Salsa 2.3 algorithm. Scaffolding with Arima Hi-C v2 data yielded similar results. We performed manual curation (curation 2) using Hi-C contact maps, which resulted in 7–8 scaffold breaks and 44–50 additional joins in each haplotype assembly (Extended Data Fig. 4b and Supplementary Tables 2h and 5b). Most of the breaks were at centromeres to allow satellite placement.

Next, we filled gaps with a conservative version of the pipeline used in the initial T2T-CHM13 assembly[5]. ONT-UL reads were base recalled with Guppy 4.2.2, haplotype binned using trio-Canu and assembled into haplotype-specific contigs using Flye. Draft ONT-UL contigs were polished to increase consensus accuracy. Variant calls were generated using Medaka on ONT long reads, and filtered with Merfin[45] using *k*-mers from Illumina short reads and then applied to increase the quality of the consensus sequence. The polished contigs were aligned to their respective haplotypes of the curated HiFi-based scaffolds from the Hi-C step above and used to fill gaps. This resulted in ten and five gaps filled in the maternal and paternal assemblies, respectively. Of these 15 gaps, 10 contained GA-rich repeats and 2 were long segmental duplications (Supplementary Fig. 5). The final manual curation (curation 3) fixed 37 items in the maternal and 60 in the paternal assemblies (Supplementary Tables 2h and 5c), much fewer than the hundreds of manual fixes that normally would be required (for example, Extended Data Fig. 4a). A contamination screen removed multiple (41 maternal and 45 paternal) human EBV viral genomes (contigs) used to transform the LCLs as well as a yeast contig in the paternal assembly; we did not find any non-human contamination within the human contigs and scaffolds. Approximately 98% of the remaining sequence was assignable to the 22 autosomes and the X and Y sex chromosomes (Fig. 3a). These new assemblies were named HPRC-HG002.mat.v1.0 and HPRC-HG002.pat.v1.0 references.

These two de novo assemblies exhibited the highest quality across most metrics, compared with the bakeoff assemblies and the GRCh38 reference: the largest contig (62.9 and 81.6 Mb) and comparable scaffold (154.4 and 146.7 Mb) NG50s, close to the theoretical scaffold

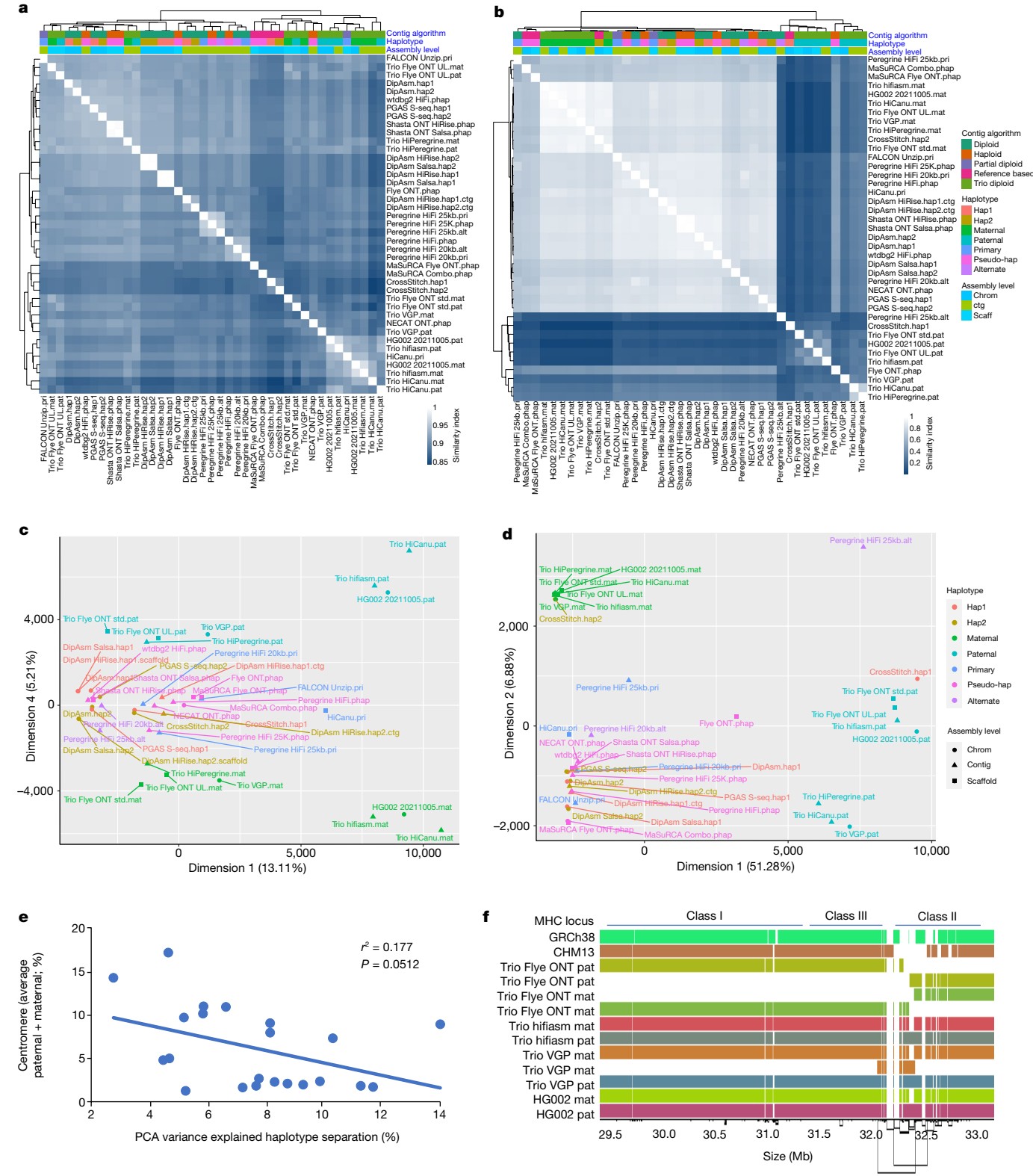

**Fig. 2 | Multidimensional relationship among assemblies. a,b**, Clustering of pairwise Jaccard similarities between pairs of assemblies, for the autosomes 1–22 (**a**) and the X and Y sex chromosomes (**b**). In the heatmap, the lighter the blue (Jaccard similarity index closer to 1), the more similar the assemblies (1 indicates identical assemblies). Assemblies are annotated with four different colour-coded classifications. **c,d**, PCA on the multidimensional Euclidean distances among assemblies, for the autosomes 1–22 (**c**) and the X and Y sex chromosomes (**d**). PCA dimensions shown are those in which the paternal and maternal haplotypes separated the strongest. **e**, Correlation between centromere size relative to chromosome size (%) and PCA variance (%) in the dimension where the Trio-based autosome assemblies separated by haplotype. **f**, Graph-based alignment of a 5-Mb region of human chromosome 6 containing the MHC locus of the Trio-based assemblies and GRCh38 and CHM13 references. Each colour is a different assembled haplotype. The Trio hifiasm assembly and the final HG002 assembly that used Trio hifiasm assembled the entire MHC locus in one single contig.

maximum (Fig. 1a,b); the fewest contigs and gaps in scaffolds (Extended Data Fig. 3a–c); the highest QVs (approximately 60; Fig. 1d); the most complete haplotype phasing (Fig. 3b and Extended Data Fig. 8a–c) with NG50 phase blocks of 106.7 and 90.4 Mb, respectively (Fig. 1c and Supplementary Table 2j); the least collapsed repeats (18.5 and 17.6 Mb, respectively; Extended Data Fig. 8a,b); among the highest values in annotation metrics (Extended Data Fig. 7a–d and Supplementary Table 2m); and among the highest SNV and small indel true-positive rates (Supplementary Table 3a,b). They clustered closest with the Trio hifiasm and Trio HiCanu assemblies (Fig. 2). Assessing against GIAB HG002 benchmarks against GRCh38, this diploid assembly produced highly accurate SNV concordance (F1 score) of 99.7% and small indel concordance of 98.6%, which were 0.2% and 0.8% lower, respectively, than the best-performing mapping-based variant callers in a 2020 precision FDA Truth Challenge[46]. We found that 70% of the discordant SNVs fell in segmental duplications, most with complex SVs that could not be accurately benchmarked. In fact, many of these differences appeared to be more accurate in the new HPRC-HG002 assemblies than in the mapping-based benchmark or precision FDA entries. The primary limitation of the assemblies was small indels in homopolymers and in 51–200-bp tandem repeats, making up 80% of all discordant indels; curation revealed that the final HPRC-HG002 assemblies had infrequent errors due to collapsing haplotypes and/or to noise in the starting HiFi reads. When benchmarking larger SVs in the new HG002 assemblies with respect to the GRCh37 GIAB v0.6 SV benchmark, which excludes segmental duplications and centromeres[47], the true-positive rate was 98% (compared with 93% for asm9a,b) and precision was 89%, with most putative errors just differences in SV representation in tandem repeats or errors in the benchmark. Some known difficult-to-assemble repetitive gene families were completely assembled in one contig, including the approximately 5-Mb histocompatibility complex (MHC) containing over 220 genes (Fig. 2f), in which variants were more than 99.99% concordant with the GIAB v4.2.1 benchmark. Overall, this high concordance between the assembly-based variants, existing benchmarks and higher accuracy than the benchmarks, demonstrates substantial promise for phased, whole-genome assemblies.

Performance in most metrics, particularly for the HG002 maternal haplotype, were on par with the T2T-CHM13 v1.1 assembly (Fig. 1 and Extended Data Figs. 2,3 and 7–9), including comparable Hi-C profiles (Fig. 3a). We aligned the two HG002 haplotype assemblies to CHM13 (with Y from GRCh38), and found high correlations (Supplementary Table 6). Most assembled HG002 chromosomes (32 of n2 = 46) were 98.0–99.9% complete (not including gaps) relative to the length of CHM13 (Fig. 3c,d). Chromosome 9 was the expected size, but 10% smaller than in CHM13 due to a known approximately 10-Mb large satellite duplication in CHM13 (ref. [5]). The biggest exceptions were the short arms of the acrocentric chromosomes, with chromosomes 21 and 22 being the two outliers at approximately 85% of the length of CHM13 for the maternal and about 75% for the paternal haplotype (Fig. 3c,d); the short arms of these chromosomes are notoriously difficult to assemble owing to their highly repetitive shared structure consisting of rDNA arrays, satellite arrays and segmental duplications[5]. Yet, the remainder of the paternal chromosomes 21 and 22, as well as maternal chromosomes 11 and 12 had no gaps, and the remaining autosomes had an average of four gaps each (range 1–12; Fig. 3e and Supplementary Table 6). Most of these gaps were in centromeres or acrocentric regions (Fig. 4a,b). All HG002 unplaced or unlocalized scaffolds that mapped to CHM13 were in the centromeres, especially of the acrocentric chromosomes (chromosomes 13, 14, 15, 21 and 22) or telomeres (asterisk in Fig. 4a). The centromeres also had the greatest amount of unaligned sequences due to greater divergence between HG002 and CHM13 haplotypes (Fig. 4a); the two ends of the Y chromosome aligned to CHM13 X chromosome, because the psuedoautosomal region at the ends of the HG002 Y chromosome has higher identity to the CHM13 X chromosome than to the GRCh38 Y chromosome.

To determine whether any of the chromosomes were T2T complete, we examined hard-to-assemble regions, centromeres and telomeres. Diploid HiFi sequence coverage and *k*-mer analyses revealed that the centromeres of 5 of 46 chromosomes (maternal 11, 12 and 16 and paternal 21 and 22) had no haplotype switch errors, no collapsed repeats and no gaps (Extended Data Fig. 9d, Supplementary Table 7a–c and Supplementary Figs. 6 and 7). We found complete canonical telomere repeats (TTAGGG) on the q and p arms for six maternal and ten paternal chromosomes, whereas nearly all others had one or the other arm (Extended Data Fig. 11a–c and Supplementary Table 7d). The approximately 70 unlocalized scaffolds on chromosomes and the several hundred remaining small unplaced scaffolds without a chromosome were largely centromeric satellites and telomeric repeats (Supplementary Table 7e). Overall, although there was no chromosome that was T2T, most were near complete, with few errors in centromeres or missing telomeres. These findings highlight that a mostly automatically generated, haplotype phased and near T2T assembly is now possible, and the remaining development needed is for the centromeres and telomeric ends. These two assemblies are available without restrictions in the INSDC archives under accession numbers GCA_021951015.1 (maternal) and GCA_021950905.1 (paternal).

## Missing genes among haplotypes

From the annotation analyses of 27,225 autosomal genes, we identified 106 genes that are completely missing from one or more of the four reference assemblies: GRCh38 (32 genes), the HG002 haplotypes (61 maternal and 65 paternal genes) and T2T-CHM13 v1.1 (62 genes; Supplementary Table 8). Among these, 20 genes were absent from all four assemblies. There was greater overlap of 74% (46 of 62 genes) not present in CHM13 and one or both HG002 haplotypes (Fig. 5). The inverse had lower overlap, with 64% (39 of 61) for the HG002.mat and 62% (40 of 65) for the HG002.pat haplotype also absent in CHM13. Similarly, the maternal and paternal haplotypes of HG002 shared 66% (40 of 61) and 62% (40 of 65) of gene loss with each other, respectively. Conversely, CHM13 and each HG002 haplotype had 11–17 genes absent specific to them (Fig. 5). However, 51 of the total HPRC-HG002.pat unaligned genes were present in one or more of the Trio paternal bakeoff assemblies, indicating that either they were missed in the HPRC-HG002.v1 reference assemblies or they were false haplotype duplications in the bakeoff assemblies. False duplication is possible given that two-thirds of the 106 genes missing among the four reference assemblies were in repetitive gene families (Supplementary Table 8), including the MHC HLA immune cluster, keratin-associated proteins, olfactory receptors and 18S and 5–8S RNA genes. There were also several long intergenic non-protein coding RNA genes and over 30 microRNA genes. The absences cannot also be explained by annotation artefacts (Supplementary Note 3). Overall, these findings indicate a diversity of missing genes, including repetitive genes, among individuals and haplotypes within an individual.

## Greater diversity between haplotypes

With a more complete diploid human assembly, we performed heterozygosity analysis between haplotypes, following approaches that we used on a VGP Trio-based marmoset assembly[48]. We noted a remarkably high amount of autosomal heterozygosity between haplotypes (3.3% of total bp, including approximately 2.6 million SNVs; about 631,000 small SVs (less than 50 bp); 11,600 large SVs (50 bp or more); or 3,294,604 bp of variants total; Fig. 4b and Supplementary Table 9). Most of the additional variation was in the newly assembled centromeres, with sharp peaks in SNVs, indels, inversions and intrachromosomal translocations (Fig. 4b). This is partially due to the lower alignments in highly repetitive centromeric satellites, which in turn can be due to higher diversity in centromeres between haplotypes. When not including the centromeres, autosomal heterozygosity in total bp was

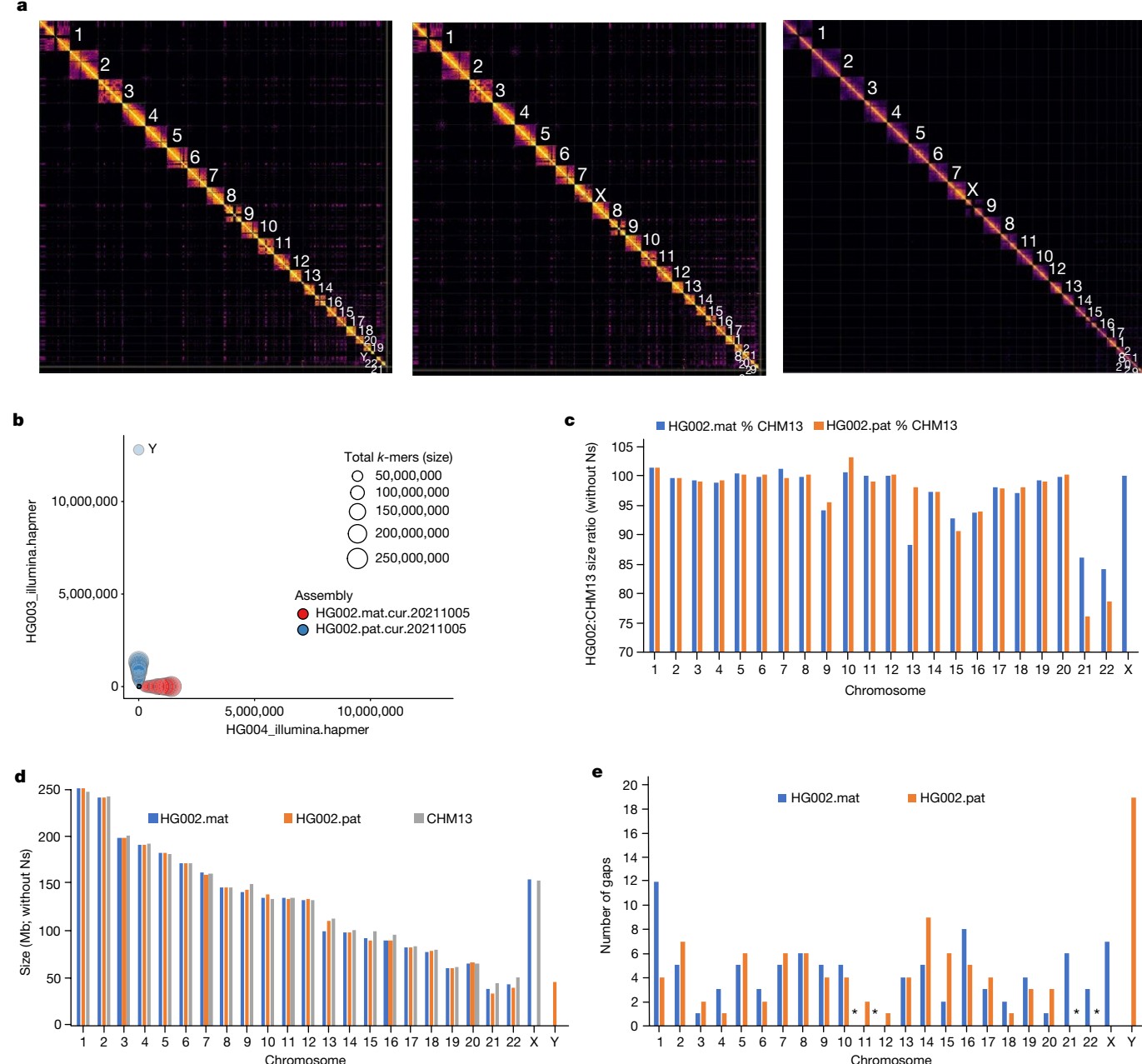

**Fig. 3 | Near-complete haplotype separation of scaffolds. a**, Hi-C contact maps to the final curated HPRC-HG002 paternal (left) and maternal (middle) assemblies in comparison to the CHM13 assembly (right). Values designate chromosome numbers, from largest to smallest size for each assembly. **b**, Blob plot using Illumina parental *k*-mers for the scaffolds of the HPRC-HG002 haplotypes. **c**, Percent size of HG002 diploid assembled chromosomes relative to CHM13 chromosomes, without including Ns. **d**, Comparison of absolute chromosome size values of all three assemblies, without including Ns. **e**, Number of remaining gaps in the chromosomes of each HG002 haplotype. Asterisks indicate assembled contigs with no gaps: maternal chromosomes 11 and 12, and assembled paternal chromosomes 21 and 22 without complete short arms.

approximately threefold less (1.2%; Supplementary Table 9), closer to previous measures between human haplotypes[49]. The increased diversity in the centromeres, although expected, was not seen at this level in the marmoset trio assembly[48]. This difference is probably due to the marmoset assembly using higher error rate CLR PacBio reads, leading to largely collapsed centromeric repeats, as well as to species differences or individual differences. The reason can be resolved with future population-level analyses on assemblies generated using the approaches developed here.

The SVs included 59 large (more than 500 bp) inversions (Fig. 4b and Supplementary Table 10). Of these, 41 had clear Watson–Crick Strand-seq alignment orientations, revealing that 30 inversions had

the correct orientations, but three paternal and eight maternal had the incorrect orientation (Extended Data Fig. 5b,c). The source of these few orientation errors appeared to be long stretches of segmental duplications on either side of the inversions, where either orientation aligns (Extended Data Fig. 5d–f). The SVs included 7,892 copy number variations between haplotypes (Supplementary Table 9), of which 220 were protein-coding gene expansions relative to GRCh38 from 81 gene families (Supplementary Table 11), approximately threefold higher than the average of 75 genes determined from less-complete short-read assemblies from the 1000 Genomes Project[50]. Of these, four genes had remarkable differences in copy number between haplotypes (Fig. 4b): (1) tandem arrays of family with sequence similarity 90 member A

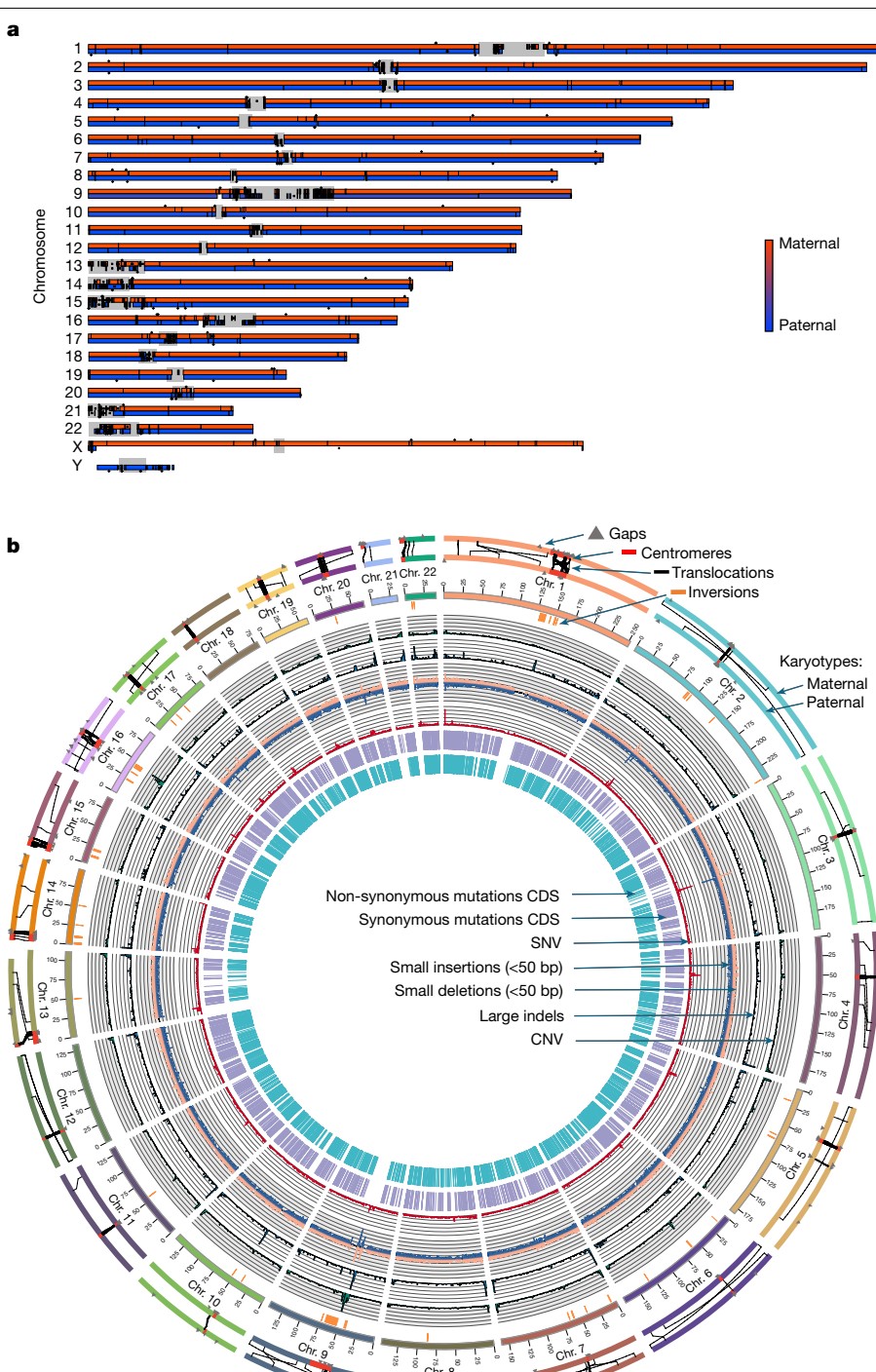

**Fig. 4 | HPRC-HG002 features. a**, Chromosome alignments between HPRC-HG002 maternal (top) and paternal (bottom) assemblies and CHM13 and the Y chromosome of GRCh38. Haplotype separation is nearly complete, and thus colours are solid blue (paternal) and red (maternal). Colour values were determined by the number of aligned haplotype-specific *k*-mers. A few ambiguous alignment blocks (purple) are highly repetitive regions, where it is hard to extract enough haplotype-specific *k*-mers. The black tick marks indicate gaps between contigs. Unaligned regions, which are mostly centromeric satellites, are shown in grey. **b**, Circos plot of the heterozygosity landscape between the two HG002 haploid assemblies. Tracks from inside out:

synonymous amino acid changes; non-synonymous changes; SNV density (window size of 500 kb, range of 0–3.1%), and small deletion and small insertion (less than 50 bp) density (window size of 1 Mb, range of 0–850); large indel density (50 bp or more, window size of 1 Mb, count of 0–20) and copy number variant (CNV) density (window size of 1 Mb, count of 0–77). The black line links in the outermost circles denote intrachromosomal translocations (50 bp or more) between paternal (inner) and maternal (outer) assemblies. The orange bars indicate inversions (50 bp or more), the red bars denote centromeres and the grey triangles indicate gaps. CDS, coding sequence.

(*FAM90A*) members present at 32 maternal, 20 paternal and 16 GRCh38 copies; (2) an expansion of nuclear pore complex interacting protein member B8 (*NPIPB8*) with 6 maternal, 10 paternal and 6 GRCh38 copies;

(3) Tre-2, Bub2p and Cdc16p domain family member 3 (*TBC1D3*) with 11 maternal, 17 paternal and 13 GRCh38 copies; and (4) an expansion of 9 copies of the Kringle domain in lipoprotein A (*LPA*) in the paternal

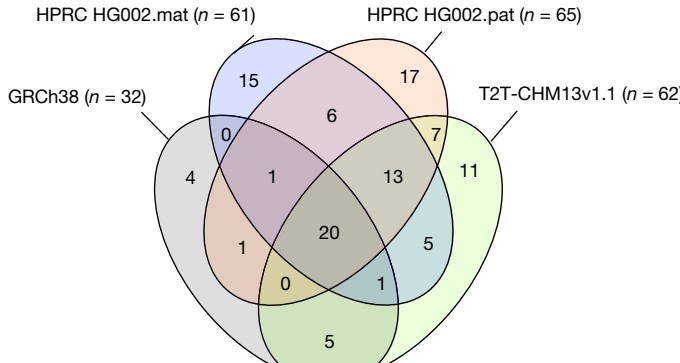

**Fig. 5 | Genes with no aligned transcript and thus presumed absent in the four main reference assemblies compared.** *n* refers to the number of genes absent in each reference assembly. Values in the four-way Venn diagram are the number of shared or uniquely absent genes among the four assemblies.

versus the maternal haplotype (Supplementary Table 11). Raw HiFi read coverage analyses of these genes did not show evidence of collapsed repeats (resolved in Supplementary Table 11), indicating that the haplotype differences are not assembly artefacts. The first two genes (*FAM90A* and *NPIPB8*) are thought to be primate specific or more rapidly evolving in primates[51,52]; *TBC1D3* is only found in great apes, and is associated with increased cortical brain folding and expansion in humans[53]; additional copies of the Kringle domain of *LPA* have been associated with increased atherosclerosis and coronary artery disease[54]. One interpretation of these findings is that in the ancestral primate lineage, duplications of these genes were selected for primate brain-specific traits.

Among the 12,241 SNVs (not including indels) located in CDS that were annotated for both haplotypes, 6,397 (52.3%) SNVs in 4,119 genes were synonymous leading to no change in the amino acid sequence, and 5,844 (47.7%) SNVs in 3,690 genes were non-synonymous, changing the amino acid sequence between haplotypes (Fig. 4b). Of 3,690, 2,466 genes had exclusively non-synonymous differences and were significantly enriched (false discovery rate < 0.01) for metabolism, smell, taste and HSV1 viral infection functions (Supplementary Table 12). These findings are consistent with more rapid evolution of smell and taste receptor genes than the average gene family in some species[55].

A well-phased diploid assembly provides an opportunity to investigate mosaicism within haplotypes. We aligned the Illumina reads against our final diploid reference, called SNVs and found an average minor allele proportion of 0.0466% and 0.0468% for the maternal and paternal haplotypes, respectively (Supplementary Fig. 8a); this is tenfold lower than mosaicism seen in the common marmoset using the same approach[48], a species that has genetic chimerism between twins and triplets in utero. There was a higher prevalence of mosaicism on the smaller chromosomes in HG002 (chromosomes 13–22; Supplementary Fig. 8b), indicative of greater mutational load on them. We also separately compared blood versus LCL genomes of another sample (HG06807), assembled with hifiasm, and did not find evidence of an increase in mosaicism (Supplementary Note 4). We did, however, find three small inversions (1.6–10 kb in size) in the maternal haplotype of the LCL genome (Supplementary Fig. 9). These findings suggest that our measure is of endogenous mosaicism, but there could be rare SV changes in LCLs.

We also assessed whether we could detect MT genome mosaicism (that is, heteroplasmy) by mapping all maternal-derived and paternal-derived HiFi reads. A total of 11,938 HiFi reads aligned to our MT genome assembly. We found six SNPs at more than 1% frequency (above the read error rate), which we interpret as mitochondrial heteroplasmy (Supplementary Fig. 10). In one case, the major allele (T) was supported by 8,033 reads (97%, 4,186+ and 3,847−), whereas the

minor allele (C) was supported by 202 reads (2%, 94+ and 108−). We note that our MT genome assembly represents a consensus of reads with this mosaicism. Overall, a more complete human diploid genome assembly reveals a greater amount of genetic diversity in the nuclear genome than otherwise expected, more copy number variation in genes associated with primate specific-traits, and nuclear and MT genome mosaicism.

## A look towards the future

This study allowed us to determine which current approaches yield the best values in quality metrics for diploid maternal-derived and paternal-derived genomes of an individual. Key factors were the use of trio-parental sequence data to sort haplotype sequences in the child, a graph-based approach to resolve these haplotypes during the assembly process rather than before or after it, and combining different sequence data types and assembly tools in which each approach captures information missed by another. Haplotype binning of reads before assembly (for example, Trio HiCanu) was prone to mispartition of some reads, leading to lower phasing metric values than graphed-based phasing (for example, Trio hifiasm).

These findings confirm and advance on those recently reported by the VGP[11], HGSVC[10] and T2T[5] consortia. The initial VGP pipeline used PacBio CLR reads, which were less accurate than the more recent PacBio HiFi reads. The improved accuracy of the HiFi reads reduces the need for short-read polishing of the assemblies. More accurate long reads also allowed generation of larger contigs, reduction of collapsed repetitive sequences in the centromeres and increased haplotype phasing accuracy[10]. Instead of FALCON-Unzip that had produced a more complete pseudohaplotype and a fragmented alternate haplotype, hifiasm, DipAsm, PGAS and CrossStitch produce two comparable pseudohaplotypes. An advance adopted from the T2T approach used on CHM13 was development of a tool for automated incorporation of polished ONT assemblies for gap filling, but here for both haplotypes, independently. We also made advances on the Trio assembly approach, by not only haplotype phasing the long reads and Hi-C reads but also the Bionano optical maps. These advancements lead to near-complete phased haplotypes. All major components of the current pipeline developed here are available on the Galaxy platform, and in modular form with different steps that can be optionally performed (https://assembly.usegalaxy.eu/)[56]. What remains is developing diploid assembly methods that prevent the remaining collapses, gaps and switch errors in the centromeric satellite arrays, large human satellite arrays and short arms of the acrocentric chromosomes.

On the basis of the findings in this study, the HPRC decided to use the trio graph-partitioning approach of hifiasm to generate the contigs of the first 47 individuals (94 haplotypes) that will contribute to the first human pangenome reference (BioProject PRJNA730822)[57]. The contig assembly metrics on these additional individuals had similar high values as we present here for HG002, indicating that overfitting of algorithms or parameters on one individual did not occur. We initially used 35× HiFi coverage for these individuals based on manufacturer recommendations. However, this was not sufficient to cover all regions of the genome for assembly, and thus we used 130× HiFi coverage. Subsequent tests with improved algorithms on humans and other species suggest that we can lower HiFi coverage from 130× to 50–60× to get the most complete assembly before curation. The trio and a non-trio version of hifiasm followed by the scaffolding with Hi-C (and/or Bionano) used here for HPRC-HG002 have been adopted by other large-scale sequencing projects, such as the VGP, the Earth Biogenome Project and the Darwin Tree of Life Project. Improvements have also been made to some of the other assembly algorithms since the versions tested here thus far[29,58–60]; the trio graph-based approach with trio-based scaffolding still yields the best combination of values in metrics. The results and methods developed here help to set the standard and benchmarks for future studies.

Future efforts will be necessary to develop a phasing method that does not require parental sequence data and works as well as a trio method. This will make it possible to generate equivalent diploid reference assemblies for human and non-human organisms where parental data may not be available. Towards this end, using Hi-C or Strand-seq data for haplotype phasing are promising alternatives, as both types of data contain within-chromosome haplotype information of an individual. To date, three methods have successfully used Hi-C, including FALCON Phase[23], hifiasm (Hi-C)[59] and pstools[61], and another has used Strand-seq[58] to generate maternal and paternal phased long-read-based human genome assemblies with fewer switch errors, including on HG002. As with trio binning, these approaches appear to work best when phasing is integrated with the assembly process, but further improvements are necessary to match or surpass the quality seen with a parental trio graph-based approach used here.

We used ONT-UL reads to fill in GA-rich repeats and other challenging sequence gaps between the HiFi-based contigs. A potential alternative is the PacBio CLR reads that do not make it to HiFi accuracy contain some of the GA-rich repeats, and could be used to fill in some of these gaps. The remaining few gaps per chromosome in the HG002 assemblies are mostly restricted to the hardest-to-assemble regions around segmental duplications, centromeres, telomeres, rDNA arrays and other complex repeats, many with differences between haplotypes. Direct integration of ONT-UL data within the assembly graph and manual curation were necessary for finishing these regions in the T2T-CHM13 assembly[5]. Thus, integration of both HiFi and ONT-UL data in a diploid assembly graph, combined with long-range phasing information from trios, Hi-C or Strand-seq may soon enable automated T2T diploid genome assemblies[62]. For each of these additional approaches, the amount of read coverage needs to be titrated. Furthermore, the ability to produce higher coverage cheaper and faster continues to improve for all technologies. For those who wish to contribute assemblies to the human pangenome references, we encourage them to utilize our recommended processes to obtain the highest-quality assemblies possible; we also encourage contribution of new methods to further improve the quality and completeness of human and other species genome assemblies. We believe that generating complete, haplotype phased and accurate genome assemblies will be critical for generating accurate pangenome graphs.

The new biological discoveries made here demonstrate that even with a single individual, additional genetic diversity contributing to the human population can be found. Using these methods for the generation of additional diploid human genomes and creation of a human reference pangenome should enable a more-complete picture of human genetic diversity, greater accuracy for precision medicine for haplotype-specific diseases and a greater understanding of the biology of genomes.

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

Erich D. Jarvis[1,2,44✉], Giulio Formenti[1,44✉], Arang Rhie[3], Andrea Guarracino[4], Chentao Yang[5], Jonathan Wood[6], Alan Tracey[6], Francoise Thibaud-Nissen[7], Mitchell R. Vollger[8], David Porubsky[8], Haoyu Cheng[9,10], Mobin Asri[11], Glennis A. Logsdon[8], Paolo Carnevali[12], Mark J. P. Chaisson[13], Chen-Shan Chin[14], Sarah Cody[15], Joanna Collins[6], Peter Ebert[16], Merly Escalona[17], Olivier Fedrigo[1], Robert S. Fulton[15], Lucinda L. Fulton[15], Shilpa Garg[18], Jennifer L. Gerton[19], Jay Ghurye[20], Anastasiya Granat[21], Richard E. Green[11], William Harvey[8], Patrick Hasenfeld[22], Alex Hastie[23], Marina Haukness[11], Erich B. Jaeger[21], Miten Jain[11], Melanie Kirsche[24], Mikhail Kolmogorov[25], Jan O. Korbel[22], Sergey Koren[3], Jonas Korlach[26], Joyce Lee[23], Daofeng Li[27,28], Tina Lindsay[15], Julian Lucas[11], Feng Luo[29], Tobias Marschall[16], Matthew W. Mitchell[30], Jennifer McDaniel[31], Fan Nie[32], Hugh E. Olsen[11], Nathan D. Olson[31], Trevor Pesout[11], Tamara Potapova[19], Daniela Puiu[33], Allison Regier[34], Jue Ruan[35], Steven L. Salzberg[33], Ashley D. Sanders[36], Michael C. Schatz[24], Anthony Schmitt[37], Valerie A. Schneider[7], Siddarth Selvaraj[37], Kishwar Shafin[11], Alaina Shumate[33], Nathan O. Stitziel[15,27,38], Catherine Stober[22], James Torrance[6], Justin Wagner[31], Jianxin Wang[32], Aaron Wenger[26], Chuanle Xiao[39], Aleksey V. Zimin[33], Guojie Zhang[40], Ting Wang[15,27,28], Heng Li[9], Erik Garrison[41], David Haussler[2,42], Ira Hall[43], Justin M. Zook[31], Evan E. Eichler[2,8], Adam M. Phillippy[3], Benedict Paten[11], Kerstin Howe[6✉], Karen H. Miga[11✉] & Human Pangenome Reference Consortium*

[1]Vertebrate Genome Laboratory, The Rockefeller University, New York, NY, USA. [2]Howard Hughes Medical Institute, Chevy Chase, MD, USA. [3]Genome Informatics Section, Computational and Statistical Genomics Branch, National Human Genome Research Institute, National Institutes of Health, Bethesda, MD, USA. [4]Genomics Research Centre, Human Technopole, Viale Rita Levi-Montalcini, Milan, Italy. [5]BGI-Shenzhen, Shenzhen, China. [6]Tree of Life, Wellcome Sanger Institute, Cambridge, UK. [7]National Center for Biotechnology Information, National Library of Medicine, National Institutes of Health, Bethesda, MD, USA. [8]Department of Genome Sciences, University of Washington School of Medicine, Seattle, WA, USA. [9]Department of Data Sciences, Dana-Farber Cancer Institute, Boston, MA, USA. [10]Department of Biomedical Informatics, Harvard Medical School, Boston, MA, USA. [11]UC Santa Cruz Genomics Institute, University of California, Santa Cruz, CA, USA. [12]Chan Zuckerberg Initiative, Redwood City, CA, USA. [13]Quantitative and Computational Biology, University of Southern California, Los Angeles, CA, USA. [14]Foundation for Biological Data Science, Belmont, CA, USA. [15]McDonnell Genome Institute, Washington University School of Medicine, St. Louis, MO, USA. [16]Institute for Medical Biometry and Bioinformatics, Medical Faculty, Heinrich Heine University, Düsseldorf, Germany. [17]Department of Biomolecular Engineering, University of California Santa Cruz, Santa Cruz, CA, USA. [18]Department of Biology, University of Copenhagen, Copenhagen, Denmark. [19]Stowers Institute for Medical Research, Kansas City, MO, USA. [20]Dovetail Genomics, Scotts Valley, CA, USA. [21]Illumina, Inc., San Diego, CA, USA. [22]European Molecular Biology Laboratory, Genome Biology Unit, Heidelberg, Germany. [23]Bionano Genomics, San Diego, CA, USA. [24]Department of Computer Science, Johns Hopkins University, Baltimore, MD, USA. [25]Department of Computer Science and Engineering, University of California San Diego, La Jolla, CA, USA. [26]Pacific Biosciences, Menlo Park, CA, USA. [27]Department of Genetics, Washington University School of Medicine, St. Louis, MO, USA. [28]The Edison Family Center for Genome Sciences and Systems Biology, Washington University School of Medicine, St. Louis, MO, USA. [29]School of Computing, Clemson University, Clemson, SC, USA. [30]Coriell Institute for Medical Research, Camden, NJ, USA. [31]Material Measurement Laboratory, National Institute of Standards and Technology, Gaithersburg, MD, USA. [32]Hunan Provincial Key Lab on Bioinformatics, School of Computer Science and Engineering, Central South University, Changsha, China. [33]Department of Biomedical Engineering, Johns Hopkins University, Baltimore, MD, USA. [34]DNAnexus, Mountain View, CA, USA. [35]Agricultural Genomics Institute, Chinese Academy of Agricultural Sciences, Shenzhen, China. [36]Berlin Institute for Medical Systems Biology, Max Delbrück Center for Molecular Medicine in the Helmholtz Association (MDC), Berlin, Germany. [37]Arima Genomics, San Diego, CA, USA. [38]Cardiovascular Division, John T. Milliken Department of Internal Medicine, Washington University School of Medicine, St. Louis, USA. [39]State Key Laboratory of Ophthalmology, Zhongshan Ophthalmic Center, Sun Yat-sen University, Guangzhou, China. [40]Center for Evolutionary & Organismal Biology, Zhejiang University School of Medicine, Hangzhou, China. [41]Department of Genetics, Genomics and Informatics, University of Tennessee Health Science Center, Memphis, TN, USA. [42]Department of Ecology and Evolutionary Biology, University of California Santa Cruz, Santa Cruz, CA, USA. [43]Yale School of Medicine, New Haven, CT, USA. [44]These authors contributed equally: Erich D. Jarvis, Giulio Formenti. *A list of authors and their affiliations appear at the end of the paper. ✉e-mail: ejarvis@rockefeller.edu; gformenti@rockefeller.edu; kj2@sanger.ac.uk; khmiga@ucsc.edu

**Human Pangenome Reference Consortium**

Erich D. Jarvis[1,2,44], Giulio Formenti[1,44], Arang Rhie[3], Andrea Guarracino[4], Chentao Yang[5], Jonathan Wood[6], Alan Tracey[6], Francoise Thibaud-Nissen[7], Mitchell R. Vollger[8], David Porubsky[8], Haoyu Cheng[9,10], Mobin Asri[11], Glennis A. Logsdon[8], Paolo Carnevali[12], Mark J. P. Chaisson[13], Chen-Shan Chin[14], Sarah Cody[15], Joanna Collins[6], Peter Ebert[16], Merly Escalona[17], Olivier Fedrigo[1], Robert S. Fulton[15], Lucinda L. Fulton[15], Shilpa Garg[18], Jennifer L. Gerton[19], Jay Ghurye[20], Anastasiya Granat[21], Richard E. Green[11], William Harvey[8], Patrick Hasenfeld[22], Alex Hastie[23], Marina Haukness[11], Erich B. Jaeger[21], Miten Jain[11], Melanie Kirsche[24], Mikhail Kolmogorov[25], Jan O. Korbel[22], Sergey Koren[3], Jonas Korlach[26], Joyce Lee[23], Daofeng Li[27,28], Tina Lindsay[15], Julian Lucas[11], Feng Luo[29], Tobias Marschall[16], Matthew W. Mitchell[30], Jennifer McDaniel[31], Fan Nie[32], Hugh E. Olsen[11], Nathan D. Olson[31], Trevor Pesout[11], Tamara Potapova[19], Daniela Puiu[33], Allison Regier[34], Jue Ruan[35], Steven L. Salzberg[33], Ashley D. Sanders[36], Michael C. Schatz[24], Anthony Schmitt[37], Valerie A. Schneider[7], Siddarth Selvaraj[37], Kishwar Shafin[11], Alaina Shumate[33], Nathan O. Stitziel[15,27,38], Catherine Stober[22], James Torrance[6], Justin Wagner[31], Jianxin Wang[32], Aaron Wenger[26], Chuanle Xiao[39], Aleksey V. Zimin[33], Guojie Zhang[40], Ting Wang[15,27,28], Heng Li[9], Erik Garrison[41], David Haussler[2,42], Ira Hall[43], Justin M. Zook[31], Evan E. Eichler[2,8], Adam M. Phillippy[3], Benedict Paten[11], Kerstin Howe[6] & Karen H. Miga[11]

## Methods

### Cell lines

The GM24385 (RRID:CVCL_1C78) EBV-immortalized LCL of HG002 was obtained from the National Institute for General Medical Sciences (NIGMS) Human Genetic Cell Repository at the Coriell Institute for Medical Research. This cell line was used to generate the Oxford Nanopore sequencing and Bionano mapping data. For the Illumina and Pacific Biosciences sequencing data, NIST Reference Material (RM) 8391 DNA was used, which was prepared from a large batch of GM24385 to control for differences arising during cell growth. For paternal (HG003) and maternal (HG004) samples, DNA was extracted from cell lines publicly available as GM24149 (RRID:CVCL_1C54) and GM24143 (RRID:CVCL_1C48), respectively, and Illumina sequencing of DNA from NIST RM 8392 (containing vials of HG002, HG003 and HG004) was used.

### Chromosome spreads and FISH

For chromosome spreads preparation, GM24385 LCL cells were arrested in mitosis by the addition of Karyomax colcemid solution (0.1 µg ml$^{-1}$; Life Technologies) to the growth medium for 6 h. Cells were collected by centrifugation at 200$g$ for 5 min and incubated in 0.4% KCl swelling solution for 10 min. Swollen cells were pre-fixed by addition of freshly prepared methanol: acetic acid (3:1) fixative solution (approximately 100 µl per 10 ml total volume). Pre-fixed cells were collected by centrifugation at 200$g$ for 5 min and fixed in methanol: acetic acid (3:1) fixative solution. Spreads were dropped on a glass slide and incubated at 65 °C overnight. Before hybridization, slides were treated with 1 mg ml$^{-1}$ RNAse A (1:100 from Qiagen) in 2× SSC for at least 45 min at 37 °C and then dehydrated in a 70%, 80% and 100% ethanol series for 2 min each. Denaturation of spreads was performed in 70% formamide/2× SSC solution at 72 °C for 1.5 min and immediately stopped by immersing slides in ethanol series pre-chilled to −20 °C. Fluorescently labelled DNA probes (DXZ1 for the X chromosome from Cytocell, and made in-house for the Y chromosome probe) were denatured separately in hybridization buffer (Empire Genomics) by heating to 80 °C for 10 min before applying to denatured slides. Spreads were hybridized to probes under HybriSlip hybridization cover (GRACE Biolabs) sealed with Cytobond (SciGene) in a humidified chamber at 37 °C for 72 h. After hybridization, slides were washed in 50% formamide/2× SSC three times for 5 min at 45 °C, then in 1× SSC solution at 45 °C for 5 min twice, and at room temperature once. Slides were then rinsed with double deionized H$_2$O, air dried and mounted in Vectashield containing DAPI (Vector Laboratories). Images were acquired on the LSM710 confocal microscope (Zeiss) using the ×63/1.40 NA oil objective or on the Nikon TiE microscope equipped with ×100 objective NA 1.45, Yokogawa CSU-W1 spinning disk and Flash 4.0 sCMOS camera. Image processing and chromosome counts were performed in FIJI.

### Genome sequencing

The sequence data used for this study (HG002 Data Freeze v1.0) are available on GitHub (https://github.com/human-pangenomics/HG002_Data_Freeze_v1.0). DNA samples were extracted from large homogenized growths of B lymphoblastoid cell lines of HG002, HG003 and HG004 from the Coriell Institute for Medical Research.

**Illumina reads. Paired-end reads.** Whole-genome data, TruSeq (LT) libraries, 300x PCR-free paired-end 150 bp + 40x, PCR-free paired-end 250 bp on Illumina HiSeq 2500, were from GIAB[25]. HG002 was sequenced to 51.7× coverage, HG003 to 69.1× and HG004 to 70.6×.

**Long-molecule linked reads.** For 10X Genomics reads, Chromium Genome Platform from 10X Genomics was sequenced to two depths: 51.7× coverage and a deeper 84.4× coverage (300 Gb) dataset.

Additional data are available from BioProject: PRJNA527321. For Transposase Enzyme Linked Long-read Sequencing (TELL-seq) linked reads, these reads were made available from another study[63].

**PacBio reads.** DNA was sheared to approximately 20 kb with a Megaruptor 3, libraries were prepared with SMRTbell Express Template Prep Kit 2.0 and size selected with SageELF to the targeted size (15 kb, 19 kb, 20 kb or 25 kb), and sequenced on the Sequel II System with Chemistry 2.0 (15 kb or 20 kb libraries; 36× and 16× coverage, respectively), pre-2.0 Early Access Chemistry (15 kb, 19 kb and 25 kb libraries; 24×, 14× and 11× coverage, respectively) and Sequel System with Chemistry 3.0 (15 kb libraries; 28× coverage). For PacBio CLRs, libraries were prepared with SMRTbell Express Template Prep Kit 2.0, size selected to a target size (more than 30 kb), and sequenced on a Sequel II System with Chemistry 1.0 and Chemistry 2.0 to more than 60-fold coverage from two SMRT cells.

**ONT reads.** All of the ONT sequencing for HG0002 was run on PromethION and GridION sequencing instruments. The GridION uses MinION flow cells and the PromethION uses PromethION flow cells. Both flow cells used the same ONT R9.4.1 sequencing chemistry. Sequencing libraries were prepared for PromethION sequencing, with the unsheared sequencing library prep protocol. We used 28 PromethION flow cells to generate a total of 658× coverage (assuming 3.1-Gb genome size) and approximately 51× coverage with 100 kb+ reads, although we never used all 658× for any one assembly. GridION sequencing prepared libraries with the ultralong sequencing library prep protocol and used 106 MinION flow cells to generate a total of approximately 52× coverage (assuming 3.1-Gb genome size) and approximately 15× coverage of 100 kb+ reads[64]. More recently, we obtained 10×+ of more than 100 kb per ultralong PromethION flow cell.

**Hi-C linked reads.** Two different Hi-C datasets were made with two distinct protocols, to reach as uniform coverage across the genome as possible: Dovetail Omni-C (named Hi-C1) and Arima Genomics High Coverage Hi-C (named Hi-C2) protocols. For Hi-C1, about 100,000 cultured HG002 cells were processed for proximity ligation libraries without restriction enzymes. High-coverage (69×) sequencing was done on a Nova-seq (250 bp PE). For Hi-C2, two libraries were prepared from two cell culture replicates and sequenced with 2× 150 bp and 2× 250 bp Illumina reactions each. The combination of restriction enzymes represent ten possible cut sites: ^GATC, G^ANTC, C^TNAG and T^TAA; '^' is the cut site on the plus DNA strand, and the 'N' can be any of the four genomic bases.

**Strand-seq.** Strand-specific libraries were generated as previously described[10], from 192 barcoded single-cells and sequenced on a NextSeq Illumina platform. The 192 barcoded single-cell libraries were pooled for sequencing of the HG002 sample. Raw demultiplexed fastq files from the paired-end sequencing run (80-bp read lengths) were uploaded for each single-cell library. These data can be found at https://s3-us-west-2.amazonaws.com/human-pangenomics/index.html?prefix=HG002/hpp_HG002_NA24385_son_v1/Strand_seq/.

**Optical maps.** Bionano DLE1 data were collected with throughput of 1,303 Gb (molecules of more than 150 kb) and Read N50 of 293 kb (molecules of more than 150 kb) provided by Bionano Genomics and the GIAB Consortium.

### Genome assembly pipelines tested

The assembly bakeoff was an open public science experiment and evaluation, in which researchers of the HRPC and anyone in the scientific community could contribute, with the goal of creating the highest-quality de novo assembly possible, of one or both haplotypes, using an automated process. We did this by contacting known assembly

experts, sending out announcements on consortium email list (for example, HPRC, VGP, T2T, HGSVC and GIAB), and announcements on HRPC-associated websites (https://humanpangenome.org/hg002/; https://github.com/human-pangenomics/HG002_Data_Freeze_v1.0). We grouped the assembly pipelines tested into five categories according to whether contigs only or contigs and scaffolds were generated, and whether the contigs and/or scaffolds were haplotype phased or merged as a pseudohaplotype (Table 1). The assemblies were further classified by whether parental trio data were used and whether they were reference based or de novo (Table 1). The assemblies were assigned ID numbers on the basis of the order received by the consortium evaluation team, and in no part reflect order of assembly metric quality. All but two assemblies (asm3 and asm23) that used PacBio data used the recommended downsampled HiFi SMRT cell runs from the 15-kb and 20-kb insert libraries totalling approximately 34× coverage (https://github.com/human-pangenomics/HG002_Data_Freeze_v1.0#hg002-data-freeze-v10-recommended-downsampled-data-mix). Asm3 used the 19-kb, 20-kb and 25-kb insert libraries. Asm23 used PacBio CLRs. The specific method details for each assembly pipeline, under each of the five major categories, are described below.

### Diploid scaffold assemblies. Trio binning phasing VGP pipeline 1.6 (asm23).
This assembly was based on a modified version of the VGP trio pipeline 1.6 (ref. [11]). All data types (PacBio CLRs, 10XG linked-reads, Bionano maps and Hi-C2 reads) were haplotype binned or filtered by haplotype. In brief, CLRs were binned (hapUnknownFraction = 0.01) and assembled into contigs using HiCanu[17] v1.8. NA24143 (maternal HG004) and NA24149 (paternal HG003) 250-bp PE Illumina reads were used for binning. CLR coverage of the child (HG002) was 74× and 72× for the maternal and paternal haplotypes, respectively. To polish the contigs, the binned CLRs were used for each respective haplotype with Arrow (variantCaller v2.3.3). The two haplotype contigs were then purged from each other using purge_dups v1.0 (ref. [65]), conducted in the haploid mode (calcuts -d1) and only JUNK and OVLP were removed. To these contigs, Bionano molecules were aligned and assigned to the haplotype bin with higher alignment confidence. Bionano molecules aligning equally well to both parental haplotype contigs (alignment score discrepancies of less than equal to $10^{-2}$) were randomly split into two clusters equally and assigned to the bins. The same method of splitting the molecules was used for molecules aligning to neither of the parental assemblies (https://github.com/andypang0714/Bionano_Trio_binning). Binned Bionano molecules were then assembled to haploid assemblies. Cross-checking was then performed by aligning the paternal and maternal Bionano assemblies to the parental assemblies to identify regions where both parents shared the same allele, and the best allele was picked for the next round of trio binning and assemblies. 10XG and Hi-C reads were filtered for *k*-mers of the alternate haplotype using Meryl (https://github.com/marbl/meryl/tree/master/src/meryl), and a custom script that is part of the VGP trio pipeline 1.6 was used to exclude read pairs containing *k*-mers only found in the other haplotype. With this prepared data, three rounds of scaffolding were then conducted on each haplotype, sequentially with the binned 10XG reads using Scaff10x v4.2, binned Bionano maps with Solve v3.4 and binned Hi-C linked reads with Salsa v2.2. The assemblies were not further polished as they already reached Q40 as judged by Merqury. Compute time was not tracked. The source code is available (https://github.com/VGP/vgp-assembly/tree/master/pipeline).

### DipAsm contig and scaffolding pipeline (asm10).
This assembly is based on a protocol similar to DipAsm reported in ref. [29]. PacBio HiFi reads were first assembled into unphased contigs using Peregrine. Contigs were grouped and ordered into scaffolds with Hi-C2 data. The HiFi reads were then mapped to scaffolds using minimap2 and heterozygous SNPs called using DeepVariant[66]. The heterozygous SNP calls were phased with both HiFi and Hi-C2 data using HapCUT2 (ref. [67]) and Whatshap[68]. The reads were then partitioned on the basis of their

phase using a custom script. The partitioned reads were re-assembled into phased contigs using Peregrine. The contigs were then ordered and joined together with 100 Ns to produce phased scaffolds. Compute time was not tracked. The source code is available at https://github.com/shilpagarg/DipAsm.

### Dovetail DipAsm variant pipeline (asm2 and asm22).
The Dovetail pipeline used is a variation of the DipAsm pipeline previously described[29]. The main difference is that DipAsm used HiFi reads for SNP calling with DeepVariant and the Dovetail protocol used Omni-C reads (Hi-C1) for SNP calling with FreeBayes. In particular, PacBio HiFi reads were assembled into contigs using the Peregrine assembler with default parameters. These contigs were joined into chromosome-scale scaffolds using Dovetail Omni-C data and either HiRise (Dovetail Genomics; asm2) or Salsa2 (ref. [18]) (asm22) scaffolders. Omni-C reads were then aligned to scaffolds and haplotype SNPs were called using FreeBayes. These SNPs were then phased with HapCUT2 and Omni-C long-range links to obtain chromosome-scale phased blocks. These phased SNPs were used to partition HiFi and Omni-C reads into two haplotypes. Reads for which the partitioning could not be done ambiguously were assigned to both haplotypes. Phased HiFi reads for each haplotype were assembled again with Peregrine and scaffolded with haplotype-specific Omni-C reads to obtain chromosome-scale phased scaffolds. Compute time was not tracked. All of the tools were run on AWS EC2 with c5d.9xlarge instance type. The source code for HiRise is proprietary. The source code for Salsa2 is available (https://github.com/marbl/SALSA/commit/974589f3302b773dcf0f20c3332fe9daf009fb93).

### PGAS pipeline (asm14).
The recent PGAS diploid genome assembly pipeline has been previously described[58]. First, a non-haplotype resolved ('squashed') contig assembly was generated from PacBio HiFi reads using Peregrine v0.1.5.5 (github.com/cschin/Peregrine). Illumina short reads from the Strand-seq data[69] were aligned against this squashed assembly to identify contigs that most likely originate from the same chromosome based on similar Watson–Crick strand inheritance patterns[70]. This information was then used to cluster the contigs into roughly chromosome-scale clusters, which helps to avoid chimeric chromosome assemblies, allows for parallelization of the assembly pipeline and facilitates phasing. Next, heterozygous SNVs were identified based on long-read alignments against the clustered assembly with DeepVariant v0.9.0. To obtain chromosome-scale haplotypes, integrative phasing with WhatsHap[68] was performed, combining local dense phase information derived from long reads with global sparse phase information inferred from Strand-seq alignments. Next, phased heterozygous SNVs were used to assign each HiFi read to its corresponding haplotype ('haplo-tagging') or remain in the fraction of haplotype-unassigned reads. The haplotags were used to split the HiFi reads into two haploid read sets, which, together with the haplotype-unassigned reads, were the input to assemble two haplotype contig sequences per chromosome-scale cluster with Peregrine v0.1.5.5. After polishing the contigs for two rounds with Racon v1.4.10 (ref. [71]) and the haploid long-read datasets, the per chromosome cluster assemblies were merged to create a genome-scale diploid assembly. The final round of scaffolding of each haplotype was performed with the short reads from the Strand-seq data, on HiFi contigs with a minimum size of 500 kb. This size thresholding was necessary as the contig order can only be inferred from strand-state changes resulting from sister chromatid exchanges (SCEs; a process during DNA replication in which two sister chromatids break, rejoin and physically exchange regions of the parental strands). SCEs are low-frequency events that are thus less likely to produce a traceable signal with decreasing contig size. The complete assembly pipeline run required less than 2,000 CPU hours on a three-node cluster (3 × 36C, 1.4 TB of RAM) with a peak RAM usage of around 600 GB (squashed Peregrine assembly). The source code is available at https://github.com/ptrebert/project-diploid-assembly; pipeline parameter version 8.

**CrossStitch (asm17).** The assembly was produced using CrossStitch, a reference-based pipeline for diploid genome assembly. SNPs and small indels were called with respect to GRCh38 for HG002 from alignments of unbinned 30× PacBio HiFi reads. Variant calling was performed on this BAM using DeepVariant v0.9 (ref. [66]) and the PacBio model. A full set of commands and parameters are available on the PacBio case study: https://github.com/google/deepvariant/blob/r0.9/docs/deepvariant-pacbio-model-case-study.md. Larger SVs were called by running Sniffles v1.0.11 (with parameters -s 10 -l 10 --min_homo_af 0.7) on minimap2 v2.17 alignments of the HiFi reads and refining these calls with Iris v1.0.1 (https://github.com/mkirsche/Iris)[72]. Then, the SNVs and small indel variants (less than 30 bp) called from DeepVariant were phased using HapCUT2 v1.1 on the ONT + Hi-C alignments, and these phase blocks were used to assign a haplotype to each HiFi read. SV phasing was performed by observing the reads supporting each heterozygous SV call and assigning the variant to the haplotype that the majority of the reads came from. Finally, vcf2diploid (https://github.com/abyzovlab/vcf2diploid) from the AlleleSeq algorithm[73] was used to incorporate small variant and SV calls into a template consisting of the GRCh38 reference genome sequence, producing the final assembly for HG002. The end-to-end assembly took less than 2 days on a high-memory machine at JHU using at most 40 cores at a time. Peak RAM utilization was less than 100 GB. The source code is available at https://github.com/schatzlab/crossstitch (commit ID: e49527b).

**Diploid contig assemblies. Trio binning Flye ONT pipeline (asm6 and asm7).** Following a trio-based assembly approach[22], parental Illumina 21-mers were counted in the child, maternal and paternal read sets (full sets, not subset coverage recommendations). Haplotype-specific mers were created using Merqury v1.0 (ref. [36]) and Meryl v1.0 (https://github.com/marbl/meryl) with the command: hapmers.sh.sh mat.k21.meryl pat.k21.meryl child.k21.meryl. These short reads were then used to bin ONT standard long (asm6) or ultralong more than 100-kb (asm7) reads into their maternal-specific and paternal-specific haplotypes. The ONT recommended subset reads were then assigned using splitHaplotigs in Canu v2.0 (ref. [17]) with the command: splitHaplotype -cl 1000 -memory 32 -threads 28 -R HG002_ucsc_ONT_lt100kb.fastq.gz \ -R HG002_giab_ULfastqs_guppy3.2.4.fastq.gz \ -H ./0-kmers/haplotype-DAD.meryl 6 ./haplotype-DAD.fasta.gz \ -H ./0-kmers/haplotype-MOM.meryl 6 ./haplotype-MOM.fasta.gz \ -A ./haplotype-unknown.fasta.gz.

Flye v2.7-b1585 (ref. [30]) was then run on the binned reads to generate maternal and paternal contigs with the command: fly --threads 128 --min-overlap 10000 --asm-coverage 40 -out_dir <MOM/DAD> --genome-size 3.1g --nano-raw haplotype-<DAD/MOM>.fasta.gz. Flye sometimes inserts gaps when it is not certain of a repeat sequence, and thus some contigs appear as scaffolds. However, the assembly is still contig level. No base-level polishing (with short or long reads) was conducted on the assembly. The ONT standard Flye runs took approximately 1,200 CPU hours (20 wall clock hours) and 500 GB of memory. The ONT-UL assemblies took approximately 3,000 CPU hours (60 wall clock hours) and 800 GB of memory. The source codes for Canu, Mercury and Flye are available (https://github.com/marbl/canu, https://github.com/marbl/merqury and https://github.com/fenderglass/Flye).

**Trio binning HiCanu contig pipeline (asm19).** Following a trio-based assembly approach[22], parental Illumina 21-mers were counted in the child, maternal and paternal read sets (full sets, not subset coverage recommendations). Haplotype-specific mers were created using Merqury v1.0 (ref. [36]) and Meryl v1.0 (https://github.com/marbl/meryl) with the command: hapmers.sh mat.k21.meryl pat.k21.meryl child.k21.meryl. The HiFi-recommended 34× subset reads were then assigned to using splitHaplotigs in Canu v2.0 (ref. [17]) with the command: splitHaplotype -cl 1000 -memory 32 -threads 28 -R m64012_190920_173625.fastq.gz -R m64012_190921_234837.fastq.gz -R m64011_190830_220126.Q20.fastq.gz -R m64011_190901_095311.Q20.fastq.gz -H ./0-kmers/haplotype-DAD.meryl 6 ./haplotype-DAD.

fasta.gz -H ./0-kmers/haplotype-MOM.meryl 6 ./haplotype-MOM.fasta.gz -A ./haplotype-unknown.fasta.gz. Any reads that were unclassified were randomly divided into two bins. The resulting maternal and paternal read sets were independently assembled with HiCanu[17] v2.0 with the commands: canu -p 'asm' 'gridOptions=--time=4:00:00 --partition=quick,norm' 'gridOptionsCns=--time=30:00:00 --partition=norm' 'genomeSize=3.1g' 'gfaThreads=48' 'batOptions=-eg 0.01 -sb 0.01 -dg 6 -db 6 -dr 1 -ca 50 -cp 5' -pacbio-hifi haplotype-[DAD|MOM].fasta.gz haplotype-unknown-batch[1|2].fastq.gz. The source codes are available at https://github.com/marbl/canu and https://github.com/marbl/merqury. Publication is available[17].

All runs used the 'quick' partition of the NIH Biowulf cluster (https://hpc.nih.gov). HiCanu required approximately 1,400 CPU hours per haplotype (19 wall clock hours) and no single job required more than 64 GB of memory.

**Trio binning Peregrine contig pipeline (asm20).** The same binned reads as for asm19 were used for this assembly. The reads were assembled with Peregrine v0.1.5.3+0.gd1eeebc.dirty with the command yes yes | python3 Peregrine/bin/pg_run.py asm \ input.list 24 24 24 24 24 24 24 24 24 \ --with-consensus --shimmer-r 3 --best_n_ovlp 8 \ --output ./. The input.list specifies the appropriate haplotype input reads. Compute time was not tracked. The source codes can be found at https://github.com/cschin/Peregrine and https://github.com/marbl/merqury.

**Trio phasing hifiasm contig pipeline (asm9).** Hifiasm finds alignments between HiFi reads and corrects sequencing errors observed in alignments[31]. It labels a corrected read with its inferred parental origin using parent-specific 31-mers counted from parental short reads. HiFi reads in long homozygous regions do not have parent-specific 31-mers and are thus unlabelled. Hifiasm then builds a string graph from read overlaps that carries read labels. It traces paternal and maternal reads in the graph to generate paternal and maternal contigs, respectively. We collected paternal 31-mers from short reads with 'yak count -b37 -o sr-pat.yak sr-pat.fq.gz' (and similarly for maternal) and assembled HiFi reads with 'hifiasm −1 sr-pat.yak −2 sr-mat.yak hifi-reads.fq.gz'. The assembly took 305 CPU hours. The source code is available (https://github.com/chhylp123/hifiasm/releases/tag/v0.3).

**DipAsm contig pipeline (asm11).** The assembly pipeline mimics the DipAsm steps explained for asm10. The pipeline takes as input HiFi and Hi-C datasets and outputs the phased contigs. Initially, the pipeline produces unphased contigs using Peregrine and then these unphased contigs are scaffolded to produce chromosome-scale sequences using HiRise. Afterwards, the heterozygous SNPs are called and are phased using HiFi and Hi-C data. These phased SNPs are informative sites to partition HiFi reads to haplotypes on the chromosome level. The phased reads are then assembled using Peregrine to produce phased contigs.

**Peregrine contig pipeline (asm3 and asm4).** The Peregrine assembler[34] was used to generate contigs on the HiFi reads, using either the full coverage sequence (asm3) consisting of 19-kb, 20-kb and longer 25-kb read libraries or downsampled to 34× and shorter 15-kb reads (asm4). A module was written to separate likely true-variant sites from differences between reads caused by sequencing errors. This was done by using the overlap data from the Peregrine assembler overlapping modules with additional alignment analysis. The variants of the read overlapped data were analysed to get a subset of variants that should belong to the same haplotypes. The reads with the same set of variants were considered to be haplotype consistent, and the overlaps between those haplotype-consistent reads were considered for constructing the contig assembly. Overlaps between different haplotypes or different repeats from the analysis results were ignored. It is expected that the generated contigs are from single haplotypes in those regions, which have enough heterozygous variants. Compute time was not tracked. The source code is available (https://github.com/cschin/Peregrine_dev/commit/93d416707edf257c4bcb29b9693c3fda25d97a29). The most up-to-date Peregrine code can be found at https://github.com/cschin/Peregrine-2021.

**FALCON-Unzip contig pipeline (asm16).** FALCON-Unzip[21] version 2.2.4-py37hed50d52_0 was run on reads from four SMRT cells from two HiFi libraries (15 kb and 20 kb, 34× coverage total reads) with 'input_type = preads, length_cutoff_pr = 8000, ovlp_daligner_option = -k24 -h1024 -e.98 -l1500 -s100, ovlp_HPCdaligner_option = -v -B128 -M24, ovlp_DBsplit_option = -s400, overlap_filtering_setting = --max-diff 200 --max-cov 200 --min-cov 2 --n-core 24 --min-idt 98 --ignore-indels' for the initial contig assembly and default parameters for unzipping haplotypes. The assembly took 2,540 CPU-core hours on nodes with Intel Xeon processor E5-2600 v4. The source code is available (https://anaconda.org/bioconda/pb-falcon/2.2.4/download/linux-64/pb-falcon-2.2.4-py37hed50d52_0.tar.bz2).

**HiCanu purge dups contig phasing pipeline (asm8).** HiCanu[17] v2.0 was used with the command canu -p 'asm' 'gridOptions=--time=4:00:00 --partition=quick,norm' 'gridOptionsCns=--time=30:00:00 --partition=norm ' 'genomeSize=3.1g' 'gfaThreads=48' -pacbio-hifi m64012_190920_173625.fastq.gz m64012_190921_234837.fastq.gz m64011_190830_220126.Q20.fastq.gz m64011_190901_095311.Q20.fastq.gz.

Purge_dups[65] was used to remove alternate haplotypes (GitHub commit ID: b5ce3276773608c7fb4978a24ab29fdd0d65f1b5), with the thresholds of 5 7 11 30 22 42. Purge_dups introduces gaps near the purged sequenced regions of the contigs, and thus some contigs appear as scaffolds. However, the assembly is still contig level. HiCanu required approximately 1,800 CPU hours and no single job required more than 64 GB of memory (22 wall clock hours). Purge_dups required 40 CPU hours and less than 1 GB of memory.

**Haploid scaffold assemblies. Flye ONT pipeline (asm5).** Flye v2.7b-b1579 (ref. [30]) was used to assemble (downsampled) ONT reads into contigs, using the default parameters with extra '--asm-coverage 50 --min-overlap 10000' options. Two iterations of the Flye polishing module were applied using ONT reads, followed by two polishing iterations using HiFi reads. Finally, Flye graph-based scaffolding module was run on the polished contigs, which generated 54 scaffold connections and slightly improved the assembly contiguity. Assembly took approximately 5,000 CPU hours and polishing (ONT + HiFi) took approximately 3,000 CPU hours. Peak RAM usage was approximately 900 GB. The pipeline was run on a single computational node with two Intel Xeon 8164 CPUs, with 26 cores each and 1.5 TB of RAM. The source code can be found at https://github.com/fenderglass/Flye/ (commit ID: ec206f8).

**Shasta ONT + HiC (asm18 and asm21).** The Shasta assembler[28] was used to assemble ONT reads into contigs. The contigs were polished using PEPPER (https://github.com/kishwarshafin/pepper), which also uses only the ONT reads. The contigs were scaffolded with Omni Hi-C (Hi-C1) using HiRise (asm18) or Salsa v2.0 (asm21). Compute time was not tracked. The source code is available (https://github.com/chanzuckerberg/shasta).

**Flye and MaSuRCA (asm15).** A subset of downsampled ONT-UL data that contained approximately 38× genome coverage of 120-kb reads or longer was used. The ONT reads were assembled into contigs using the Flye assembler[30] v2.5. The contigs were polished with downsampled 30× coverage of PacBio HiFi 15-kb and 20-kb reads, using POLCA, a tool distributed with the MaSuRCA[32]. To scaffold and assign the assembled contigs to chromosomes, a reference-based scaffolding method was used embodied in the chromosome_scaffolder script included in MaSuRCA. GRCh38.p12 was used as a reference (without the ALT scaffolds) for scaffolding, with the chromosome_scaffolder option enabled, which allows it to fill in the gaps in scaffolds, where possible, with GRCh38 sequence, in lowercase letters. The final assembly was named JHU_HG002_v0.1. Compute time was not tracked. The source code can be found at https://github.com/alekseyzimin/masurca.

**MaSuRCA (asm1).** MaSuRCA v3.3.1 (ref. [32]) with default parameters was run on the combined Illumina, ONT and PacBio HiFi data to obtain a set of contigs designated the Ash1 v0.5 assembly. The ONT and PacBio data were an earlier release, from 2018, and the read lengths were shorter than the later release used by most other methods in this evaluation. After initial scaffolding, MaSuRCA was used to remove redundant haplotype-variant scaffolds by aligning the assembly to itself and looking for scaffolds that were completely covered by another larger scaffold and that were more than 97% identical to the larger scaffold. To scaffold and assign the assembled contigs to chromosomes, we used a reference-based scaffolding method embodied in the chromosome_scaffolder script included in MaSuRCA. We used the GRCh38.p12 as the reference (without the ALT scaffolds), and we enabled an option in chromosome_scaffolder that allows it to fill in gaps with the GRCh38 sequence, using lowercase letters. Finally, we examined SNVs reported at high frequency in an Ashkenazi population from the Genome Aggregation Database (gnomAD). GnomAD v3.0 contains SNV calls from short-read whole-genome data from 1,662 Ashkenazi individuals. At 273,866 heterozygous SNV sites where HG002 contained the Ashkenazi major allele and where our assembly used a minor allele, we replaced the allele in Ash1 with the Ashkenazi major allele. A publication of the final curated asm1 assembly has been made[60]. Compute time was not tracked. The source code is available (https://github.com/alekseyzimin/masurca).

**Haploid contig assemblies. wtdbg2 (asm13).** The standard wtdbg2 assembly pipeline[35] was applied on HiFi reads. Parameters '-k 23 -p 0 -S 0.8 --no-read-clip --aln-dovetail -1' were customized to improve the contiguity. The source code is available (https://github.com/ruanjue/wtdbg2; commit ID: d6667e78bbde00232ff25d3b6f16964cc7639378). Commands and parameters used were: '#!/bin/bash'; 'wtdbg2 -k 23 -p 0 -AS 4 -s 0.8 -g 3g -t 96 --no-read-clip --aln-dovetail -1 -fo dbg -i ../rawdata/SRR10382244.'; 'fasta -i ../rawdata/SRR10382245.fasta -i ../rawdata/SRR10382248.fasta -i ../rawdata/SRR10382249.fasta'; 'wtpoa-cns -t 96 -i dbg.ctg.lay.gz -fo dbg.raw.fa'; 'minimap2 -I64G -ax asm20 -t96 -r2k dbg.raw.fa ../rawdata/SRR10382244.fasta ../rawdata/SRR10382245.fasta'; '../rawdata/SRR10382248.fasta ../rawdata/SRR10382249.fasta | samtools sort -m 2g -@96 -o dbg.bam'; 'samtools view -F0x900 dbg.bam | wtpoa-cns -t 96 -d dbg.raw.fa -i - -fo dbg.cns.fa'; 'ref.[35]'; 'compute time'; 'wtdbg2: real 20,349.731 s, user 1,178,897.390 s, sys 18,351.090 s, maxrss 194,403,704.0'; 'kB, maxvsize 209,814,736.0 kB'; 'wtpoa-cns(1): real 3,350.517 s, user 260,551.730 s, sys 1,040.200 s, maxrss 9,978,492.0'; 'kB, maxvsize 15,839,032.0 kB'; 'wtpoa-cns(2):real 2,181.084 s, user 149,528.810 s, sys 815.380 s, maxrss 11,134,244.0 kB'; 'maxvsize 16,012,012.0 kB'; 'others: unknown'.

**NECAT Feng Luo group (asm12).** We used the NECAT assembler[33] to assemble ONT reads of HG002, which contained about 53× coverage excluding ONT-UL reads. The command 'necat.pl config cfg' was first used to generate the parameter file 'cfg'. The default values in 'cfg' were replaced with the following parameters: 'GENOME_SIZE = 3000000000, THREADS = 64, PREP_OUTPUT_COVERAGE=40, OVLP_FAST_OPTIONS=-n 500 -z 20 -b 2000 -e 0.5 -j 0 -u 1 -a 1000', 'OVLP_SENSITIVE_OPTIONS=-n 500 -z 10 -e 0.5 -j 0 -u 1 -a 1000, CNS_FAST_OPTIONS=-a 2000 -x 4 -y 12 -l 1000 -e 0.5 -p 0.8 -u 0, CNS_SENSITIVE_OPTIONS=-a 2000 -x 4 -y 12 -l 1000 -e 0.5 -p 0.8 -u 0, TRIM_OVLP_OPTIONS=-n 100 -z 10 -b 2000 -e 0.5 -j 1 -u 1 -a 400, ASM_OVLP_OPTIONS=-n 100 -z 10 -b 2000 -e 0.5 -j 1 -u 0 -a 400, CNS_OUTPUT_COVERAGE=40'. The command 'necat.pl bridge cfg' was run to generate the final contigs. It took approximately 12,555 CPU hours (error correction 2,500 h, assembling 8,123 h, bridging 1,216 h, polishing 716 h) on a 4-core 24-thread Intel(R) Xeon(R) 2.4 GHz CPU (CPU E7-8894[v4]) machine with 3 TB of RAM. The source code can be found at https://github.com/xiaochuanle/NECAT (commit ID: 47c6c23).

### HPRC-HG002 references

HiFi reads with adaptors were removed with HiFiAdapterFilt (https://github.com/sheinasim/HiFiAdapterFilt). After removing reads with

adaptors and other problems we went from 133× to 130× (using a genome size of 3.1 Gb). Maternal and paternal contigs were generated from 130× coverage of the remaining HiFi reads using hifiasm v0.14.1 in trio mode. Any remaining adapter sequences were hard masked in the assemblies and any contigs that were identified as contamination were removed. Both maternal and paternal assemblies were screened for mitochondrial sequences using BLAST against the reference sequence NC_012920.1, and the results were filtered with a modified version of MitoHiFi (https://github.com/marcelauliano/MitoHiFi). Mitochondrial contigs were removed from the assemblies and mapped against the reference sequence with Minimap2 (with -cx asm5 --cs). The contig with the highest alignment score (AS field: DP alignment score) were pulled, and if there were multiple, the contig with the lowest number of mismatches (NM field) was chosen. The selected contig was then rotated to match the reference sequence and appended to the maternal assembly.

The remaining maternal and paternal HiFi contigs were then separately scaffolded with paternal and maternal Trio-Bionano Solve v1.6 maps (205× and 195×), respectively. Conflicts between the Bionano maps and PacBio HiFi-based contigs were manually reviewed by three experts and decisions were made to accept or reject the cuts proposed by Bionano Solve. Further scaffolding of the paternal and maternal scaffolds was done with Salsa v2.3 and approximately 40× OmniC Hi-C reads excluding reads from the other haplotype with Meryl; that is, paired reads with $k$-mers only seen in one parent were removed before mapping and scaffolding. The resulting scaffolds were manually curated to ensure the proper order and orientation of contigs within the scaffolds, leading to additional joins and breaks. In parallel, approximately 78× ONT-UL reads were haplotype-binned with Canu v2.1 using Illumina short reads of the parents and then were assembled into their respective maternal and paternal contigs using Flye v2.8.3-b1695 with --min-overlap 10000. Bases were recalled with Guppy v4.2.2 before the assembly. The contigs were polished calling variants with Medaka (https://github.com/nanoporetech/medaka). The variants were filtered with Merfin using $k$-mers derived from Illumina short reads. Bcftools was used to apply the variants. The resulting ONT-UL assembly was used to patch gaps of the scaffolded HiFi-based assembly, using custom scripts (https://github.com/gf777/misc/tree/master/HPRC%20HG002/for_filling). Finally, a decontamination and an additional round of manual curation were conducted[40].

**Trio binning with optical mapping.** Using the Bionano direct label and stain chemistry and the Saphyr machine, high coverage of Bionano optical maps were generated of the HG002 (son), HG003 (father) and HG004 (mother) trio of samples and each assembled into diploid assemblies (Supplementary Table 13) with Bionano Solve 3.6. To separate the paternal and maternal alleles in the child assembly, the child molecules were aligned to the father and mother assemblies and assigned to the bin with higher alignment confidence. Molecules that aligned equally well (alignment score difference ⩽ $10^{-2}$) to the parents were split into two clusters equally and assigned to the bins. Similarly, molecules that aligned to neither of the parents were split into two clusters equally and assigned to the bins. As this method utilizes the unique SV sites in the diploid assemblies of parents to bin the molecules, it does not distinguish molecules for regions where the parents have the same SVs. Without special adjustment for the sex chromosome, this method does not eliminate the assembly of the X chromosome in the paternal assembly but the contigs are much shorter due to the missing proband molecules of regions where the father has unique SVs.

To further improve the separation of the parental alleles, a cross-checking step is performed. The binned paternal and maternal assemblies are aligned to both the father and mother assemblies (cross_check_alignment.py with RefAligner 11741 and optArguments_customized_for_diploid_reference.xml). Using these alignments, regions where the parents share an allele but are homozygous in one

and heterozygous in the other are identified. For example, in regions where the father has allele AA and the mother has allele AB, the B allele of the proband would be from the maternal and the A allele would be from the paternal side. Unless there are nearby SVs, molecules with allele A in the child can align equally to both parents, where the maternal assembly will then also include allele A, but with cross-checking, the correct allele (allele B) is identified and the wrong allele (allele A) gets eliminated through breaking it in the maternal assembly. A total of 54 regions of such characteristics were identified and broken in the paternal assembly, and 45 regions were identified and broken in the maternal assembly (haplotype_segregation_cross_check_rscript.R, haplotype_segregation_cut_step.py). Breaking at these regions allowed further separation of alleles in the next round of trio binning, using the binned and cross-checked assemblies as anchors. For the trio binning after cross-checking, 586 Gb of the probe molecules were binned to the paternal haplotype and 615 Gb binned to the maternal haplotype. These binned molecules were then assembled into the paternal haploid assembly (2.98 Gb with N50 of 79.22 Mb) and the maternal haploid assembly (2.96 Gb with N50 of 66.60 Mb) using Bionano Solve 3.6.

### Evaluation methods
For evaluation, we considered the following overarching framework. The 'assumed truth' is not one given a 'true assembly', but rather the consensus of multiple types of evidence. This evidence includes reference-free consistency between all raw data types (for example, HiFi, ONT, Illumina and Bionano) and the assembly, orthogonal data (for example, Strand-seq), and relative to complete and accurate T2T-CHM13 assembly of another individual, although haploid. We used the Mercury analysis tool kit for many metrics, which uses a $k$-mer approach on the raw sequence reads and/or the genome assembly to estimate QV, level of false duplication, degree of haplotype separation and assembly completeness[36]. Here we automated the Mercury took kit, for more rapid analyses of assemblies (https://dockstore.org/workflows/github.com/human-pangenomics/hpp_production_workflows/Merqury:master?tab=info).

**Contamination and manual curation.** Curation was conducted as described in the VGP[11]. In brief, for contamination identification, a succession of searches was used to identify potential contaminants in the generated assemblies. This included: (1) a megaBLAST98 search against a database of common contaminants (ftp://ftp.ncbi.nlm.nih.gov/pub/kitts/contam_in_euks.fa.gz); and (2) a vecscreen (https://www.ncbi.nlm.nih.gov/tools/vecscreen/) search against a database of adaptor sequences (ftp://ftp.ncbi.nlm.nih.gov/pub/kitts/adaptors_for_screening_euks.fa). The mitochondrial genome was identified by a megaBLAST search against a database of known organelle genomes (ftp://ftp.ncbi.nlm.nih.gov/blast/db/FASTA/mito.nt.gz). Organelle matches embedded in nuclear sequences were found to be NuMTs. On the basis of lessons learned in this study, we created an automated contamination removal pipeline (https://github.com/human-pangenomics/hpp_production_workflows/blob/master/QC/wdl/tasks/contamination.wdl).

For structural error identification, for each assembly, all sequence data (CLR, HiFi, ONT and optical maps) were aligned and analysed in gEVAL (https://vgp-geval.sanger.ac.uk/index.html). Separately, Hi-C data were mapped to the primary assembly and visualized using HiGlass. These alignments were then used by curators to identify mis-joins, missed joins and other anomalies. We identified sex chromosomes on the basis of half coverage alignments to sex chromosomes in GRCh38.

We categorized the assembly structural errors as follows: 'missed joins' are contigs that should have been neighbours in a scaffold, but were kept apart, including on different scaffolds. Missed joins are only counted if they could be resolved during curation with the available data, thereby implying that an automated process should have been able to get them right. 'Misjoins' are the opposite situation, colocalized

contigs within scaffolds that do not belong together, where one of them is often an erroneous translocation. 'Other' includes additional structural errors, which in some cases appear as erroneous inversions or false duplications. 'Unlocalized' is a sequence found in an assembly that is associated with a specific chromosome but cannot be ordered or oriented on that chromosome with the available data. 'Unplaced' are contigs or scaffolds that could not be placed on a chromosome. Finally, a 'chimeric contig' is a continuous gapless sequence that includes either an erroneous join without a gap, sequence expansions or sequence collapses.

**Continuity metrics.** Assembly continuity statistics were collected using asm_stats.sh from the VGP pipeline (https://github.com/VGP/vgp-assembly/tree/master/pipeline/stats)[11], using a genome size of 3 Gb for calculating NG50 values. All N bases were considered as gaps.

**Completeness, phasing and base call accuracies.** We collected 21-mers from Illumina reads of HG002 (250-bp paired end) and the parental genomes (HG003 and HG004) using Meryl[36], and used Merqury[36] to calculate QV, completeness and phasing statistics. Like continuity metrics, phase block NG50 was obtained using a genome size of 3 Gb. False duplications were post-calculated using false_duplications.sh in Merqury and spectra-cn histogram files for each haploid representation of the assemblies.

**Collapsed analyses.** We calculated collapsed and expandable sequences using previously described methods[74]. In brief, we aligned downsampled HiFi reads from HG002 independently to each assembly of HG002 and defined collapsed bases as regions in the assembly with greater than expected coverage (mean plus at least three standard deviations) that were at least 15 kb in length. We performed analyses with common repeat collapses included and excluded, defining common repeat collapses as sequences that were over 75% common repeat elements as identified by RepeatMasker (v4.1.0) and TRF (v4.09). This filter removed many collapses corresponding to alpha satellite and human satellite to get a better estimate of collapsed segmental duplications. Furthermore, we defined expandable Mb as the estimate of how much sequence would be in the collapsed regions had they been correctly assembled. We estimated this by multiplying the length of each collapse against the read depth divided by the average genome coverage. The code used for this analysis is available at https://github.com/mrvollger/SDA (commit ID: 23fa175).

**Strand-seq analyses.** To evaluate structural accuracy of each assembly, we first aligned Strand-seq data from HG002 to each assembly using BWA-MEM (version 0.7.15-r1140)[75] with the default parameters. Subsequently, all secondary and supplementary alignments were removed using SAMtools (version 1.9)[76] and duplicate reads were marked using Sambamba (version 0.6.8)[77]. Duplicated reads and reads with mapping quality less than 10 were removed before subsequent Strand-seq data analysis. To evaluate structural and directional contiguity of each assembly, we used R package SaaRclust[58] with the following parameters: bin.size = 200,000; step.size = 200,000; prob.th = 0.25; bin.method = 'fixed'; min.contig.size = 100,000; min.region.to.order = 500,000; ord.method = 'greedy'; num.clusters = 100; remove.always.WC = TRUE; mask.regions = FALSE; and max.cluster.length.mbp = 300. SaaRclust automatically reports contigs that probably contain a misassembly and marks them as either misorientation (change in directionality of a piece of contig) or chimerism (regions of a contig that originate from different chromosomes). To reduce false-positive calls, we report only misoriented and chimeric regions that are at least 400 kb and 1 Mb in length, respectively. Current version of the R package SaaRclust can be found at https://github.com/daewoooo/SaaRclust (devel branch).

To evaluate large (50 kb or more) inversion accuracy in the final HPRC-HG002 assembly of this study, we aligned Strand-seq separately to maternal and paternal haplotypes. Only chromosomes or scaffolds of 1 Mb or more were processed. We used breakpointR[78] to detect changes in read directionality and thus putative misassemblies across all Strand-seq libraries. We concatenated all directional reads across all available Strand-seq libraries using the breakpointR function 'synchronizeReadDir'. Next, we used the breakpointR function 'run-Breakpointr' to detect regions that are homozygous ('ww'; 'HOM') or heterozygous inverted ('wc'; 'HET') using the following parameters: bamfile = <composite_file>, pairedEndReads = FALSE, chromosomes = [chromosomes/scaffolds >=1 Mb], windowsize = 50,000, binMethod = "size", background = 0.1, minReads = 50, genoT = 'binom'. Regions designated as 'HOM' have the majority of reads in the minus direction, suggesting a homozygous inversion or misorientation assembly error. Those designated at 'HET' have roughly equal mixture of plus and minus reads, validating a true heterozygous inversion. In an ideal scenario, one would expect that assembly directionality matches the directionality of Strand-seq reads and thus no homozygous inverted regions should be visible.

**Variation benchmark analysis.** We used v4.2.1 GIAB benchmark variants with GA4GH, with v3.0 stratifications, which enabled comparative performance assessment inside and outside challenging genomic regions such as segmental duplications, homopolymers and tandem repeats[37]. Benchmarking tools from GIAB and GA4GH enabled performance to be stratified by type of error (for example, genotyping errors) and genome context (for example, segmental duplications). Variants were first called using the dipcall assembly variant calling pipeline (https://github.com/lh3/dipcall)[41]. Dipcall first aligns an assembly to the GRCh38 reference genome (ftp://ftp.ncbi.nlm.nih.gov/genomes/all/GCA/000/001/405/GCA_000001405.15_GRCh38/seqs_for_alignment_pipelines.ucsc_ids/GCA_000001405.15_GRCh38_no_alt_analysis_set.fna.gz) using minimap2 (https://github.com/lh3/minimap2)[79]. We used optimized alignment parameters -z200000,10000 to improve alignment contiguity, as this is known to improve variant recall in regions with dense variation, such as the MHC[80]. Dipcall uses the resulting alignment to generate a bed file with haplotype coverage and call variants. All filtered variants except those with the GAP2 filter were removed. GAP2-filtered variants occurred particularly in primary-alternate assemblies in homozygous regions where the alternate contig was missing. These GAP2 variants were kept as filtered to give separate performance metrics, and treated as a homozygous variant with respect to GRCh38 by changing the genotype field from 1|. to 1|1. The resulting variant calls were benchmarked using hap.py v0.3.12 with the RTG Tools (v3.10.1) vcfeval comparison engine (https://github.com/Illumina/hap.py)[42]. Earlier versions of hap.py and vcfeval do not output lenient regional variant matches to the FP.al field. The hap.py comparison was performed with the v4.2.1 GIAB HG002 small-variant benchmark vcf and bed (https://ftp-trace.ncbi.nlm.nih.gov/ReferenceSamples/giab/release/AshkenazimTrio/HG002_NA24385_son/NISTv4.2.1/GRCh38/)[26] and V3.0 of the GIAB genome stratifications (https://doi.org/10.18434/mds2-2499). To improve reproducibility and transparency, Snakemake (https://snakemake.readthedocs.io/en/stable/)[81] was used for pipeline construction and execution (https://doi.org/10.18434/mds2-2578). The extensive performance metrics output by hap.py in the extended.csv files were summarized in the following metrics for completeness, correctness and hard regions.

Completeness metric values were calculated from SNV, where the false negative (FN) rate or recall was used to assess how much of the benchmark does the callset cover, in which 100% means capturing all variants and 0% means capturing none. These completeness metric values were calculated at different stringencies with SNP.Recall or as a true positive. 'SNP.Recall_ignoreGT' is a measure of how well the assembly captures at least one of the variant alleles, and is considered true positive if at least one allele in a variant was called correctly, regardless of whether genotype was correct. This is calculated from

'(SNP.TRUTH.TP + SNP.FP.gt)/SNP.TRUTH.TOTAL' for the row with 'ALL' in the FILTER column. 'SNP.Recall' is a measure of how well the assembly represents genotypes, and counted as true positive if the variant and genotype are called correctly. When only one contig is present, we assumed the region is homozygous. This is calculated from METRIC. Recall for Type=SNP, SubType=*, SUBSET=*, FILTER=ALL. 'SNP.Recall. fullydiploid' is a measure of how well the assembly represents both haplotypes correctly, requiring that exactly one contig from each haplotype align to the location (contigs smaller than 10 kb are ignored by dipcall by default). This is calculated from METRIC.Recall for Type=SNP, SubType=*, SUBSET=*, FILTER=PASS.

Correctness metric values were calculated from the false-positive rate for SNVs and indels, converted into a phred scaled per base error rate. Each SNP and indel was counted as a single error on one haplotype regardless of size and genotype. 'QV_dip_snp_indel' is the error rate in all benchmark regions, calculated as '$-10 \times \log_{10}((SNP.QUERY.FP + INDEL.QUERY.FP)/(Subset.IS\_CONF.Size \times 2))$'. 'NoSegDup.QV_dip_snp_indel' is the same as QV_dip_snp_indel, except that it excludes segmental duplication regions.

Hard region metric values were calculated for particularly difficult-to-assemble regions such as segmental duplications. 'Segdup. QV_dip_snp_indel' is the same as the 'QV_dip_snp_indel' correctness metric, but only for segmental duplication regions.

To benchmark SVs, we aligned the final HG002 assembly to GRCh37 and used truvari v3.1.0 to benchmark variants against the GIAB tier 1 v0.6 benchmark vcf in v0.6.2 benchmark regions.

**BUSCO analyses.** Busco completeness for the 41 assemblies was calculated with BUSCO v3.1.0 using the mammalia_odb9 lineage set (https://busco-archive.ezlab.org/v3/)[82].

**Annotations.** Human RefSeq transcripts of type 'known' (with NM or NR prefixes[83]) were queried from RefSeq on 8 December 2021. The query to access these is: 'Homo_sapiens[organism] AND srcdb_refseq_known[properties] AND biomol_rna[properties]', although because of curation, this query will return a different set of transcripts today that it did in December 2021. Each transcript is the child of exactly one gene, but a given gene can be the parent of multiple transcripts (alternative variants). The returned 81,571 transcripts were aligned to the 43 assembled haplotypes and to GRCh38 (GCF_000001405.26) and T2T-CHM13 v1.1 (GCA_009914755.3). The coding transcripts and non-coding transcripts longer than 300 bp were first aligned with BLAST (e-value of 0.0001, word size 28 and best-hits options best_hit_overhang = 0.1 and best_hit_score_edge = 0.1) to the genomes masked with Repeat-Masker (www.repeatmasker.org)[84] or WindowMasker[85]. Sets of results obtained with both masking methods were passed to the global alignment algorithm Splign[86] (75% minimum exon identity, 50% minimum compartment identity and 20% minimum singleton identity) to refine the splice junctions and align exons missed by BLAST. Sequences for which no alignment with coverage higher than 95% of the query and sequences with unaligned overhangs at the 5' or 3' end were realigned with BLAST and Splign to the unmasked genome, and then submitted to the same filter. Non-coding transcripts shorter than 300 bp were aligned with BLAST to the unmasked genome (e-value of 0.0001, word size 16, 98% identity and best-hits options best_hit_overhang = 0.1 and best_hit_score_edge = 0.1) and then with Splign (75% minimum exon identity, minimum compartment identity and minimum singleton identity) and submitted to the same filter as the other transcripts. The alignments for each transcript were then ranked on the basis of identity and coverage. Transcripts that aligned best to GRCh38 sex chromosomes were filtered out of the alignments to all assemblies, resulting in 78,492 transcripts in 27,225 corresponding autosomal genes, for which we calculated the statistics in Supplementary Table 2k.

Several measures for assembly completeness and correctness, and for sequence accuracy were compared across all assembly haplotypes:

(1) genes with unaligned transcripts, either due to one or more transcript alignments being absent or too low in sequence identity; (2) split genes, across two or more scaffolds; (3) low-coverage genes, with less than 95% of the coding sequence in the assembly; (4) dropped genes, most often due to collapsed regions in the assembly; and (5) genes with frameshifted CDSs, where the best-ranking alignment requires insertions or deletions to compensate for suspected insertions or deletions in the genomic sequence that cause frameshift errors. For category 4, as each RefSeq transcript is associated with a single gene[87] and genes are not expected to overlap, unless explicitly known to, collapsed regions were identified as loci where transcripts from multiple genes co-aligned, and measured as the count of genes for which the best alignment of a transcript needed to be dropped to resolve the conflict. A set of 119 genes with transcripts that failed to align to either GRCh38, T2T-CHM13 v1.1, HG002.mat or HG002.pat were examined further. A total of 106 of these had no other aligned transcripts and were therefore completely missing from one or more assemblies. The remaining 13 genes had some but not all children transcript spliced variants that aligned.

## Pangenomic assessment of the assemblies

We performed pairwise alignments for all chromosomes of all 45 assemblies with the wfmash sequence aligner (https://github.com/ekg/wfmash; commit ID: 09e73eb), requiring homologous regions at least 300 kb long and nucleotide identity of at least 98%. We used the alignment between all assemblies to build a pangenome graph with the seqwish variation graph inducer[88] (commit ID: ccfefb0), ignoring alignment matches shorter than 79 bp (to remove possible spurious relationships caused by short repeated homologies). To obtain chromosome-specific pangenome graphs, contigs were partitioned by aligning all of them with wfmash against the GRCh38 and CHM13 reference sequences. Graph statistics, visualizations and pairwise Jaccard similarities and Euclidean distances between haplotypes were obtained with the ODGI toolkit[89] (commit ID: 67a7e5b). We performed the multidimensional analyses in the R development environment (version 3.6.3), equipped with the following packages: tidyverse (version 1.3.0), RColorBrewer (version 1.1.2), ggplot2 (version 3.3.3), ggrepel (version 0.9.1) and stats (version 3.6.3). Specifically, we applied the classical multidimensional scaling on the Euclidean distance matrix to perform the PCA. Pangenome graphs at selected loci were built and visualized by using the PGGB pipeline (commit ID: 5d26011) and the ODGI toolkit. Code and links to data resources used to build the pangenome graphs to perform the multidimensional analyses and to produce all of the figures can be found at the following repository: https://github.com/AndreaGuarracino/HG002_assemblies_assessment.

## Heterozygosity analysis

To call the full spectrum of heterozygosity between the two haploid sequences, we directly compared two haploid assemblies using Mummer (v4.0.0rc1) with the parameters of 'nucmer -maxmatch -l100 -c 500'. SNP and small indels were generated by 'delta-filter -m -i 90 -l 100' and followed by 'dnadiff'. Several custom scripts were used to analyse the Mummer output, as described in our previous marmoset study (https://github.com/comery/marmoset). We used SyRi[90] (v1.5) to detect SVs from Mummer alignments using default parameters. SVs in which more than half the feature consisted of gaps were excluded. For CNVs, we only included local tandem contractions or expansions; whole-genome copy number changes were not included in these results. To avoid false positives caused by assembly issues and insufficient detection power, we only included intrachromosomal translocations (50 bp or more) in which haplotypes reciprocally share the best alignment.

## Alignments between reference assemblies

All scaffolds of the final HG002 maternal and paternal assemblies were aligned by minimap2 to the T2T-CHM13 reference and the Y

chromosome of GRCh38. Some of the contigs within the HG002 scaffolds had alternate alignments to CHM13, which we did not include in the analyses to avoid the ambiguity. The phase density of contigs were calculated using the parental short reads. We then extracted haplotype-specific *k*-mers from contigs and determined the colour value by the number of these *k*-mers.

### Gene duplication analysis

Gene duplications were measured using multi-mapped gene bodies and read depth. Gencode v29 transcripts were aligned using minimap2 (version 2.17-r941) to annotate gene models. The genomic sequence of each gene was re-mapped to both HPRC-HG002 v1.0 maternal and paternal assemblies allowing for multimapped alignments. Gene duplications were counted as genome sequences aligned with at least 90% identity and 90% of the length of the original gene. Spurious duplications were annotated by mapping all reads back to each haplotype assembly, and filtering on low (less than 0.05) read depth. The code to annotate duplicated genes is available (https://github.com/ChaissonLab/SegDupAnnotation/releases/tag/vHPRC).

### Consent

Informed consent was obtained by the Personal Genome Project, which permits open sharing of genomic data, phenotype information and redistribution of cell lines and derived products.

### Reporting summary

Further information on research design is available in the Nature Research Reporting Summary linked to this article.

## Data availability

All raw sequence data used in this study are available at the following HPRC GitHub: https://github.com/human-pangenomics/HG002_Data_Freeze_v1.0. The final HPRC-HG002 curated assemblies are available in the NCBI under the BioProject IDs PRJNA794175 and PRJNA794172, with the accession numbers GCA_021951015.1 and GCA_021950905.1, for the maternal and paternal haplotypes, respectively. Assemblies, variant calls and GIAB benchmarking results are available at https://doi.org/10.18434/mds2-2578.

## Code availability

The following codes were used for the assemblies and analyses of this study. Detailed use of codes are mentioned in the methods and main text. For assemblers: Bionano Trio Binning https://github.com/andypang0714/Bionano_Trio_binning; Canu https://github.com/marbl/canu; CrossStitch https://github.com/schatzlab/crossstitch; DipAsm https://github.com/shilpagarg/DipAsm; FALCON-Unzip: https://anaconda.org/bioconda/pb-falcon/2.2.4/download/linux-64/pb-falcon-2.2.4-py37hed50d52_0.tar.bz2; Flye https://github.com/fenderglass/Flye; HiCanu https://bioinformaticshome.com/tools/wga/descriptions/HiCANU.html; hifiasm https://github.com/chhylp123/hifiasm/releases/tag/v0.3; HPRC v1 pipeline: https://github.com/gf777/misc/tree/master/HPRC%20HG002/for_filling; MaSuRCA https://github.com/alekseyzimin/masurca; MitoHiFi https://github.com/marcelauliano/MitoHiFi; NECAT https://github.com/xiaochuanle/NECAT; Peregrine https://github.com/cschin/Peregrine; PGAS pipeline https://github.com/ptrebert/project-diploid-assembly; SALSA https://github.com/marbl/SALSA/commit/974589f3302b773dcf0f20c3332fe9daf009fb93; Shasta https://github.com/chanzuckerberg/shasta; VGP Trio binning assembly tools: https://github.com/VGP/vgp-assembly/tree/master/pipeline; and wtdbg2 https://github.com/ruanjue/wtdbg2. For base polishing: Medaka https://github.com/nanoporetech/medaka; Mercury https://github.com/marbl/merqury; and PEPPER https://github.com/kishwarshafin/pepper. For processing

and analysis tools: automated analyses https://dockstore.org/workflows/github.com/human-pangenomics/hpp_production_workflows/Merqury:master?tab=info; automated contamination removal https://github.com/human-pangenomics/hpp_production_workflows/blob/master/QC/wdl/tasks/contamination.wdl; CutAdapt https://github.com/marcelm/cutadapt; HiFiAdapterFilt https://github.com/sheinasim/HiFiAdapterFilt; hap.py https://github.com/Illumina/hap.py; Meryl https://github.com/marbl/meryl/tree/master/src/meryl; pangenome assessment https://github.com/AndreaGuarracino/HG002_assemblies_assessment; SaaRclust https://github.com/daewoooo/SaaRclust; SDA https://github.com/mrvollger/SDA; and SegDupAnnotation https://github.com/ChaissonLab/SegDupAnnotation/releases/tag/vHPRC.

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

**Acknowledgements** Certain commercial equipment, instruments or materials are identified to specify adequately experimental conditions or reported results. Such identification does not imply recommendation or endorsement by the National Institute of Standards and Technology, nor does it imply that the equipment, instruments or materials identified are necessarily the best available for the purpose. T.M. and P.E. (asm14 team) acknowledge the computational infrastructure and support provided by the Centre for Information and Media

Technology at Heinrich Heine University Düsseldorf. We thank P. Audano of the Eichler laboratory for bioinformatic support. This work utilized the computational resources of the National Institutes of Health HPC Biowulf cluster (https://hpc.nih.gov). M.C.S and M. Kirsche (asm17 team) acknowledge A. Carroll and P.-C. Chang from Google AI for assistance running DeepVariant. The primary source of funding for this study was from NHGRI grant U01HG010971 (https://grants.nih.gov/grants/guide/rfa-files/rfa-hg-19-004.html). Additional funding included: Howard Hughes Medical Institute (to E.D.J.), NHGRI grants HG002385 and HG010169 (to E.E.E.), and NHGRI grant R01HG011274-01 (to K.H.M). The computational resources and personnel support for the DipAsm assemblies were HG010906/HG/NHGRI NIH HHS and NNF21OC0069089 to S.G.; NCI U01CA253481 to M.C.S.; and NSF DBI-1627442 to M.C.S. T.M. and P.E. (asm14 team) acknowledge the support by the BMBF-funded de.NBI Cloud within the German Network for Bioinformatics Infrastructure (de.NBI) (031A532B, 031A533A, 031A533B, 031A534A, 031A535A, 031A537A, 031A537B, 031A537C, 031A537D and 031A538A), and the German Research Foundation (391137747 to T.M.), and the National Natural Science Foundation of China (62150048 to J. Wang). E.D.J., E.E.E. and D.H. are investigators of the Howard Hughes Medical Institute. This work was supported in part by the National Center for Biotechnology Information of the National Library of Medicine, National Institutes of Health; the Intramural Research Program of the NHGRI, National Institutes of Health; and by National Institute of Standards and Technology intramural research funding.

**Author contributions** E.D.J. co-coordinated evaluation, performed analyses and wrote the paper. K.H. co-coordinated evaluation and curation. K.H.M. provided overall coordination. A. Rhie performed the mercury metric evaluations. T.L. carried out the assembly validations. V.A.S. performed genome alignment and annotation and/or frameshift issues. B.P. carried out the ONT assemblies and provided overall coordination. M.H., T. Pesout and K.S. performed the Shasta assembly, CAT-annotation and QUAST. J.M.Z., I.H., J.M., N.D.O., J. Wagner and A. Regier performed the variant analyses. E.B.J. provided Illumina data. E.E.E. and M.R.V. performed the segmental duplication assembly (SDA) and collapsed duplication analyses. P.E. performed assembly asm14. D. Porubsky performed inversion validation and Strand-seq analyses. W.H. provided bioinformatic support. H.L. carried out genome assembly and overall analyses. S.G. performed genome assembly and phasing evaluation. E.G. and A. Guarracino carried out pangenome alignments, Jaccard, PCA and MHC. M.E. performed PacBio HiFi adaptor assessments. F.T.-N. performed annotations, gene loss and associated analyses. Curation was done by J. Wood, J.C., J.T. and A.T. J.L.G., T. Potapova and M.W.M. performed karyotype and cell culture analyses. M.A. carried out genome assemblies and titration. A. Granat generated Illumina data. C.Y. and G.Z. performed heterozygosity circos plot analyses. G.A.L. carried out centromere and telomere analyses. G.F. provided asm23 and the final HRPC HG002 references. H.C. provided asm9. S.K. provided asm6, asm7, asm19, asm20 and MHC evaluation. A.M.P. coordinated assembly. J.G. provided asm2 and asm22. T.M. and P.E. provided asm14. M.C.S. provided asm17. M. Kirsche provided asm17. C.-S.C. provided asm3 and asm4. J.K. and A.W. provided asm16. S.L.S. provided asm1 and asm15. J.R. provided asm13. F.N., C.X., J. Wang and F.L. provided asm12. A.V.Z. provided asm1 and asm15. A. Shumate and D. Puiu provided asm1. M.J.P.C. performed HG002 reference segmental duplication expansions. P.C. and T. Pesout performed Shasta assembly. N.O.S. carried out sample collection, processing and coordination. O.F. performed Bionano analyses. M. Kolmogorov carried out Flye haploid assemblies. J. Lucas carried out sample processing, analyses and submissions. M.J. generated nanopore data, performed analysis and managed data. R.E.G. performed Dovetail Genomics Hi-C. S.S. and A. Schmitt performed Arima Genomics Hi-C. R.S.F. carried out PacBio production activities and evaluation methods. H.E.O. generated nanopore data. A.D.S., J.O.K., P.H. and C.S. produced Strand-seq data. A.H. and J. Lee provided Bionano diploid assembly maps. D.H. and T.W. provided HPRC coordination. L.L.F., D.L. and S.C. carried out logistics and data release, and provided project support.

**Competing interests** E.E.E. was a scientific advisory board member of Variant Bio, Inc. J.K. and A.W. were full-time employees at Pacific Biosciences, a company developing single-molecule sequencing technologies. A.H. and J. Lee. were employees of Bionano Genomics, a company developing optical maps for genome assembly. J.G. and R.E.G. were affiliated with Dovetail Genomics, a company developing genome assembly tools, including Hi-C. A. Granat and E.B.J. were employees of Ilumina, Inc., a genome company generating short reads. A. Schmitt and S.S. were employees of Arima Genomics, a company developing Hi-C data for genome assemblies. All other authors declare no competing interests.

**Additional information**
**Correspondence and requests for materials** should be addressed to Erich D. Jarvis, Giulio Formenti, Kerstin Howe or Karen H. Miga.

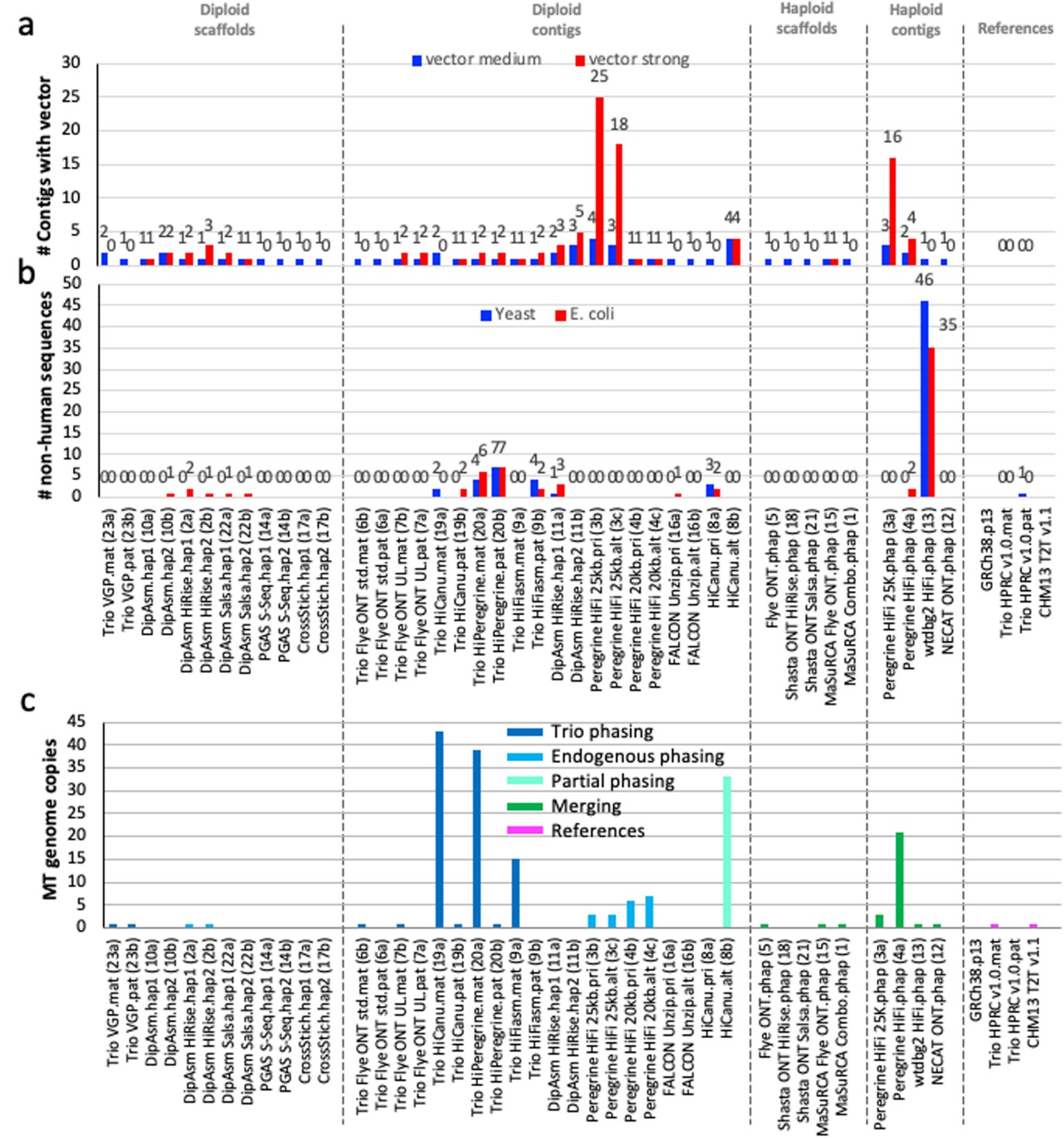

**Extended Data Fig. 1 | Non-human and organelle genomes found in the human genome assemblies. a**, The number of contigs that had remaining library clone vector sequences in each assembly. Medium used a blastn score 19-29; strong a score > 30 https://www.ncbi.nlm.nih.gov/tools/vecscreen/ about/. **b**, The number of contigs with non-human yeast and *E.coli* sequences. Values above columns are the specific numbers. **c**, The number of endogenous mitochondrial genome sequences found in each assembly.

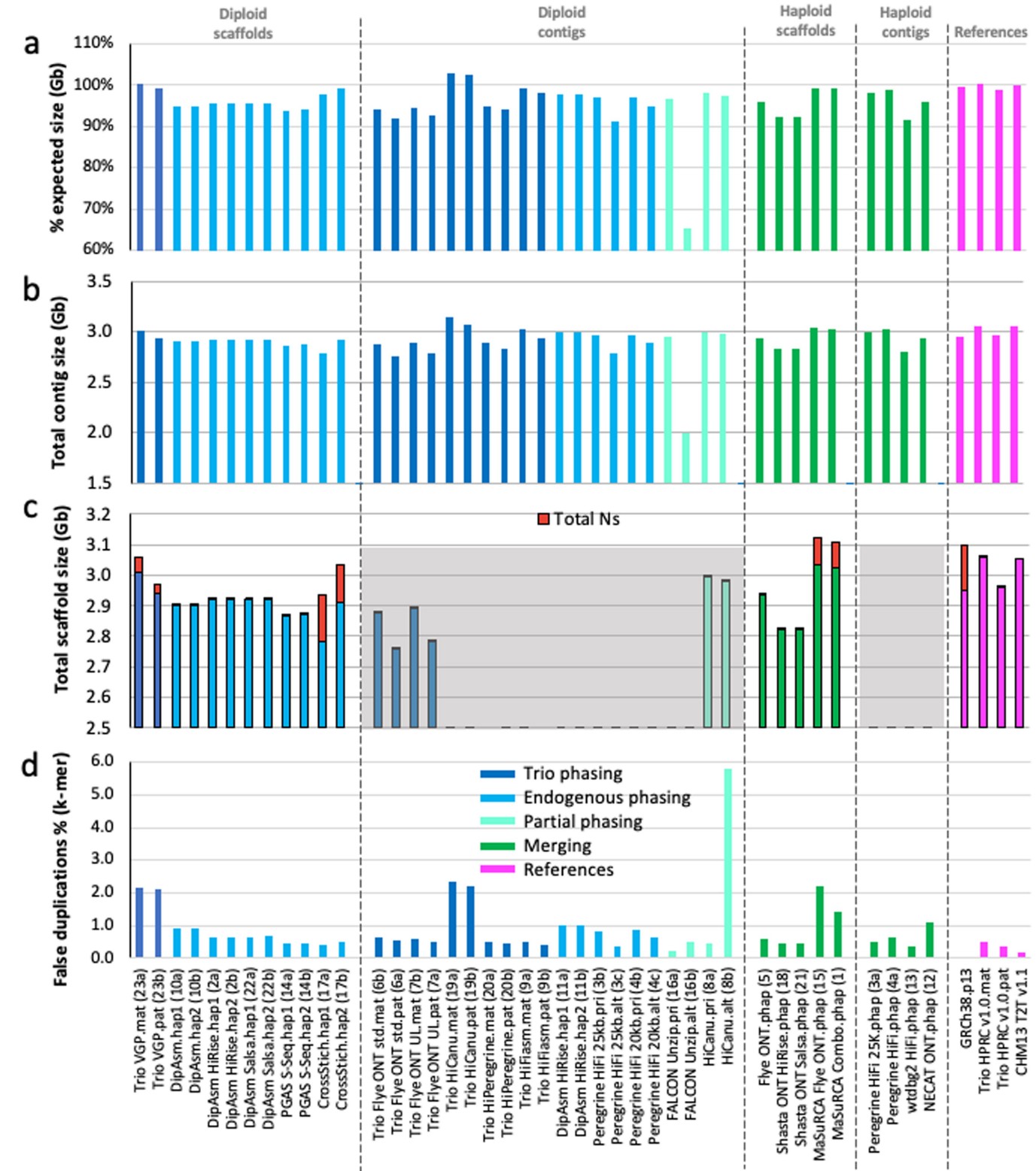

**Extended Data Fig. 2 | Assembly size and false duplication metrics.**
**a**, Percent assembly sizes of expected maternal with Chr X (3,054,832,041 bp) or paternal with Y (2,995,432,041 bp) for trio-based assemblies, or simply relative to maternal size for all other assemblies. **b**, Total summed length of all contigs. **c**, Total summed length of scaffolds, with proportion contributed by Ns (red) in gaps. **d**, estimated percent of assembly size that is due to false duplications based on k-mer values for each haplotype. Color coding and gray shaded regions are as described in Fig. 1.

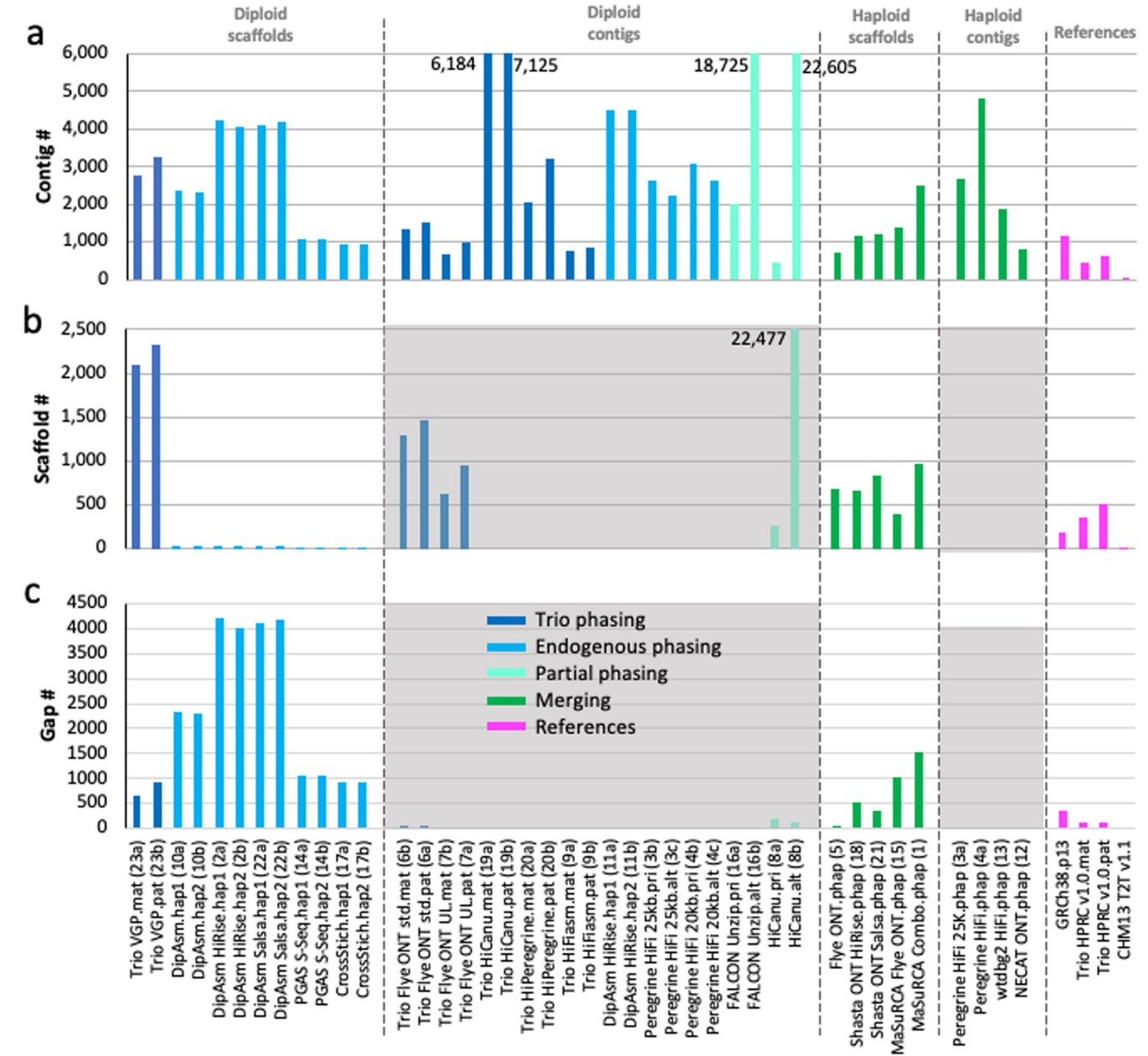

**Extended Data Fig. 3 | Contig, scaffold, and gap metrics. a**, Total number of contigs in each assembly. **b**, Total number of scaffolds in each assembly. **c**, Total number of gaps in each assembly. Values above the maximum on the y-axis are written in the graph so as to not visually scale down the majority of the results. Color coding and gray shaded regions are as described in Fig. 1.

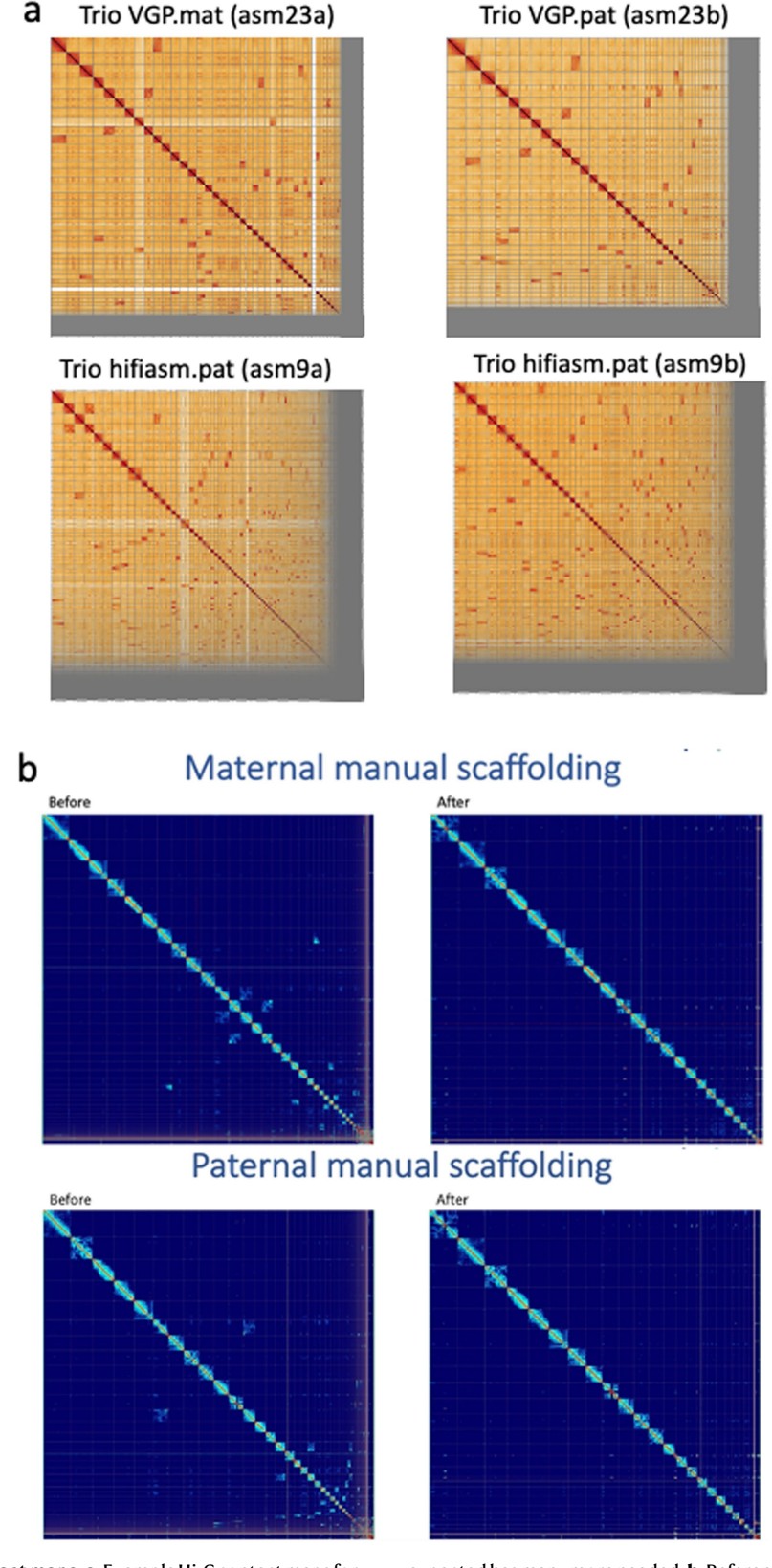

**Extended Data Fig. 4 | Hi-C contact maps. a**, Example Hi-C contact maps for bakeoff maternal (mat) and paternal (pat) haplotype assemblies. The Trio VGP scaffolded assembly has several dozen large joins and many small ones to make from the off-diagonal signals. The Trio hifiasm contig only assembly as expected has many more needed. **b**, Reference HPRC HG002 assemblies for each haplotype before and after manual curation, showing less off diagonal signals and no major scaffolds/contigs not placed in chromosomes after curation.

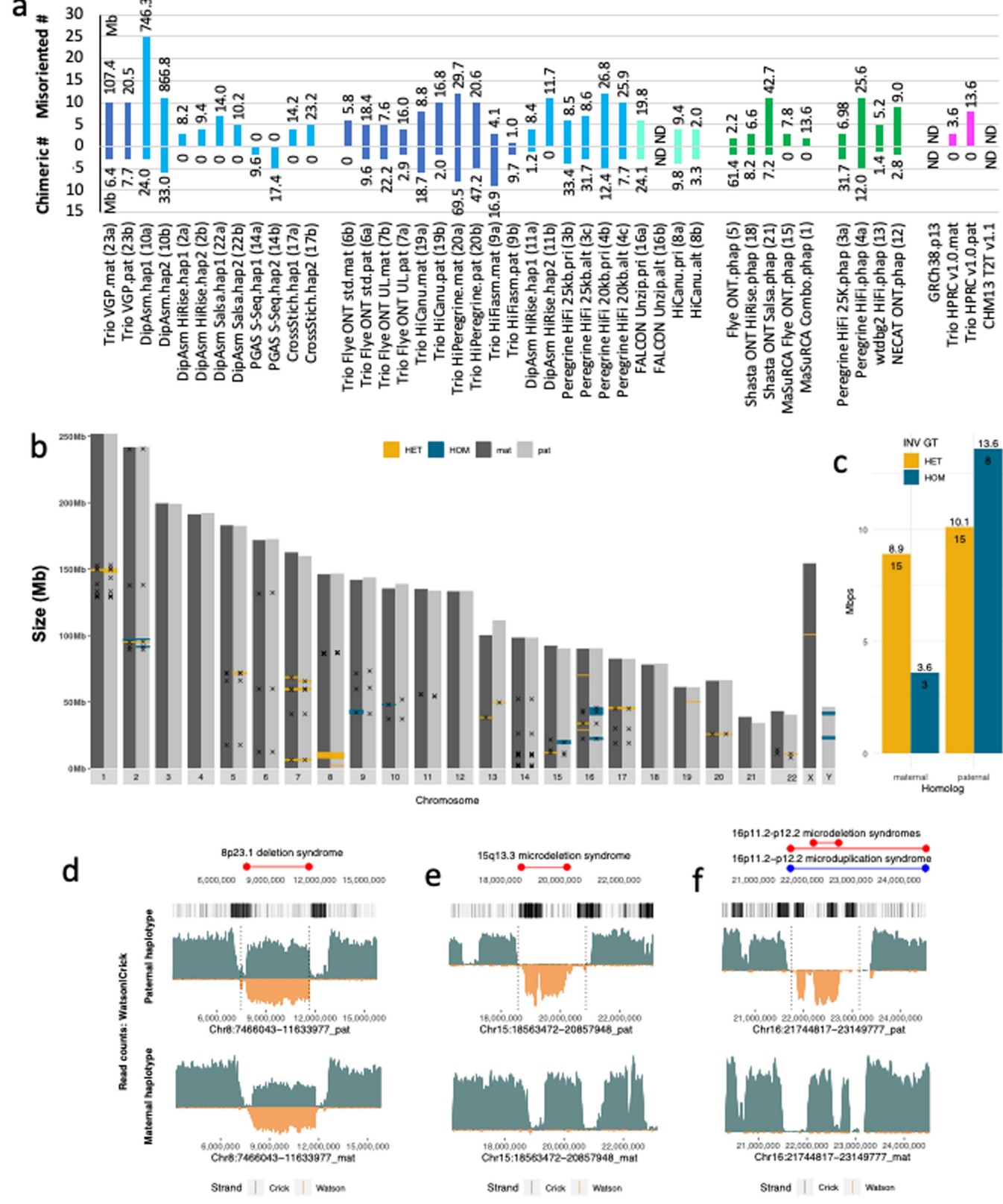

**Extended Data Fig. 5 |** See next page for caption.

**Extended Data Fig. 5 | Strand-seq validations. a**, Total number and total Mb of chimeric and misorientation errors for each assembly according to Strand-seq validations. **b**, Large (> 50 kb) Strand-seq supported and unsupported inversions (x location; n = 59) between HG002 haplotypes. HET, regions with roughly equal mixture of plus (Crick) and minus (Watson) Strand-seq reads supporting the heterozygous inversions (yellow, n = 30). HOM, regions with Strand-seq reads mapped to the opposite orientation in disagreement with heterozygous inversions and thus a possible assembly error (blue, n = 11). **c**, Barplot of total size and total number of regions genotyped as HET and HOM validated inversions. **d-f**, Example heterozygous assembly inversions that matched (**d**) or did not match (**e,f**) the Strand-seq read direction in the final HG002 assembly. First track: Known morbid CNVs (red, deletions; blue,

duplications). Second track: Segmental duplications (black marks - DupMasker) in the paternal assembly. Third and fourth tracks: Coverage of Strand-seq reads aligned to the HG002 paternal and maternal assemblies (binsize: 50 kb, stepsize: 1 kb) with Crick (teal, above) and Watson (below, orange) read counts. Regions with roughly equal coverage of Watson and Crick counts represent validated heterozygous inversions, as only one homolog is inverted with respect to the de novo assembly (**d**); Regions with only Watson coverage orientation represent an assembly error, because assembly directionality does not match Strand-seq read directionality (**e,f**). Vertical dotted lines highlight the predicted breakpoints of assembly errors as well as predicted heterozygous inversion.

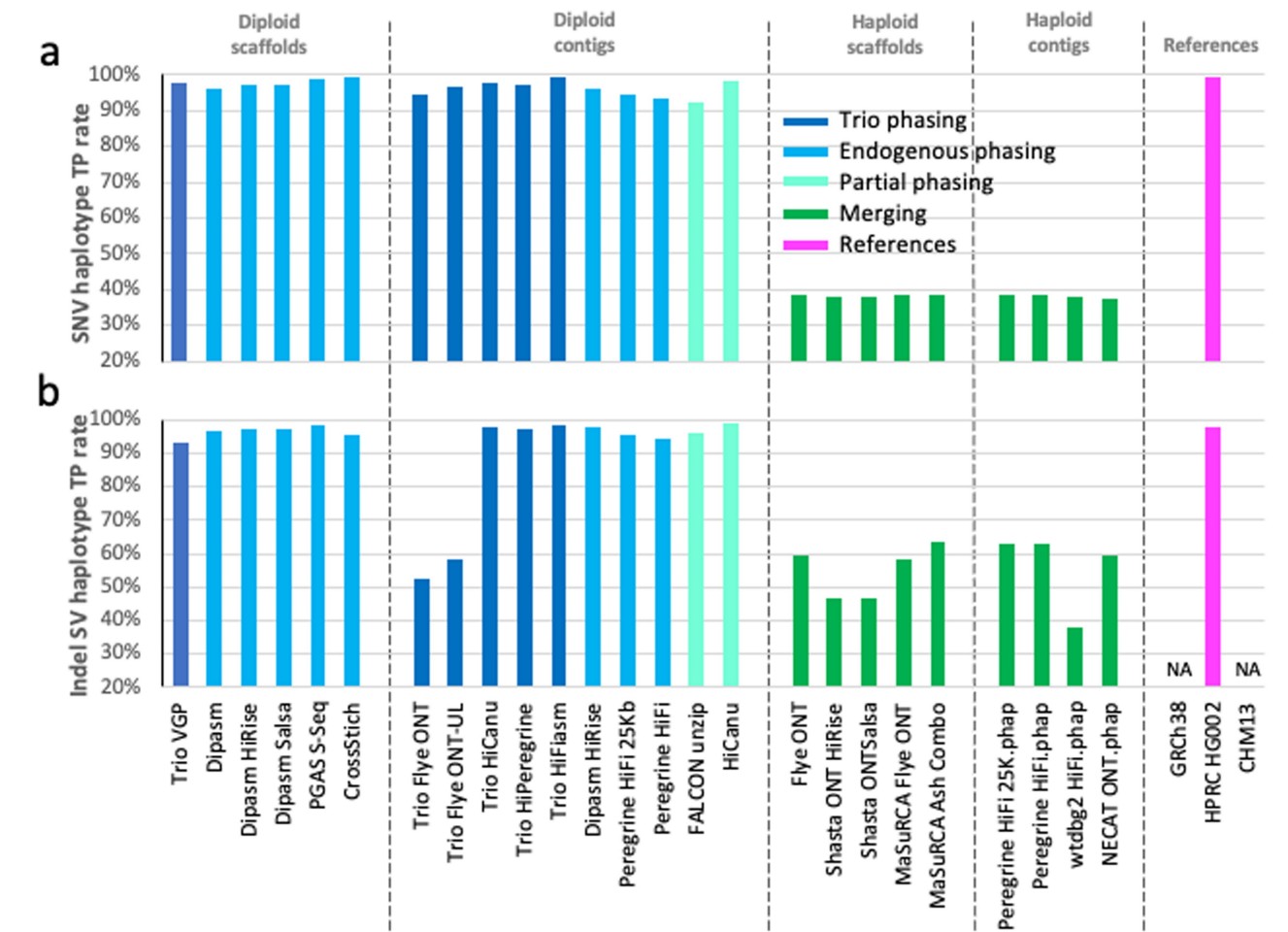

**Extended Data Fig. 6 | Variant benchmarking. a**, True positive percent of known SNVs found between HG002 haplotypes in each assembly. **b**, True positive percent of known small indels found between HG002 haplotypes in each assembly. For the diploid assemblies, comparisons were made between the two haplotypes (maternal vs paternal for the trio assemblies; haplotype 1 vs haplotype 2 for the non-trio assemblies). For the haploid assemblies, we scored as TP if at least one of the variants were found.

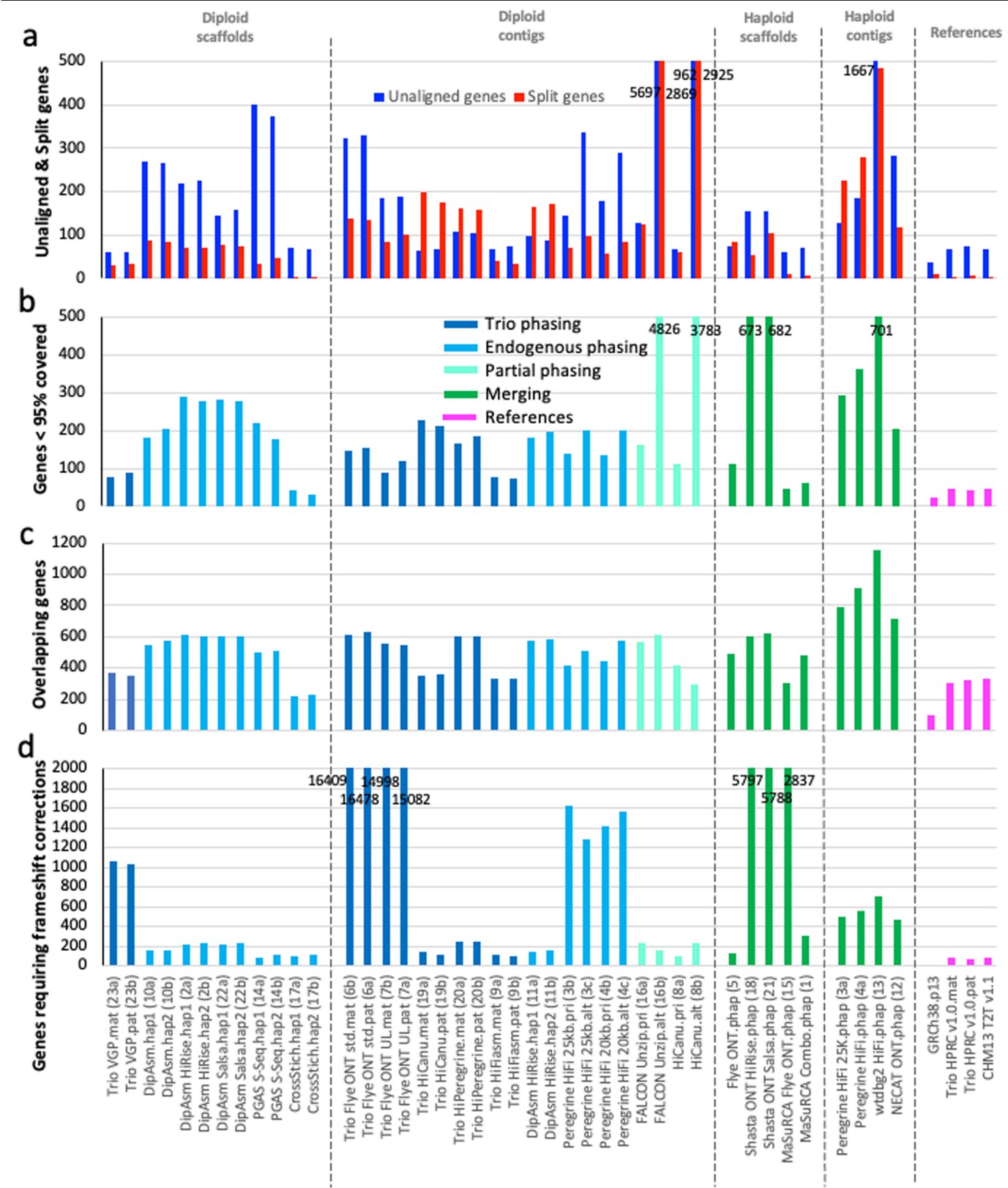

**Extended Data Fig. 7 | Annotation benchmarking. a**, Side-by-side comparisons of gene transcripts that did not align to each assembly (blue) versus those that were split between two or more scaffolds/contigs (red). **b**, Number of genes that had less than 95% the length covered in the assembly. **c**, Genes in the assemblies with overlapping transcripts due to possible collapse in the assemblies. **d**, Genes requiring frameshift corrections to make a complete protein. Values written in the graphs are for those off the chart, in order to not mask the lower values of most other assemblies.

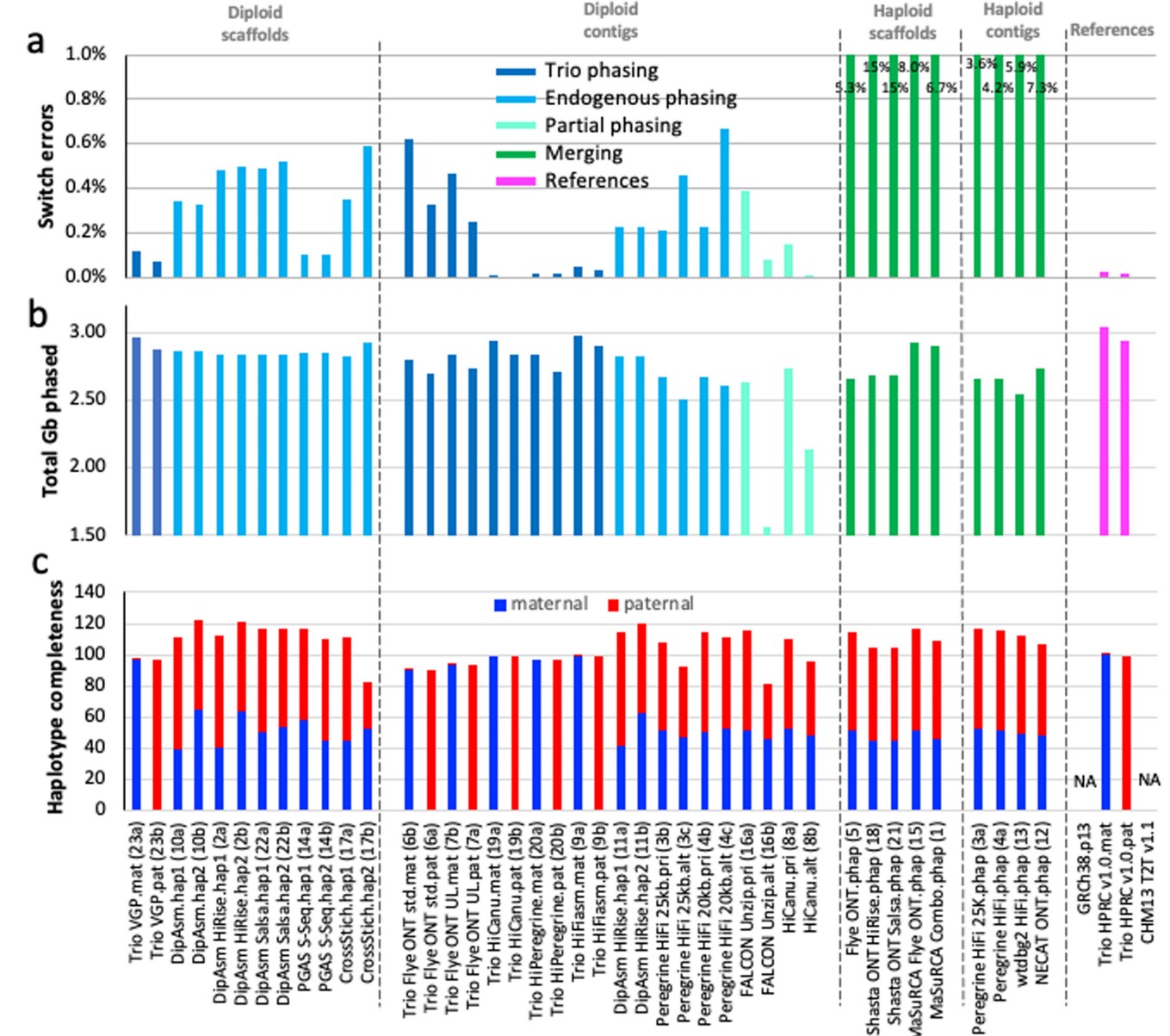

**Extended Data Fig. 8 | Haplotype phasing metrics. a**, Haplotype switch errors within scaffolds and/or contigs of each assembly (lower % is more accurate). Values written in the graphs for the haploid assemblies (greens) are off the chart, in order to not mask the lower switch error values of most other assemblies. **b**, Total Gb of each assembly that has been haplotype phased (~3.0 is the theoretical maximum of the maternal haplotype; 2.9 for the paternal). **c**, Haplotype phasing completeness according to parental *k-mer* statistics for each assembly. A complete phased assembly will have both maternal (blue) and paternal (red) each at 100% without mixture from the other. The trio approaches had nearly full phase separation, whereas the non-trio approaches nearly had half and half separation because there was not an attempt to phase across contigs or scaffolds/chromosomes belonging to the same maternal or paternal haplotypes. Combined values over 100% indicate a mixture of haplotype presumably due to false duplications; although values under 100% could still have false duplications.

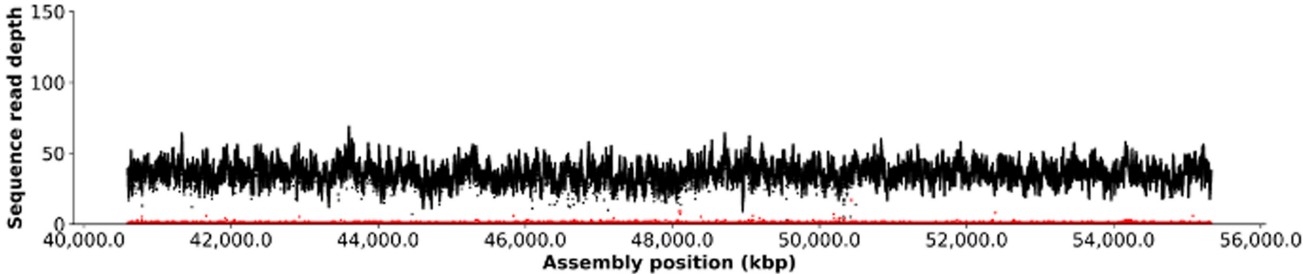

**a**, Estimated amount of bp that are collapsed in each assembly (smaller is better). Collapses are most often due to repetitive sequences. **b**, Estimated amount of bp that are potentially expandable. The smaller, the more accurate the assembly. We estimate that most of these collapses are in centromeric regions and satellites, with a smaller proportion coming from segmental duplications. Abbreviations and color coding explanations are the same as in Fig. 1 legend. **c**, Example collapse region of one of the HG002 assemblies, where read coverage pile up in the collapsed region is two or more times higher than the mean coverage of the genome. **d**, Example of HiFi read coverage across a centromere, of HG002 maternal Chr 11, showing no evidence of collapsed repeats or coverage dropouts.

**Extended Data Fig. 9 | Collapsed sequence metrics.**

## a, Orientation vs. whole genome alignments across assemblies

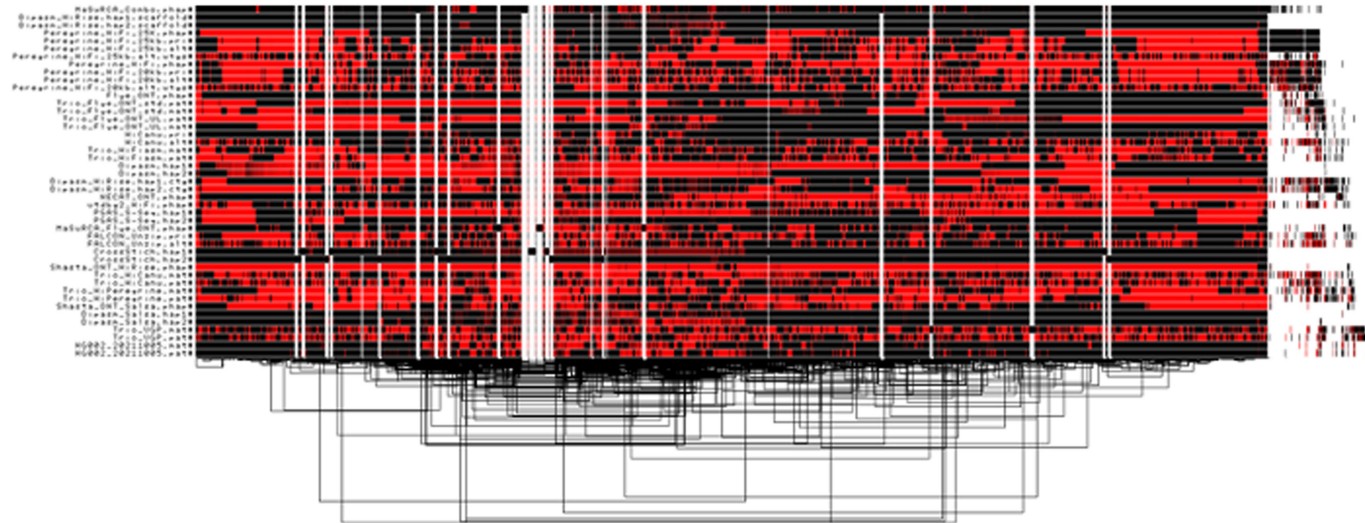

## b, HPRC Trio pipeline v1.0 used to generate first HPRC HG002 references

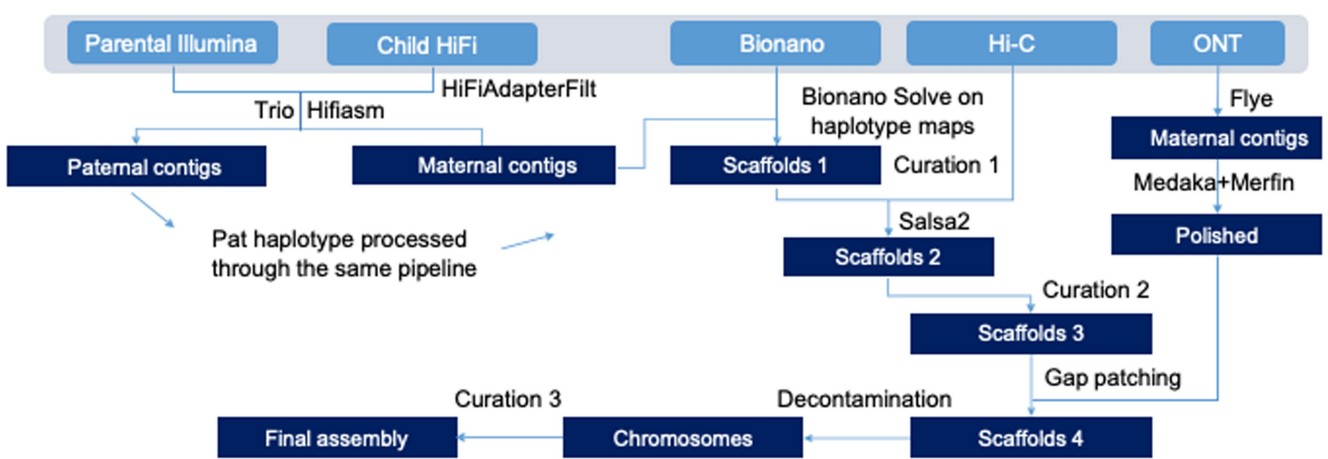

**Extended Data Fig. 10 | Pangenome alignment and generation of high-quality HPRC-HG002 v1.0 diploid assemblies. a,** Output of graph-based alignment of all chromosomes concatenated from all 45 HG002 assemblies (both haplotypes of diploid assemblies). Red vs Black, different orientations. Dendogram at bottom is a clustering of the alignments. **b,** HPRC v1.0 pipeline developed to produce the reference quality HPRC-HG002 v1.0 maternal and paternal assemblies. All steps shown are highlighted for the maternal data. The key steps of the pipeline are available in the Galaxy Server (https://assembly.usegalaxy.eu/) and best practices from this study at https://github.com/human-pangenomics/hpp_production_workflows/wiki/Assembly-Best-Practices.

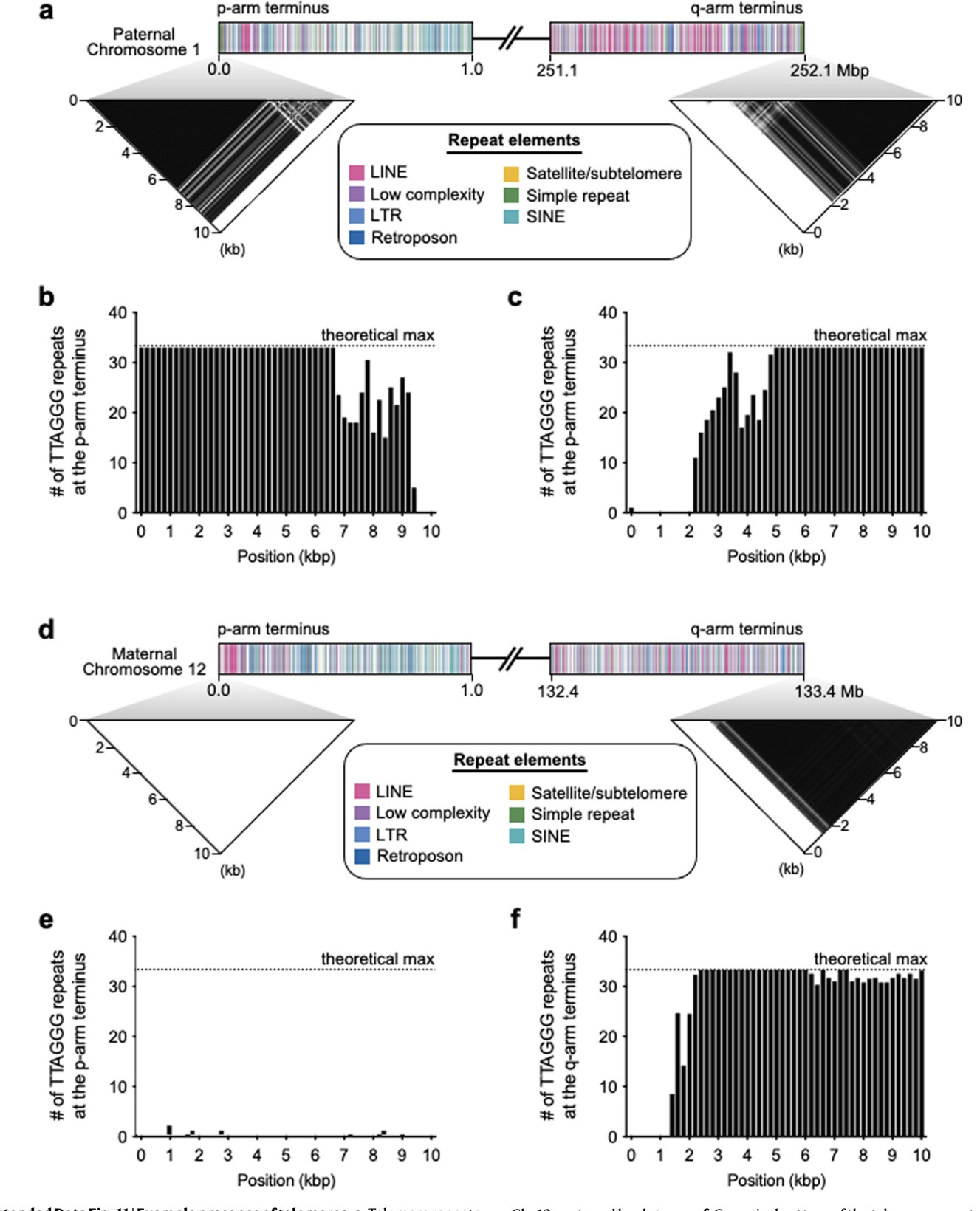

**Extended Data Fig. 11 | Example presence of telomeres. a**, Telomere repeats within 10 kb of each arm of HG002 Chr 1, paternal haplotype. The darker the density, the higher the repeat copy number. **b-c**, Density of telomere repeats for each arm, in 200 bp bins. 33 x 6-bp repeats is the theoretical maximum per 200 bp. **d**, Telomere repeats within 10 kb only found for the q-arm of HG002 Chr 12, maternal haplotype. **e-f**, Canonical pattern of the telomere repeats only found in the q-arm of the HG002 Chr 12, maternal assembly. Color coding, the different types of repeats found within 1 Mb of each arm. The similar patterns between Chr 1 and 12 indicate that only the p-arm telomere is missing from Chr 12.

# Reporting Summary

## Statistics

For all statistical analyses, confirm that the following items are present in the figure legend, table legend, main text, or Methods section.

| n/a | Confirmed | |
|---|---|---|
| ☐ | ☒ | The exact sample size (*n*) for each experimental group/condition, given as a discrete number and unit of measurement |
| ☐ | ☒ | A statement on whether measurements were taken from distinct samples or whether the same sample was measured repeatedly |
| ☐ | ☒ | The statistical test(s) used AND whether they are one- or two-sided  *Only common tests should be described solely by name; describe more complex techniques in the Methods section.* |
| ☒ | ☐ | A description of all covariates tested |
| ☒ | ☐ | A description of any assumptions or corrections, such as tests of normality and adjustment for multiple comparisons |
| ☐ | ☒ | A full description of the statistical parameters including central tendency (e.g. means) or other basic estimates (e.g. regression coefficient) AND variation (e.g. standard deviation) or associated estimates of uncertainty (e.g. confidence intervals) |
| ☐ | ☒ | For null hypothesis testing, the test statistic (e.g. *F*, *t*, *r*) with confidence intervals, effect sizes, degrees of freedom and *P* value noted  *Give P values as exact values whenever suitable.* |
| ☒ | ☐ | For Bayesian analysis, information on the choice of priors and Markov chain Monte Carlo settings |
| ☒ | ☐ | For hierarchical and complex designs, identification of the appropriate level for tests and full reporting of outcomes |
| ☒ | ☐ | Estimates of effect sizes (e.g. Cohen's *d*, Pearson's *r*), indicating how they were calculated |

*Our web collection on statistics for biologists contains articles on many of the points above.*

## Software and code

Policy information about availability of computer code

| Data collection | All code used to collect data are cited in github links and citations of the manuscript |
|---|---|
| Data analysis | All code used to analyzes the data are cited in github links and citations of the manuscript |

For manuscripts utilizing custom algorithms or software that are central to the research but not yet described in published literature, software must be made available to editors and reviewers. We strongly encourage code deposition in a community repository (e.g. GitHub). See the Nature Portfolio guidelines for submitting code & software for further information.

## Data

Policy information about availability of data

All manuscripts must include a data availability statement. This statement should provide the following information, where applicable:
- Accession codes, unique identifiers, or web links for publicly available datasets
- A description of any restrictions on data availability
- For clinical datasets or third party data, please ensure that the statement adheres to our policy

All raw sequence data used in this study is available at the following HPRC Github: https://github.com/human-pangenomics/HG002_Data_Freeze_v1.0. The final HPRC-HG002 curated assemblies are available in NCBI under the BioProject IDs PRJNA794175 and PRJNA794172, with accession numbers GCA_021951015.1 and GCA_021950905.1, for the maternal and paternal haplotypes respectively.

# Human research participants

Policy information about studies involving human research participants and Sex and Gender in Research.

| | |
|---|---|
| Reporting on sex and gender | The biological sex (male and female) of the samples used have been noted, as our goal was to include genome sequencing and assembly of both X and Y sex chromosomes. |
| Population characteristics | The population that the individuals are from (Ashkenazi, African American) have been noted, and approved in the original informed consents. |
| Recruitment | Participants were recruited in separate studies from this study |
| Ethics oversight | Personal Genome Project (HG002) and Washington University (HG06807) |

Note that full information on the approval of the study protocol must also be provided in the manuscript.

# Field-specific reporting

Please select the one below that is the best fit for your research. If you are not sure, read the appropriate sections before making your selection.

☒ Life sciences      ☐ Behavioural & social sciences      ☐ Ecological, evolutionary & environmental sciences

For a reference copy of the document with all sections, see nature.com/documents/nr-reporting-summary-flat.pdf

# Life sciences study design

All studies must disclose on these points even when the disclosure is negative.

| | |
|---|---|
| Sample size | No sample size estimate was necessary. This is a high-quality reference genome sequencing project, focused on one individual as representative of the species. The variation that exist within humans is not expected to dramatically change the outcome of the findings. |
| Data exclusions | No data were excluded from the analyses |
| Replication | Most algorithms applied in this study were repeated two or more times on the same sample |
| Randomization | Randomization is not applicable to this genome assembly study, as there is a sample size of one individual. |
| Blinding | Blinding is not applicable, as there are not any quantifications that are subjective. |

# Reporting for specific materials, systems and methods

We require information from authors about some types of materials, experimental systems and methods used in many studies. Here, indicate whether each material, system or method listed is relevant to your study. If you are not sure if a list item applies to your research, read the appropriate section before selecting a response.

## Materials & experimental systems

| n/a | Involved in the study |
|---|---|
| ☒ | ☐ Antibodies |
| ☐ | ☒ Eukaryotic cell lines |
| ☒ | ☐ Palaeontology and archaeology |
| ☒ | ☐ Animals and other organisms |
| ☒ | ☐ Clinical data |
| ☒ | ☐ Dual use research of concern |

## Methods

| n/a | Involved in the study |
|---|---|
| ☒ | ☐ ChIP-seq |
| ☒ | ☐ Flow cytometry |
| ☒ | ☐ MRI-based neuroimaging |

# Eukaryotic cell lines

Policy information about cell lines and Sex and Gender in Research

| | |
|---|---|
| Cell line source(s) | GM24385 (RRID:CVCL_1C78) EBV-immortalized lymphoblastoid cell line (LCL) of HG002 |
| Authentication | Genome sequence authentication |

Mycoplasma contamination | None detected in the genome data

Commonly misidentified lines
(See ICLAC register)

*Name any commonly misidentified cell lines used in the study and provide a rationale for their use.*

