## [Peer Review File · Nature]

Manuscript Title: Semi-automated assembly of high-quality diploid human reference genomes

Reviewer Comments & Author Rebuttals

Reviewer Reports on the Initial Version:

Referee expertise:

Referee #1: genome informatics

Referee #2: genome assembly

Referee #3: human genomics

Referees' comments:

Referee #1 (Remarks to the Author):

This article is the first report from the Human Pangenome Reference Consortium (HPRC), and it provides guidelines and results towards generating a better representation of the human genome. This work is much needed since the current human reference (GRCh38) has important limitations. Firstly, GRCh38 is built from only few individuals, and secondly, it is represented as a linear haploid sequence despite the fact that humans are diploid. These issues are not easily resolved by further patches of GRCh38. To this end, HPRC aims to construct a new representation of the human reference genome from scratch, by leveraging the latest sequencing technologies and bioinformatics tools. Through the use of these methods, new diploid human genomes will be assembled from a diverse selection of individuals. The resulting genome sequences will then be combined into a pan-genome graph, which provides a much better representation of the human genome and its variability as compared to a linear reference. Once such a pan-genome reference is available, along with efficient analysis tools, this will likely lead to new discoveries about our genetic code and higher success of human sequencing projects. The work described in this manuscript is of high importance since it is a first step towards the construction of the human pan-genome reference.

The manuscript is structured into two main parts. In the first part, the authors evaluate strategies for diploid human genome assembly. This is done through applying a wide range of sequencing technologies and bioinformatics pipelines on the well-characterized sample HG002, as well as samples from both parents of HG002. In the second part of the paper, the authors apply additional methods to further refine the HG002 assembly and construct a near-complete diploid phased human genome. I really enjoyed reading this paper, and in particular the second part, which has high scientific novelty. The fact that a maternal and paternal genome can now be completely separated in a human sample, through the use of the latest sequencing technologies and computational tools is something I find truly fascinating. The manuscript is well written and is easy to follow, at least for readers with expertise in genomics and bioinformatics. However, I also have some specific

comments, questions and suggestions for the authors, outlined in the points below:

Main points:

1. As far as I understand, the DNA for the HG002 individual studied in the paper was obtained from immortalized cell lines and not from blood or other sample sources taken directly from the individual. This makes me wonder whether some of the genomic events (such as SVs) might be found only in the cell line while being absent in the individual's DNA. While not being an expert in this area, I imagine that some genomic artefacts might be introduced during the generation or maintenance of a cell line. In the methods section, it is mentioned that some steps have been taken to minimize this risk. However, I think the authors could elaborate on whether they believe the use of cell line DNA could be a real concern or not. Ideally, additional data analysis could be performed comparing cell line DNA to some other sample taken from the individual HG002, but there might be other analyses or arguments that can be made here. It would also be interesting to know whether HPRC will recommend to use cell lines or other sample types for the continuation of the pan-genome reference project.

2. I'm also curious about the practical utility of diploid (instead of haploid) reference sequences. Clearly this work is interesting from a methodological point of view, and the near-complete phased diploid assembly for sure gives a more accurate representation of the genome. But how can we make use of this diploid information to better understand biology and improve human medicine? In the last paragraph of the discussion, the authors state that the reference pangenome will give "greater accuracy for precision medicine, and a greater understanding of the biology of genomes". I totally agree with this statement. But do we really need diploid genomes to achieve these goals, or would haploid genomes be enough? It would be interesting if the authors could provide some more insights into why diploid assemblies are necessary.

3. The evaluation of different assembly diploid approaches for HG002 is useful for researchers aiming to construct high-quality genomes, not only in humans but also for other species. However, genome assembly is a quickly moving field and the best practices presented here could soon become outdated. For this reason, I think it could be useful to have a dedicated website or resource where the best-practice information is continuously updated. It would then be possible to access the latest information also after the article has been published. I don't know whether such a resource already exist from HPRC, but in that case it could simply be included as a URL or reference from this manuscript.

4. Based on the high coverage of long-read data, I'd be very interested to see an analysis of genetic mosaicism in this HG002 sample. Mosaicism has previously been analyzed in targeted long-read experiments, where for example tandem repeat lengths have been shown to differ between cells. This HG002 dataset should give a great opportunity to characterize mosaicism at a genome-wide scale and at haplotype resolution. If mosaicism indeed is present in HG002, then that would be both biologically interesting and technically relevant, since this is a well-characterized benchmark sample used in many labs around the world. Moreover, I suspect that mosaicism could be a challenge for the assembly tools and maybe that we can see such effects in the results. It is mentioned on page 15 that homopolymers and tandem repeats can be challenging (lines 518-521). Could this to some

extent be due to mosaic variation taking place in those regions?

5. Automated analysis is mentioned several times throughout the manuscript, including in the title. However, I'm not completely sure what the authors mean by 'automated' in this context. Will the entire analysis be run simply by pressing a button or are additional interventions required? At least for the final diploid sequences (HPRC-HG002.mat.v1.0 and HPRC-HG002.pat.v1.0), these seem to have been curated manually. I think it would be good if the authors can clarify the level of automation for the different analyses. Such information could for example be included in Table 1, or in the supplementary material.

Minor comments:

1. In the abstract (lines 85-86) it is mentioned that 1/4th of protein coding genes have synonymous amino acid substitutions between haplotypes. Synonymous substitutions are mentioned also in the legend of figure 4b. But I didn't find any mentions of synonymous substitutions in the results text where I expected to read more about this analysis. Also, instead of synonymous substitutions wouldn't it be more interesting to mention the fraction of genes having different coding sequence between the two haplotypes (i.e. non-synonymous amino acid changes)?

2. The results on missing genes are interesting (page 18). But can we be sure that these missing genes are real, or could some of them correspond to artefacts or mis-annotations in gene databases? Also, the heading for this section could be changed to "Missing genes among samples and haplotypes" or similar, just in case some genes are missing only from the cell line while being present in the individual (as mentioned in the first point).

3. In abstract, line 85. The authors write that most gaps are "within +1% of CHM13's length". I guess this means that the gap sizes were compared the corresponding regions from the complete CMH13 assembly? Maybe this can be rephrased as it was a little unclear

4. I think the 'Team' column can be removed from Table 1. This information doesn't feel so relevant for a reader and could instead be placed in the supplementary material

5. The word "bakeoff" is first used on page 15, line 502. I'm not so familiar with this term and it could either be replaced or explained when first mentioned

6. Page 16, line 561. It is mentioned here that most chromosomes are "nearly complete". This feels a little vague and more exact information could be useful

7. On page 19, lines 611-612. Here the authors say that they detect a remarkably high amount of autosome heterozygosity. What does this mean? Is autosome heterozygosity in HG002 higher than what would be expected based on SNV analyses of the parents?

8. On page 19, lines 647-649 it is said that more copy number variation is detected in genes associated with primate-specific traits. This sounds interesting. Do you have any explanation or

hypothesis for this observation?

9. In Extended Data Figure 1: Contamination is detected only in some assemblies but not others. Do you have any idea why?

10. In Extended Data Figure 10: Assembled haplotypes are shown over the MHC locus in panel C. I think this is a very nice result and visualization. Don't know if there is enough space, but maybe you could consider moving this panel to one of the main figures.

Adam Ameer

Referee #2 (Remarks to the Author):

This paper by Jarvis and Fomenti, et al., describes the results of an assembly bake-off aimed at figuring out what approaches work best for assembling diploid human genomes. They follow this with further development of an assembly workflow using the best practices distilled from the bakeoff. The purported goal is to establish a recipe for assembly of “near-complete” diploid (both parental haplotypes assembled – ideally phased completely) that can be done at scale. The estimated number of genomes to create a new human pangenome reference that would capture maximum diversity is 450 (900 haplotype assemblies), thus the need to do this “mostly automatically” (p.16). This is an ambitious but I believe achievable goal, and this paper lays out a path to get there. The exercise that was carried out was important, it was done well, and the paper for the most part reflects this. The main conclusions from the first part of the study (the bake-off) were acted upon and found useful for generating a new improved workflow. The analysis of the resulting diploid assembly led to useful insights into human genetic variation and specific areas for future diploid genome assembly improvement. The data presented are quite thorough and key methodologies/software are open and should facilitate reproducibility. In fact, one of the very goals of the exercise is to establish a reproducible workflow. Other aspects of the analyses done are generally well-documented as well, although I've made comments where some further details could be provided. The paper, including the abstract, discussion and conclusions are clear and appropriate. I'd like to complement the authors on the “A look towards the future” section at the end. I appreciate the discussion of what still needs to be done and suggestions on how.

HOWEVER, I would like to air my biggest criticism of the paper immediately right here:

The title of the paper, “Automated assembly of high-quality diploid human reference genomes” suggests that the authors have themselves developed a fully automatic pipeline to produce high-quality diploid human reference genomes. In fact, what they have done is solicit automated (i.e. not curated) assemblies for the bake-off. However, what seems to be the main result of the project (a newly developed “HPRC Trio pipeline v1.0”) involves curation at at least three points: “Conflicts between the HiFi contigs and Bionano optical maps were manually evaluated,” they “performed manual curation using Hi-C contact maps,” and after gap-filling and decontamination another they do another round of manual curation, for a total of at least 105 breaks or joins or other fixes and more for the paternal assembly. By the way, Extended Data 10b is missing the manual step in

resolving the Bionano SOLVE conflicts (the two curation steps are shown so it would be only fair to show the Bionano curation step). It is also not clear to what extent the decontamination requires manual review. While what the authors report is impressive and very thorough, in order to scale to 450 genomes, this process would benefit from being made at least a little more automated. REGARDLESS, I suggest that the title of the paper be changed to reflect the current semi-automated nature of reference genome production. It is obvious from the bake-off that automated methods are becoming much better than before, but that they don't yet meet the lofty standards that the HPRC is striving for -- thus the need to do manual curation.

Another general comment/gripe I have is about the use of the term "metric" throughout the text in phrases like "surpass the quality metrics..." or "yield the highest quality metrics..." A metric is a standard of measurement -- in this case, different measurements that reflect the quality of an assembly. Please rephrase these statements in terms of performing better or scoring higher on these metrics rather than yielding the metrics themselves.

Now begins a long list of specific comments that I would like to see addressed in order to correct errors and improve clarity.

SPECIFIC COMMENTS

Pg3.par1. Suggested change: "...GRCh38 was advanced..." change to "Over the years, the primary assembly has improved from having over 150,000 gaps to just 995 in the current GRCh38 assembly."

Pg3.par4 "...new biological discoveries" mentioned at the end of the intro. Perhaps you could include a hint that they have to do with uncovering more genetic variation. The conclusion spells it out more clearly, but this sentence is vague.

Pg4.par2. Data used. Perhaps mention, either in the text or directly on Table S1 what is stated on the github: "We encourage assembly groups to use as much of the data from the HG002 freeze as possible to get the best assembly they can. However, as no two groups are likely to use exactly the same subset of data, making comparison more difficult, and the size and variety of the HG002 freeze is not representative of what is likely to be available in future freezes, we recommend that assembly groups also run their pipeline on the following set of 4 downsampled datasets from the HG002 (NA24385) human cell line:"

Pg7.par1. reference to Supplementary Table 2f. You should probably include the haploid scaffolded assemblies, too, in which case the range should be adjusted to 132-242 Mb.

Pg8.par2. "Missed joins" and "misjoins". Please take the definitions that are in the legend of Supplementary Figure 1 and place them in the main text, because my immediate naïve question was "What are contigs with missed joins? Don't almost all contigs have missed joins, meaning they have not been extended T2T?" I think this is confusing. This is really referring to lack of scaffolding or scaffolding errors. What about unlocalized contigs – do these get classified as missed joins, too? The term misjoined contigs is hard to separate from the idea of chimeric contigs instead of the intended definition of "contigs that should not have been brought together" Moreover, how are misjoined

contigs different than inversions or translocations?

Pg8.par3. Base call accuracy. This section should be renamed something like “Consensus base accuracy.” The term base call accuracy should be reserved for either the self-reported QV values given by the sequencing platform base caller or alignments of base calls to a truth sequence. The assemblers themselves increase the consensus base accuracy, not base call accuracy. Change “generated the highest base call accuracy” to “achieved the highest consensus base accuracy.” I prefer the word “achieve”, not “generate”. Likewise for other references. The last sentence is OK, where it refers to “high base calling accuracy of HiFi reads” even though it is a consensus-based base caller.

Pg8.par4. Variant benchmarking. I must complain about the use of dipcall to call variants on haploid assemblies (those assemblies that are haploid on purpose). This is not the recommended way to run dipcall, which is designed for diploid assemblies. “Dipcall is a reference-based variant calling pipeline for a pair of phased haplotype assemblies” according to the github README. Obviously, the authors know this, but the results for haploid assemblies should be presented with this caveat or removed entirely.

Pg12.Figure2. I was going to say that this analysis and figure were unnecessary, but mainly because in a printed pdf, it was hard to read. In the ppt, the figures are much easier to zoom and make sense of. Keep this in mind for the final version of the paper. As long as on-line access to hi-res figures is guaranteed, then no problem. I would suggest a re-labeling of the PCAs in Fig2.c-d with simply the assembly numbers, while maintaining the color codes. The long names and connecting lines actually make it harder to interpret – looks a bit like an RDA plot.

Pg13.par1. It is stated that the titration of child HiFi data for optimal hifiasm assembly is ~130x, while the methods state that 160x was used (genomes deposited at NCBI also state 160x coverage). So, in the end, for which assembly, 130x or 160x, are the stats reported. Also, if 130x is optimal, why was this assembly not further analyzed and deposited in Genbank. And why were the 34x downsampled reads recommended for the bake-off? At what coverage will the ~450 genomes be sequenced? 34x or 130x or 160x? This is an important detail to be discussed in the discussion.

Also, I think that the amount of “ONT-UL reads” referred to in the main text (please label this set properly) used for the HPRC-HG002 reference should be included in the main text or the methods for clarity. Table S1 has a row labeled “Ultra-long GridION data” that refers to 100kb+ reads (15x from minion and 45x from PromethION filtered from over 52x coverage GridION and 658x PromethION, respectively). Please re-label this set of reads as ONT-100kb+ (or similar). The UL protocol yields a mix of reads, not all of which are 100kb+; in fact, some definitions of UL reads refer to a set of reads with N50 \geq 100kb. I would also recommend a statement in the discussion regarding how this amount of ultralong (100kb+) ONT reads can be generated efficiently as I understand 28 PromethION and 106 MinION flowcells were used to obtain this amount of data. Even at the current lowest bulk rate for PromethION flow cells and assuming higher efficiency, this would still cost quite a bit -- fine for producing one reference, but for 450 reference genomes? Same question as for the HiFi reads. How applicable is this approach in general? Of course, the human genome is deserving of such attention and money and time, but casual readers may incorrectly

suppose that the approach being put forth is universally applicable, when in fact, generating over 700x ONT, 130 HiFi, Bionano optical maps, 70x Hi-C, 300x Illumina of each parent, etc. is a costly endeavor, not to mention the expertise of at least 3 curators at three different points during assembly. I don't argue that the human pangenome is not worth it, just that for other projects (of the EBP variety, for example), it is likely impractical.

Pg13.par1,2. Three references to manual evaluation/curation in a paper titled "Automated assembly of..." See my general comments on this. But one additional clarification is needed: for the contamination screening, is the process automated or are possible contaminants reviewed manually and is removal done automatically or manually?

Pg14.Fig.3. last line of legend: "the remainder of" seems to be a copy-paste error.

Pg15.par1. "These two de novo assemblies had the highest quality of all metrics compared to..." Change to something like "exhibited the highest quality across most metrics" or "exhibited the highest combined quality across all metrics"

Pg.15 last sentence "k-mere" -> "k-mer"

Pg17.Fig.4a. I could not find the methods for alignment and coloring by haplotype for this figure in the main text, legend or methods. Are the paternal assembly contigs on top and maternal assembly contigs on bottom? What determines the color value in the heatmap? Despite there being nearly complete haplotype separation, some apparently purple alignment blocks can be seen – is the hue based on parental k-mers somehow? Also, perhaps different colors or symbols could be used to distinguish gaps from unaligned divergent sequence? It is too hard to flip back and forth between 4a and 4b to see where scaffold gaps are located.

Pg17.Fig.4b and Table S9. What is meant by "translocation" in this context? My understanding of a translocation is that of a chromosomal break and fusion event, causing a large chromosomal abnormality. This is clearly not the definition being used here. It would appear that no chromosome arms are being exchanged and that these SVs are bits of sequence that are homologous, yet located in different parts of the chromosome, like mobile elements. I would just use the term "rearrangements." I know SyRI classifies some rearrangements as "translocations" but I think the term is overused in SV detection methods and should be reserved for large events seen in things like cancer.

Pg19.par1. "This difference could be due to [the] marmoset assembly using higher error [rate] CLR Pacbio..." please add "the" and "rate"

Pg19.par2. The 15Mb inversion on Chr9 apparently has no clear Strand-seq alignment orientation evidence to support or reject the inversion according to Extended Data Fig5a and b. It might be nice to see such examples. Could it be a SALSA2 or curation error, given that in the maternal assembly, this region seems to be flanked by gaps (Fig.4b) and could just be a contig orientation error? I wouldn't have noticed if attention hadn't been called to it.

Pg19.par2. "...6,397 (50%) were synonymous, changing the amino acid sequence..." I think the words synonymous and non-synonymous were switched in this sentence. 46% are (non-?)synonymous. What are the other 4%? Are they nonsense? Are small indels included in the SNV calls that would cause frameshifts? Please make the numbers add to 100% and make sure they make sense.

Pg19.par3. "yield the highest quality metrics"? I think you mean something like perform the best on the metrics we tested. See previous comment.

"could mispartition reads" => "has the potential to mispartition" or "is prone to..."

Pg32.Extended Data Figure 11. Inconsistencies between the legend and the figure. Panel d says maternal in the figure and paternal in the legend. e-f legend says q-arm of the assembly. Change "assembly" to Chr12 if in fact it is Chr12 (maternal? Paternal?) The last comment of the legend says that the q-arm telomere is missing from Chr12, while it would appear that it is really the p-arm telomere that is missing. Please review and correct the all aspects of this figure.

Pg34.Supplementary Fig. 2. States near full 130x coverage. 160x is not shown. Why? What is full coverage? 133x? 160x? It states that the "The 130x-300x v0.14 assembly was used for the final HPRC-HG002 assembly of this study." Why does the assembly submitted to NCBI say 160x? Also, the methods state "Maternal and paternal contigs were generated from 160X coverage of the remaining HiFi reads using hifiasm v0.14.1 in trio mode" on page 47. Sorry, this is a repeat of my previous comment above, referring to this figure.

Pg35.Supplementary Fig. 3. That's an awful lot of sequencing (28 PromethION FCs and 106 MinION flowcells to fill 15 gaps that take up less than 1 page in a Word doc! Just saying. Maybe in the "A look towards the future" section a small comment could be made about this. Perhaps some targeted sequencing (cas9 enrichment) on the ONT platform could help with the gapfilling effort and bring the cost down...

Methods

Pg40. ONT reads. What are "readsShasta"? Is this the output of a Shasta preprocessing of ONT data? I don't see anything in the shasta documentation about this. Why is Shasta needed to filter for 100kb+ reads? Potential typo?

Pg51. Annotations. How many RefSeq transcripts were used? What was the exact query used in Entrez? How were isoforms handled? Was only one transcript per gene chosen or were they clustered on GRCh38 first? If not, how were Unaligned genes calculated: was one missing transcript criteria for being unaligned or all isoforms missing? For calculating collapsed, how is it known the transcripts aligning to the same locus come from the same or different genes?

Referee #3 (Remarks to the Author):

Summary:

The work presented is a progress report of the Human Pangenome Reference Consortium, which aims to sequence ~450 humans, to assemble both the maternal and paternal genomes of those individuals and finally construct a pangenome graph that incorporates all those assemblies. This first step progress report describes computational methods for constructing the assembly of a single individual, HG002. Single individual assemblies have been described before, particularly for CHM13. The difference between this assembly and the CHM13 one is that CHM13 was essentially a haploid genome and its assembly required extensive manual curation. The assembly presented here is mostly able to resolve the two haplotypes. The authors argue that their work requires less manual curation, although considerable manual curation seems to be required and a fairly large number of gaps still remain, although much fewer than in GRCh38.

The work of the Human Pangenome Reference Consortium of no doubt has the potential to be of immense value when finished. However, it is not clear to me what the value of the current report is and the report also suggests that the study design for the final project could be improved upon.

Major Comments:

1) A fairly convincing argument is made that the reference presented here accurately represents the two haplotypes of the individual being studied. However, no results are given where this assembly is used and no recommendation is given to the reader on how to use it. Does this mean that the authors do not recommend the usage of the assembly presented? As far as I can tell the assembly is not being released.

2) The assembly is performed on an extensively studied individual, it is understandable that the authors would like to test their methods on this individual as a curated set of true variants exist for the individual. This however makes the study design susceptible to overfitting. First much of the same data was used in the construction of the truth set, but more importantly the algorithms have been trained to fit biases in these dataset. A better study design would have been to use separate test and training sets, i.e. the algorithms would be trained on one set of data and then evaluated on another set. At minimum it is clear that the number of assembly errors estimated using this approach is a downward biased estimate of the numbers of errors one would expect to get for additional assemblies.

3) The study design is to evaluate a number of algorithms, compare that to an assumed truth and then the results for this comparison is used to develop a final super-algorithm by correcting errors made by other algorithms. This even further biases the results.

4) The HG002 benchmarks mainly target relatively easy parts of the genome and mostly those that are already in GRCh38. However, the main problems in assembly are in those regions not reached by GRCh38, giving a further downward bias of the estimated error of the method.

5) The authors don't explicitly state how they plan to use this assembly or others in the construction of the pangenome, but I infer from the text that the plan is to construct an assembly of each one of the individuals and then merge the assemblies into a graph. I am very much concerned about this study design if this is the case. It would be prudent of the authors to reconsider this design given the

results of this first phase study, in particular as the results suggest that this is not likely to be the best possible design. In particular the proposed focus in future work on improving diploid assembly seems to be misplaced faith on part of the authors and a poor design decision.

a) The problem of merging assemblies into a graph is not well solved. In particular it is clear that graphs allow for multiple equivalent representations of the same sets of variants and that different merging strategies could lead to different graphs. This means that once they have constructed the assembly of all these individuals they still have to solve the problem of merging. The merging process is likely to propagate errors from each of the individual assemblies to the combined graph assembly and could also introduce further errors into the final graph assembly.

b) Although the number of assembly errors appear to be much smaller than in GRCh38, the merging of 900 assemblies (at least naively) requires that the errors per assembly be 900 times lower so that the errors in the combined assembly are fewer than GRCh38, a threshold not reached in the current study.

c) The results presented here show that the best algorithm is to use sequence reads from parents along with the sequence read of the individual. In addition they show that algorithms can benefit from a reference, although caution needs to be taken in which inferences to draw from the reference. In light of this, why would the authors not consider a joint assembly of the multiple genomes for their future work?

6) The authors of Giraffe (<https://www.science.org/doi/full/10.1126/science.abg8871>) found that there was a saturation in alignment accuracy when combining more than 64 haplotypes in an assembly graph. Do the authors have evidence to refute these claims?

7) Whole genome assembly algorithms have been in development for over 20 years. They have advanced with different sequencing methods, which have and continue to evolve rapidly. Do the authors have reason to believe that this will not continue to be the case during the course of the Human Pangenome Project?

8) The authors seem to implicitly make the assumption that all variants in the proposed final graph will be from the individual assemblies. Given the large number of whole genome sequenced individuals it seems a waste to not use some of the information already present in those.

9) Most of the paper focuses on comparing algorithms that went into the bakeoff competition. I understand that this was important work but it is not clear to me if this is of general interest to the readers of the journal. Also it is not clear to me why the results for all the algorithms need to be presented as some are clearly inferior to others.

10) The authors should rephrase their discussion of the mitochondria assembly and perhaps also reconsider their modeling assumptions. The mitochondria is not expected to have a single haplotype. Individual cells may carry multiple mitochondria, which may differ from each other.

Minor comments:

1) “Our findings serve as a foundation for assembling near-complete diploid human genomes at the scale required for constructing a human pangenome reference that captures all genetic variation from single nucleotides to large structural rearrangements”. - A pangenome that captures all genetic variation is likely an unattainable goal. The authors state later that the work is to be the foundation for a graph with most variants > 1% frequency, a more reasonable goal.

2) “Its current build, GRCh38, reflects two decades of additional effort by the Genome Reference Consortium and others to correct the initial draft.” - A lot of work on creating alternate assemblies has contributed to GRCh38.

3) “and does not adequately capture the full spectrum of humankind genomic variation” - capturing “the full spectrum” of humankind genomic variation is likely an unattainable goal, considering that de novo mutations occur in most meiosis and mitosis.

4) “While resequencing efforts using less expensive short reads have contributed to revealing more single-nucleotide variation, structural variation is not fully captured” - Neither SNPs nor SVs are fully captured. Long read efforts have also contributed extensively to the characterization of SVs.

5) Figure 1: The color coding (blue, light blue and turquoise) is confusing. A more diverse palette of color would be helpful. What does “more complete” mean in the caption?

6) I think it would be useful to have somewhere in the paper a quick list of what data were available/used to perform the assemblies and more importantly at what coverage. Additionally, I would have liked to see a quick reminder of what distinguishes the different types of phased assemblies such as in <http://lh3.github.io/2021/04/17/concepts-in-phased-assemblies>.

7) The HPRC Trio pipeline as illustrated in Figure 10b is very useful but also shows that a lot of different sequencing technologies (most likely at very high coverage) are required to build such a type of assembly. I would have appreciated to see a summary of what each data type brings to the final assembly such as “UL ONT enables (mostly) gap filling”. I think this would be extremely useful for many labs who cannot afford all these sequencing technologies but instead, would need to prioritize some type of sequencing technology that fits the needs of their project.

8) The authors hint at the fact that the variant calls which differ between the assembly and the Genome In a Bottle truth set are most likely correct in the assembly but incorrect in the GIAB truth set. This is a major finding in my opinion given that many methods nowadays are optimizing their results or creating models based on this truth set. A list of these differences should be provided and eventually a new truth set without these variants should be released.

Nitpicking:

Can “latter” be used for the last one of three?

Author Rebuttals to Initial Comments:

Referee #1: genome informatics

Reviewer # 1 comment: This article is the first report from the Human Pangenome Reference Consortium (HPRC), and it provides guidelines and results towards generating a better representation of the human genome. This work is much needed since the current human reference (GRCh38) has important limitations. Firstly, GRCh38 is built from only few individuals, and secondly, it is represented as a linear haploid sequence despite the fact that humans are diploid. These issues are not easily resolved by further patches of GRCh38. To this end, HPRC aims to construct a new representation of the human reference genome from scratch, by leveraging the latest sequencing technologies and bioinformatics tools. Through the use of these methods, new diploid human genomes will be assembled from a diverse selection of individuals. The resulting genome sequences will then be combined into a pan-genome graph, which provides a much better representation of the human genome and its variability as compared to a linear reference. Once such a pan-genome reference is available, along with efficient analysis tools, this will likely lead to new discoveries about our genetic code and higher success of human sequencing projects. The work described in this manuscript is of high importance since it is a first step towards the construction of the human pan-genome reference.

The manuscript is structured into two main parts. In the first part, the authors evaluate strategies for diploid human genome assembly. This is done through applying a wide range of sequencing technologies and bioinformatics pipelines on the well-characterized sample HG002, as well as samples from both parents of HG002. In the second part of the paper, the authors apply additional methods to further refine the HG002 assembly and construct a near-complete diploid phased human genome. I really enjoyed reading this paper, and in particular the second part, which has high scientific novelty. The fact that a maternal and paternal genome can now be completely separated in a human sample, through the use of the latest sequencing technologies and computational tools is something I find truly fascinating. The manuscript is well written and is easy to follow, at least for readers with expertise in genomics and bioinformatics. However, I also have some specific comments, questions and suggestions for the authors, outlined in the points below:

Response: The reviewer gave an accurate summary of our paper. We are glad to see that our main message got across, and to see the reviewer's enthusiasm about the study and overall project. We kept this main message in the revised paper.

Reviewer # 1 comment:

Main points:

1. As far as I understand, the DNA for the HG002 individual studied in the paper was obtained from immortalized cell lines and not from blood or other sample sources taken directly from the individual. This makes me wonder whether some of the genomic events (such as SVs) might be found only in the cell line while being absent in the individual's DNA. While not being an expert in this area, I imagine that some genomic artefacts might be introduced during the generation or maintenance of a cell line. In the methods section, it is mentioned that some steps have been taken to minimize this risk. However, I think the authors could elaborate on whether they believe the use of cell line DNA could be a real concern or not. Ideally, additional data analysis could be performed comparing cell line DNA to some other sample taken from the individual HG002, but there might be other analyses or arguments that can be made here. It would also be interesting to know whether HPRC will recommend to use cell lines or other sample types for the continuation of the pan-genome reference project.

Response: The reviewer is correct, in that we have been using lymphoblastoid cell lines (LCLs) for the HPRC genomes. The reasons are because they make it easier to: isolate high-quality DNA; to return to the same sample for both short- and long-term needs (such as upgrading the genome without having to go back to the original source person); and to be used for functional gene experiments in the person's specific genetic background. However, like the reviewer, we were concerned about the possibility of cell culture induced genome rearrangements. For this reason, we have been using low passage numbers (between 4-10) for this and all current cell culture samples for the HPRC. We have also been karyotyping all cell lines to check for any rearrangements before genome sequencing. But we had not done this for the HG002 sample used in this study. Thus, in response to the reviewer's concern, we performed karyotype spreads of HG002 LCLs with X and Y chromosome probes, and found with 58 randomly selected cells an expected modal distribution of 46+XY chromosomes (**new Supplementary Fig. 1**). There were some cells with a missing chromosome that appeared to be artifacts of imaging and 3 cells with tetraploid arrangements, but none with major deletions or rearrangements. We now include these findings in the revised results and methods.

In a separate sample, we compared genome contigs generated using hifiasm on HiFi reads from cell lines versus blood of the same individual. We found that the cell-based and blood-based variants are highly similar (99.8% in SNPs). The remaining 0.2% could be the result of mosaicism and/or consensus assembly sequence errors. We also searched for common and different variants that intersected with the ClinVar database to find if there exist any Pathogenic variants. We could not find anything pathogenic. We also performed G-banded karyotyping of HG06807 LCLs and did not find signs of chromosomal rearrangements. All cells analyzed showed a normal karyotype of 46,XX. These findings suggest that the LCL immortalization process and low passaging have not introduced major changes in the genome. We have included these findings as a **Supplementary Note 4**. A more detailed analysis will follow in a separate publication.

Reviewer # 1 comment: 2. I'm also curious about the practical utility of diploid (instead of haploid) reference sequences. Clearly this work is interesting from a methodological point of view, and the near-complete phased diploid assembly for sure gives a more accurate representation of the genome. But how can we make use of this diploid information to better understand biology and improve human medicine? In the last paragraph of the discussion, the authors state that the reference pangenome

will give "greater accuracy for precision medicine, and a greater understanding of the biology of genomes". I totally agree with this statement. But do we really need diploid genomes to achieve these goals, or would haploid genomes be enough? It would be interesting if the authors could provide some more insights into why diploid assemblies are necessary.

Response: We and others find that diploid assemblies are necessary to prevent assembly errors. It is difficult to make a haploid assembly that merges differences between haplotypes without making errors in the form of gaps, false haplotype duplications, sequence consensus errors, or structural variants. One also needs a diploid assembly to separately assemble the X and Y sex chromosomes. We had mentioned this in the introduction, but now expand on it. For precision medicine and new biological discoveries, there are a number of genes with maternal or paternal imprinting that lead to haplotype-specific diseases and biological functions separate from the sex chromosomes. Also allele combinations that co-segregate on the same haplotype will have different functional consequences when they are split across haplotypes. We have added this additional insight in the introduction and discussion.

Reviewer # 1 comment: 3. The evaluation of different assembly diploid approaches for HG002 is useful for researchers aiming to construct high-quality genomes, not only in humans but also for other species. However, genome assembly is a quickly moving field and the best practices presented here could soon become outdated. For this reason, I think it could be useful to have a dedicated website or resource where the best-practice information is continuously updated. It would then be possible to access the latest information also after the article has been published. I don't know whether such a resource already exist from HPRC, but in that case it could simply be included as a URL or reference from this manuscript.

Response: This is a good idea. We now generated a wiki in the HPRC production Github repository (https://github.com/human-pangenomics/hpp_production_workflows/wiki/Assembly-Best-Practices) where we list the pipeline used for HG002 and best practices. We will use this site to update our best practices going forward.

Reviewer # 1 comment: 4. Based on the high coverage of long-read data, I'd be very interested to see an analysis of genetic mosaicism in this HG002 sample. Mosaicism has previously been analyzed in targeted long-read experiments, where for example tandem repeat lengths have been shown to differ between cells. This HG002 dataset should give a great opportunity to characterize mosaicism at a genome-wide scale and at haplotype resolution. If mosaicism indeed is present in HG002, then that would be both biologically interesting and technically relevant, since this is a well-characterized benchmark sample used in many labs around the world. Moreover, I suspect that mosaicism could be a challenge for the assembly tools and maybe that we can see such effects in the results. It is mentioned on page 15 that homopolymers and tandem repeats can be challenging (lines 518-521). Could this to some extent be due to mosaic variation taking place in those regions?

Response: Following up on the reviewer's request, we performed a single nucleotide mosaicism analysis. We aligned the Illumina reads against the HG002 maternal and paternal haplotypes of the final diploid reference and called mosaic variants. We found an average mosaic level (=of 0.0466% and 0.0468% for each haplotype, respectively, that was 10-fold less than what we observed in a marmoset in a previous study using similar methods (Yang et al 2021 *Nature*). Interestingly, mosaicism was higher on the smaller chromosomes (Chr13-22; **new Supplementary Fig. 8**), indicative of possible greater mutational load on them. We did not find a change in % mosaic variation when repeat regions were removed.

Supplementary Fig. 8. Mosaicism in HG002 haplotypes. a, Graphed are the rate of minor alleles in the raw reads found for each haplotype. Each dot represents the value for each autosome. Chromosomes with mosaicism higher than the 95% confidence interval are numbered. **b,** Mosaicism relative to chromosome size, with chromosomes ordered from largest to smallest. Note that a higher prevalence of mosaicism among the smaller chromosomes.

Reviewer # 1 comment: 5. Automated analysis is mentioned several times throughout the manuscript, including in the title. However, I'm not completely sure what the authors mean by 'automated' in this context. Will the entire analysis be run simply by pressing a button or are additional interventions required? At least for the final diploid sequences (HPRC-HG002.mat.v1.0 and HPRC-HG002.pat.v1.0), these seem to have been curated manually. I think it would be good if the authors can clarify the level of automation for the different analyses. Such information could for example be included in Table 1, or in the supplementary material.

Response: The reviewer makes a good point. Based on a similar comment by reviewer #2, we replaced "automated" with "semi-automated" in the title. We assume the reviewer is referring to "automated assembly", not "automated analysis" of the assembly? Just in case, yes we are also building automated analysis pipelines for this and other consortiums, which we now cite more clearly as automated analyses pipelines for genomes broadly (e.g. Merqury). But in this study, our goal is to generate as complete and accurate assembly as possible using an automated approach, followed by cleaning up with as minimal manual curation as possible. Yes, our goal is to develop a push button approach as much as possible. We have most of the pipeline automated with multiple push button steps now set up in the Galaxy Server (<https://assembly.usegalaxy.eu/>). The entire pipeline is diagrammed in **Extended Data Fig. 10b**. Following the reviewer's request, we more clearly state this goal in the revised paper, and include each manual curation step in the figure. We felt Table 1 wasn't appropriate for this information, since the table was not dedicated to the final diploid assembly, but the bakeoff assemblies.

Reviewer # 1 comment:

Minor comments:

1. In the abstract (lines 85-86) it is mentioned that 1/4th of protein coding genes have synonymous amino acid substitutions between haplotypes. Synonymous substitutions are mentioned also in the legend of figure 4b. But I didn't find any mentions of synonymous substitutions in the results text where I expected to read more about this analysis. Also, instead of synonymous substitutions wouldn't it be more interesting to mention the fraction of genes having different coding sequence between the two haploypes (i.e. non-synonymous amino acid changes)?

Response: We present additional results of the synonymous and non-synonymous differences between haplotypes. Genes with only non-synonymous haplotype differences were enriched for metabolism, smell, taste, and HSV1 viral infection functions (**new Supplementary Table 12**). Yes, we meant to focus more on non-synonymous changes, not synonymous, which was a typo.

Reviewer # 1 comment: 2. The results on missing genes are interesting (page 18). But can we be sure that these missing genes are real, or could some of them correspond to artefacts or mis-annotations in gene databases? Also, the heading for this section could be changed to “Missing genes among samples and haplotypes” or similar, just in case some genes are missing only from the cell line while being present in the individual (as mentioned in the first point).

Response: The assessment of gene content was done by aligning known RefSeq transcripts (with accession prefixes NM_ or NR_) for human genes to all assemblies. This set of transcripts is actively maintained by the NCBI RefSeq group based on experimental data. Over 97% of RefSeq known transcripts have been curated and are in status “REVIEWED” or “VALIDATED”. To address the reviewer’s concern, the level of review that a transcript has undergone has been added to **Supplemental Table 8** for all genes that are missing from at least one of the four reference assemblies. Missing genes were enriched for status “PROVISIONAL” due to microRNAs, which were originally predicted by miRbase but have yet to be validated. Excluding miRNAs, nearly 90% of the transcripts not found in one of the four reference assemblies have been curated and are therefore not artefacts. We added these additional analyses in a **Supplementary Note 3**. This review made us realize that 13 of the 119 genes missing were really different splice variants that were missing. Thus we updated the number of totally missing genes among one or more reference assemblies as 106, which did not change the major conclusions about the categories and combinations of the missing genes.

We prefer to not replace “individual” with “sample” in the title of this section “**Missing genes among individuals and haplotypes**”, because we did not assess different samples of HG002, but different haplotypes of one individual and of different individuals (HG002, CHM13, and GRCh38). By putting ‘samples’ in the title, we would be making the claim that we know some of the missing gene differences are due to different mosaics between samples of the same individual, which we thus far we do not have evidence for.

Reviewer # 1 comment: 3. In abstract, line 85. The authors write that most gaps are “within +1% of CHM13’s length”. I guess this means that the gap sizes were compared the corresponding regions from the complete CMH13 assembly? Maybe this can be rephrased as it was a little unclear

Response: Thank you for noting the unclear sentence. This value was referring to the relative sizes of the chromosomes, not the gaps. We have revised the wording to state that most assembled chromosomes of HG002 were within $\pm 1\%$ of CHM13's length.

Reviewer # 1 comment: 4. I think the 'Team' column can be removed from Table 1. This information doesn't feel so relevant for a reader and could instead be placed in the supplementary material

Response: Our primary purpose of including the team name was to give more specific credit to each group that generated the bakeoff assemblies, and serve as a contact source for them. We moved this column to the end of Table 1, so it is not the first information readers see. If this change still seems not relevant to the reviewer or editor, we can remove it. We do have more details in the associated supplementary table.

Reviewer # 1 comment: 5. The word "bakeoff" is first used on page 15, line 502. I'm not so familiar with this term and it could either be replaced or explained when first mentioned

Response: The meaning of 'bakeoff' is a contest, derived from baking contests, but in this case an assemblathon competition. We have now infer the meaning of bakeoff on first use in the paper.

Reviewer # 1 comment: 6. Page 16, line 561. It is mentioned here that most chromosomes are "nearly complete". This feels a little vague and more exact information could be useful

Response: We now mention the numbers, which is that most chromosomes (32 of 46) were 98.0-99.9% complete relative to CHM13.

Reviewer # 1 comment: 7. On page 19, lines 611-612. Here the authors say that they detect a remarkably high amount of autosome heterozygosity. What does this mean? Is autosome heterozygosity in HG002 higher than what would be expected based on SNV analyses of the parents?

Response: These values refer to higher autosome heterozygosity (SNV and SV) between haplotypes than expected based on average values in the prior literature for human genomes (Samuels et al 2016 Genetics). We have revised this sentence to make this point clearer.

Reviewer # 1 comment: 8. On page 19, lines 647-649 it is said that more copy number variation is detected in genes associated with primate-specific traits. This sounds interesting. Do you have any explanation or hypothesis for this observation?

Response: Genes with more copy number differences between haplotypes are likely those that recently evolved and/or have more recombination in terms of gene expansion and contractions. For example, human specific duplications show a slight enrichment for genes associated with neurodevelopment while chimpanzee-specific duplications were preferentially associated with immune response genes (Cheng et al, 2005; Sudmant et al, 2013). We interpret the findings to be that in the ancestral primate lineage, duplications of these genes were selected for specific brain and other primate traits. Such differential haplotype duplication is still occurring within primate lineages such as the TBC1D3 gene family in humans (Vollger et al, Science, 2021). We added this additional explanation to the main text.

Reviewer # 1 comment: 9. In Extended Data Figure 1: Contamination is detected only in some assemblies but not others. Do you have any idea why?

Response: All assemblies had vector contamination. Thus, we assume the reviewer means the E. coli and yeast contamination. There are several reasons why some assemblies did not have them (which essentially removed them): 1) Not matching the GRCh38 reference in the reference-based assemblies; 2) Filtering out scaffolds below a specific length in some of the scaffolded assemblies; and 3) Moved them to the alternative assembly that did not assemble with the primary assembly. We have now added these explanations.

Reviewer # 1 comment: 10. In Extended Data Figure 10: Assembled haplotypes are shown over the MHC locus in panel C. I think this is a very nice result and visualization. Don't know if there is enough space, but maybe you could consider moving this panel to one of the main figures.

Response: We are already over the word and figure limit. But Extended Data Figures are part of the main text in the online version of the manuscript in *Nature*.

Referee #2 genome assembly

Referee #2 comment: This paper by Jarvis and Fomenti, et al., describes the results of an assembly bake-off aimed at figuring out what approaches work best for assembling diploid human genomes. They follow this with further development of an assembly workflow using the best practices distilled from the bakeoff. The purported goal is to establish a recipe for assembly of “near-complete” diploid (both parental haplotypes assembled – ideally phased completely) that can be done at scale. The estimated number of genomes to create a new human pangenome reference that would capture maximum diversity is 450 (900 haplotype assemblies), thus the need to do this “mostly automatically” (p.16). This is an ambitious but I believe achievable goal, and this paper lays out a path to get there. The exercise that was carried out was important, it was done well, and the paper for the most part reflects this.

The main conclusions from the first part of the study (the bake-off) were acted upon and found useful for generating a new improved workflow. The analysis of the resulting diploid assembly led to useful insights into human genetic variation and specific areas for future diploid genome assembly improvement. The data presented are quite thorough and key methodologies/software are open and should facilitate reproducibility. In fact, one of the very goals of the exercise is to establish a reproducible workflow. Other aspects of the analyses done are generally well-documented as well, although I’ve made comments where some further details could be provided. The paper, including the abstract, discussion and conclusions are clear and appropriate. I’d like to complement the authors on the “A look towards the future” section at the end. I appreciate the discussion of what still needs to be done and suggestions on how.

Response: We are happy to see the reviewer’s accurate summary and positive feedback, and we have maintained these features of the paper.

Referee #2 comment: HOWEVER, I would like to air my biggest criticism of the paper immediately right here:

The title of the paper, “Automated assembly of high-quality diploid human reference genomes” suggests that the authors have themselves developed a fully automatic pipeline to produce high-quality diploid human reference genomes. In fact, what they have done is solicit automated (i.e. not curated) assemblies for the bake-off. However, what seems to be the main result of the project (a newly developed “HPRC Trio pipeline v1.0”) involves curation at at least three points: “Conflicts between the HiFi contigs and Bionano optical maps were manually evaluated,” they “performed manual curation using Hi-C contact maps,” and after gap-filling and decontamination another they do another round of manual curation, for a total of at least 105 breaks or joins or other fixes and more for the paternal assembly. By the way, Extended Data 10b is missing the manual step in resolving the Bionano SOLVE conflicts (the two curation steps are shown so it would be only fair to show the Bionano curation step). It is also not clear to what extent the decontamination requires manual review. While what the authors report is impressive and very thorough, in order to scale to 450 genomes, this process would benefit from being made at least a little more automated. REGARDLESS,

I suggest that the title of the paper be changed to reflect the current semi-automated nature of reference genome production. It is obvious from the bake-off that automated methods are becoming much better than before, but that they don't yet meet the lofty standards that the HPRC is striving for -- thus the need to do manual curation.

Response: The author is correct and perhaps we became over enthusiastic in a title that implied a fully automated complete and accurate assembly approach. We replaced "automated" with "semi-automated" in the title. We added the two earlier mid-pipeline curation steps in **Extended Data Fig. 10b**.

Referee #2 comment: Another general comment/gripe I have is about the use of the term "metric" throughout the text in phrases like "surpass the quality metrics..." or "yield the highest quality metrics..." A metric is a standard of measurement -- in this case, different measurements that reflect the quality of an assembly. Please rephrase these statements in terms of performing better or scoring higher on these metrics rather than yielding the metrics themselves.

Response: Thanks for catching this semantic issue on "metrics" versus "quality". We have corrected them throughout the text.

Referee #2 comment: Now begins a long list of specific comments that I would like to see addressed in order to correct errors and improve clarity.

SPECIFIC COMMENTS

Pg3.par1. Suggested change: "...GRCh38 was advanced..." change to "Over the years, the primary assembly has improved from having over 150,000 gaps to just 995 in the current GRCh38 assembly."

Response: Change made

Referee #2 comment: Pg3.par4 "...new biological discoveries" mentioned at the end of the intro. Perhaps you could include a hint that they have to do with uncovering more genetic variation. The conclusion spells it out more clearly, but this sentence is vague.

Response: Hint now included in the introduction.

Referee #2 comment: Pg4.par2. Data used. Perhaps mention, either in the text or directly on Table S1 what is stated on the github: “We encourage assembly groups to use as much of the data from the HG002 freeze as possible to get the best assembly they can. However, as no two groups are likely to use exactly the same subset of data, making comparison more difficult, and the size and variety of the HG002 freeze is not representative of what is likely to be available in future freezes, we recommend that assembly groups also run their pipeline on the following set of 4 downsampled datasets from the HG002 (NA24385) human cell line:”

Response: Following the reviewer’s recommendation, we added a slightly shortened version of the above sentences to the end of the second to last paragraph of discussion; we already had a sentence there that began with a statement of what we encourage assembly groups to do for the future.

Referee #2 comment: Pg7.par1. reference to Supplementary Table 2f. You should probably include the haploid scaffolded assemblies, too, in which case the range should be adjusted to 132-242 Mb.

Response: Good point. We revised the scaffold range to 132-242 Mb.

Referee #2 comment: Pg8.par2. “Missed joins” and “misjoins”. Please take the definitions that are in the legend of Supplementary Figure 1 and place them in the main text, because my immediate naïve question was “What are contigs with missed joins? Don’t almost all contigs have missed joins, meaning they have not been extended T2T?” I think this is confusing. This is really referring to lack of scaffolding or scaffolding errors. What about unlocalized contigs – do these get classified as missed joins, too? The term misjoined contigs is hard to separate from the idea of chimeric contigs instead of the intended definition of “contigs that should not have been brought together” Moreover, how are misjoined contigs different than inversions or translocations?

Response: Contig misjoins and missed joins refer to contigs within scaffolds, not within the contigs themselves. “Missed joins” are contigs that should have been neighbors in a scaffold, but were kept apart, including on different scaffolds. Missed joins are only counted if they could be resolved during curation with the available data, thereby implying that an automated process should have been able to get them right. “Misjoins” are the opposite situation, co-localized contigs within scaffolds that don’t belong together; where one of them is often an erroneous translocation. “Unlocalized” is a sequence found in an assembly that is associated with a specific chromosome but cannot be ordered or oriented on that chromosome with the available data. These and other additional structural errors in some cases appear as erroneous inversions or false duplications. Finally, a “chimeric contig” is a continuous

gapless sequence that includes either an erroneous join without a gap, sequence expansions, or collapses. We added these definitions to the main text and the full description above in the methods.

Referee #2 comment: Pg8.par3. Base call accuracy. This section should be renamed something like “Consensus base accuracy.” The term base call accuracy should be reserved for either the self-reported QV values given by the sequencing platform base caller or alignments of base calls to a truth sequence. The assemblers themselves increase the consensus base accuracy, not base call accuracy. Change “generated the highest base call accuracy” to “achieved the highest consensus base accuracy.” I prefer the word “achieve”, not “generate”. Likewise for other references. The last sentence is OK, where it refers to “high base calling accuracy of HiFi reads” even though it is a consensus-based base caller.

Response: We made each of these changes. Thank you for catching them.

Referee #2 comment: Pg8.par4. Variant benchmarking. I must complain about the use of dipcall to call variants on haploid assemblies (those assemblies that are haploid on purpose). This is not the recommended way to run dipcall, which is designed for diploid assemblies. “Dipcall is a reference-based variant calling pipeline for a pair of phased haplotype assemblies” according to the github README. Obviously, the authors know this, but the results for haploid assemblies should be presented with this caveat or removed entirely.

Response: We chose to analyze haploid assemblies in this way because it enabled us to develop performance metrics for how many variants are captured by them. We agree that this nuance is important for readers to understand, so we have added the following statement to the main text “Since dipcall is designed for diploid genomes, for the haploid assemblies we developed separate performance metrics that ignore genotype errors (when only one haplotype has to match the benchmark variant), which indicates how many variants are accurately identified with haploid assemblies independent of genotype.”

Referee #2 comment: Pg12.Figure2. I was going to say that this analysis and figure were unnecessary, but mainly because in a printed pdf, it was hard to read. In the ppt, the figures are much easier to zoom and make sense of. Keep this in mind for the final version of the paper. As long as on-line access to hi-res figures is guaranteed, then no problem. I would suggest a re-labeling of the PCAs in Fig2.c-d with simply the assembly numbers, while maintaining the color codes. The long names and connecting lines actually make it harder to interpret – looks a bit like an RDA plot.

Response: We will make sure that the paper is published with high-resolution versions of the figures, as the reviewer had seen during the review. Thank you for the cautionary note. We tried multiple labeling options of the PCA plots, but still found our original labeling the most informative with the same pattern in the plot. Using assembly IDs only, the reader can't tell differences between assemblies mentioned in the main text (trio based assemblies in green and turquoise), without having to refer to Table 1. Nevertheless, as a compromise, we added a **Supplementary Figure 3** version with the assembly numbers (comparison below), while maintaining the color codes as requested by the reviewer.

Figure showing the PCA plot with fuller assembly descriptions (upper two plots for autosomes and sex chromosomes) and assembly number only (lower two plots).

Referee #2 comment: Pg13.par1. It is stated that the titration of child HiFi data for optimal hifiasm assembly is ~130x, while the methods state that 160x was used (genomes deposited at NCBI also state 160x coverage). So, in the end, for which assembly, 130x or 160x, are the stats reported. Also, if 130x is optimal, why was this assembly not further analyzed and deposited in Genbank. And why were the 34x downsampled reads recommended for the bake-off? At what coverage will the ~450 genomes be sequenced? 34x or 130x or 160x? This is an important detail to be discussed in the discussion.

Response: 130x HiFi (not 160x) was used for the final assembly. We corrected the error in the methods. In the bakeoff we used 34x downsampled levels as it was recommended by the manufacturer of HiFi technology, Pacific Biosciences, and we needed at least one constant coverage level, so that assembly algorithm was the biggest variable difference. For the first set of 45 individuals for which the HPRC decided to generate assemblies we generated 35x HiFi coverage. However, this may not be sufficient to cover all regions of the genome sufficiently. Subsequent tests with improved algorithms we mention here suggest that we can lower coverage from 130x to 60x to get the most complete near –T2T assembly before curation, and we are adding more coverage to some of these assemblies. We have now mentioned these updates on coverage in the discussion.

Referee #2 comment: Also, I think that the amount of “ONT-UL reads” referred to in the main text (please label this set properly) used for the HPRC-HG002 reference should be included in the main text or the methods for clarity. Table S1 has a row labeled “Ultra-long GridION data” that refers to 100kb+ reads (15x from minion and 45x from PromethION filtered from over 52x coverage GridION and 658x PromethION, respectively). Please re-label this set of reads as ONT-100kb+ (or similar). The UL protocol yields a mix of reads, not all of which are 100kb+; in fact, some definitions of UL reads refer to a set of reads with N50 \geq 100kb. I would also recommend a statement in the discussion regarding how this amount of ultralong (100kb+) ONT reads can be generated efficiently as I understand 28 PromethION and 106 MinION flowcells were used to obtain this amount of data. Even at the current lowest bulk rate for PromethION flow cells and assuming higher efficiency, this would still cost quite a bit –fine for producing one reference, but for 450 reference genomes? Same question as for the HiFi reads. How applicable is this approach in general? Of course, the human genome is deserving of such attention and money and time, but casual readers may incorrectly suppose that the approach being put forth is universally applicable, when in fact, generating over 700x ONT, 130 HiFi, Bionano optical maps, 70x Hi-C, 300x Illumina of each parent, etc. is a costly endeavor, not to mention the expertise of at least 3 curators at three different points during assembly. I don’t argue that the human pangenome is not worth it, just that for other projects (of the EBP variety, for example), it is likely impractical.

Response: Similar to as suggested, we changed “Ultra-long GridION data” to “ONT-UL 100kb+” in Table S1; we prefer to keep the UL designation to indicate it is ultra long. For HG002 we purposely generated higher sequence coverage than needed for all technologies, so that different coverage levels can be tested, but asked that at all assemblers test at least the same downsampled manufacturer recommended levels to prevent coverage as a variable when comparing different assembly algorithms. This statement has now been added to the main text. The ONT-UL for HG002 was generated on a GridION for historical reasons; we now obtain 10x+ of >100kb per UL PromethION flow cell. Although we generated 658x ONT reads, we never used this large amount. For the final assembly of this study, we used 78x ONT-UL. We have now made these clarifications in the main text and methods. Improvements in technology continue to be made, such that the lessons learned here can be adopted for large scale projects like the EBP.

Referee #2 comment: Pg13.par1,2. Three references to manual evaluation/curation in a paper titled “Automated assembly of...” See my general comments on this. But one additional clarification is needed: for the contamination screening, is the process automated or are possible contaminants reviewed manually and is removal done automatically or manually?

Response: At the time we performed the assembly contamination screening, removal was mostly manual. Based on lessons learned in this study, we have now automated it. This update is now included in the paper, with a link to the automated pipeline (https://github.com/human-pangenomics/hpp_production_workflows/blob/master/QC/wdl/tasks/contamination.wdl).

Referee #2 comment: Pg14.Fig.3. last line of legend: “the remainder of” seems to be a copy-paste error.

Response: This grammar error is now corrected.

Referee #2 comment: Pg15.par1. “These two de novo assemblies had the highest quality of all metrics compared to...” Change to something like “exhibited the highest quality across most metrics” or “exhibited the highest combined quality across all metrics”

Response: We took the first recommended sentence revision by the reviewer.

Referee #2 comment: Pg.15 last sentence “k-mere” -> “k-mer”

Response: This is now corrected.

Referee #2 comment: Pg17.Fig.4a. I could not find the methods for alignment and coloring by haplotype for this figure in the main text, legend or methods. Are the paternal assembly contigs on top and maternal assembly contigs on bottom? What determines the color value in the heatmap? Despite there being nearly complete haplotype separation, some apparently purple alignment blocks can be seen – is the hue based on parental k-mers somehow? Also, perhaps different colors or symbols could be used to distinguish gaps from unaligned divergent sequence? It is too hard to flip back and forth between 4a and 4b to see where scaffold gaps are located.

Response: Yes, for each chromosome, the paternal and maternal assembly contigs correspond to the top and bottom tracks, respectively. We extracted haplotype-specific k-mers from contigs and determined the color value by the number of these k-mers. For the ambiguous alignment blocks in purple, most of them are very repetitive regions, so that it is hard to extract enough haplotype-specific k-mers, making the trio-based phasing not as clear as non-repeat-rich regions. We included the methods description of the plot, revised the figure legend with more explanation, and revised the figure to show gaps within scaffolds and between scaffolds, and the unaligned regions between the HG002 haplotype assemblies and CHM13.

Referee #2 comment: Pg17.Fig.4b and Table S9. What is meant by “translocation” in this context? My understanding of a translocation is that of a chromosomal break and fusion event, causing a large chromosomal abnormality. This is clearly not the definition being used here. It would appear that no chromosome arms are being exchanged and that these SVs are bits of sequence that are homologous, yet located in different parts of the chromosome, like mobile elements. I would just use the term “rearrangements.” I know SyRI classifies some rearrangements as “translocations” but I think the term is overused in SV detection methods and should be reserved for large events seen in things like cancer.

Response: We now use the more precise term “intrachromosomal translocation” to specify that these are translocations within the same chromosome, determined by the presence of homologous sequences between different parts of the same maternal and paternal chromosome haplotypes. The minimum size for translocations detection we set was 50 bp, but the minimum size we found was 899 bp. We did not observe any type of chromosome arm exchanges. We think that using the term “rearrangements” as equivalent to “translocations” is too broad, as rearrangements is an umbrella term often used for large intra- and interchromosomal translocations, inversions, insertions, deletions, and duplications.

Referee #2 comment: Pg19.par1. “This difference could be due to [the] marmoset assembly using higher error [rate] CLR Pacbio...” please add “the” and “rate”

Response: This is now corrected

Referee #2 comment: Pg19.par2. The 15Mb inversion on Chr9 apparently has no clear Strand-seq alignment orientation evidence to support or reject the inversion according to Extended Data Fig5a and b. It might be nice to see such examples. Could it be a SALSA2 or curation error, given that in the maternal assembly, this region seems to be flanked by gaps (Fig.4b) and could just be a contig orientation error? I wouldn't have noticed if attention hadn't been called to it.

Response: We thank the reviewer for being very observant here. We studied this further, and the reviewer is correct, that it is a contig orientation error flanked by repetitive sequences associated with a gap, in the centromeric region of chromosome 9. We have updated this information in the revised paper.

Referee #2 comment: Pg19.par2. "...6,397 (50%) were synonymous, changing the amino acid sequence..." I think the words synonymous and non-synonymous were switched in this sentence. 46% are (non-?)synonymous. What are the other 4%? Are they nonsense? Are small indels included in the SNV calls that would cause frameshifts? Please make the numbers add to 100% and make sure they make sense.

Response: Yes, we made a typo error here. We meant non-synonymous instead of synonymous. The 4% (e.g. remaining 455 SNVs) were gene annotation errors between either the maternal or paternal assemblies, due to inconsistent gene content liftover from GRCh38. We now filtered out these 455 SNVs from the total number, to get 12,241 correctly annotated SNVs in both the maternal and paternal haplotypes. This results in 6,397 (52.3%) synonymous SNVs and 5,844 (47.7%) non-synonymous SNVs, with the numbers adding up to 100%. Small indels were not included in the SNV analysis, and if included would raise the amino acid changes between maternal and paternal haplotypes. We only included single nucleotide variants. Small indels were included in a separate track of the circos plot. We have now updated these findings with the above clarifications in the revised paper.

Referee #2 comment: Pg19.par3. "yield the highest quality metrics"? I think you mean something like perform the best on the metrics we tested. See previous comment. "could mispartition reads" => "has the potential to mispartition" or "is prone to..."

Response: We corrected the grammar on metrics. We included the reviewer's revision on "is prone to mispartition some reads".

Referee #2 comment: Pg32.Extended Data Figure 11. Inconsistencies between the legend and the figure. Panel d says maternal in the figure and paternal in the legend. e-f legend says q-arm of the assembly. Change "assembly" to Chr12 if in fact it is Chr12 (maternal? Paternal?) The last comment of the legend says that the q-arm telomere is missing from Chr12, while it would appear that it is really the p-arm telomere that is missing. Please review and correct the all aspects of this figure.

Response: We thank the reviewer for discovering these discrepancies. We have corrected each of these aspects of this figure legend and figure. We change the wording to “HG002 Chr 12, maternal assembly”.

Referee #2 comment: Pg34.Supplementary Fig. 2. States near full 130x coverage. 160x is not shown. Why? What is full coverage? 133x? 160x? It states that the “The 130x-300x v0.14 assembly was used for the final HPRC-HG002 assembly of this study.” Why does the assembly submitted to NCBI say 160x? Also, the methods state “Maternal and paternal contigs were generated from 160X coverage of the remaining HiFi reads using hifiasm v0.14.1 in trio mode” on page 47. Sorry, this is a repeat of my previous comment above, referring to this figure.

Response: We apologize for the confusion in coverage numbers. Full coverage refers to all reads. 133x (not 160x) was the full coverage; after removing reads with adaptors and other problems we went from 133x to 130x. We have cleaned up this description.

Referee #2 comment: Pg35.Supplementary Fig. 3. That’s an awful lot of sequencing (28 PromethION FCs and 106 MinION flowcells to fill 15 gaps that take up less than 1 page in a Word doc! Just saying. Maybe in the “A look towards the future” section a small comment could be made about this. Perhaps some targeted sequencing (cas9 enrichment) on the ONT platform could help with the gapfilling effort and bring the cost down...

Response: As mentioned above, we revised the relevant section about the reduced expected coverage levels and cost for complete genomes at this quality level. We agree that this is a lot of ONT sequencing to fill in only 15 of ~195 remaining gaps, albeit reflecting gaps only in very complex regions. In the look towards the future section, we now mention that this high coverage of ONT is unlikely to be required with further targeted improvement of incorporating ONT into graphed based assemblies that combine multiple technologies. We and others have been testing such an approach based on findings in this study.

Methods

Referee #2 comment: Pg40. ONT reads. What are “readsShasta”? Is this the output of a Shasta preprocessing of ONT data? I don’t see anything in the shasta documentation about this. Why is Shasta needed to filter for 100kb+ reads? Potential typo?

Response: We apologize, as this was a typo. This is now fixed. It now reads “We used 28 PromethION flow cells to generate a total of 658x coverage (assuming 3.1 Gb genome size) and ~51x coverage with 100kb+ reads, although we never used all 658x for any one assembly.”

Referee #2 comment: Pg51. Annotations. How many RefSeq transcripts were used? What was the exact query used in Entrez? How were isoforms handled? Was only one transcript per gene chosen or were they clustered on GRCh38 first? If not, how were Unaligned genes calculated: was one missing transcript criteria for being unaligned or all isoforms missing? For calculating collapsed, how is it known the transcripts aligning to the same locus come from the same or different genes?

Response: We now include more details in the methods to address the reviewer’s questions. A total of 81,571 transcripts were retrieved at the start of the process on Dec 8 2021. The query to access these is: “Homo_sapiens[organism] AND srcdb_refseq_known[properties] AND biomol_rna[properties]”, although because of curation this query will return a different set of transcripts today that it did in Dec 2021. The relationship between gene and transcript is established by RefSeq and independent of alignments to any assembly: each transcript is the child of exactly one gene, but a given gene can be the parent of multiple transcripts (alternative variants). All transcripts for a gene were aligned to the assemblies. Of the starting 81,571 transcripts, 78,492 in 27,225 corresponding genes that aligned best to the autosomes of GRCh38 were included in the results in **Table S2**. One unaligned transcript is sufficient to count a gene as “Unaligned” in this table. We changed “Unaligned genes” to “Genes with unaligned transcripts”, to make this clearer, which were either due to one or more transcript alignments being absent or too low in sequence identity.

For the comparison of missing genes in the HG002.pat, HG002.mat, GRCh38 and CHM13 assemblies, we restricted the analysis to genes for which ALL children transcripts failed to be found, and modified the text in the paragraph entitled “Missing genes among individuals and haplotypes” and in the Methods, **Figure 5** and **Table S8** accordingly. This more stringent reporting reduced the total number genes reported missing in at least one of the four assemblies from 119 to 106, which did not change the conclusions.

Referee #3: human genomics

Summary:

Referee #3 comment: The work presented is a progress report of the Human Pangenome Reference Consortium, which aims to sequence ~450 humans, to assemble both the maternal and paternal genomes of those individuals and finally construct a pangenome graph that incorporates all those assemblies. This first step progress report describes computational methods for constructing the assembly of a single individual, HG002. Single individual assemblies have been described before, particularly for CHM13. The difference between this assembly and the CHM13 one is that CHM13 was essentially a haploid genome and its assembly required extensive manual curation. The assembly presented here is mostly able to resolve the two haplotypes. The authors argue that their work requires less manual curation, although considerable manual curation seems to be required and a fairly large number of gaps still remain, although much fewer than in GRCh38.

Response: The reviewer's summary is an accurate description of our study, except that the CHM13 assembly required more extensive manual curation relative to what was done for HG002 in this study to get to the same assembly quality before final completion. We note that the final manual curation of the assembly in this manuscript was meant to achieve the complementary goal of generating the best possible assembly with existing technologies and algorithms, which will serve as a benchmark for future efforts. This manual curation led to an assembly of very high quality. In the revised paper, we emphasized this point further.

Referee #3 comment: The work of the Human Pangenome Reference Consortium of no doubt has the potential to be of immense value when finished. However, it is not clear to me what the value of the current report is and the report also suggests that the study design for the final project could be improved upon.

Response: The value of the current report is that it is the first major step in generating methods for near complete, diploid reference genome assemblies, at scale. The study allowed us to generate the highest quality diploid human genome to date, and simultaneously identify areas of needed improvement that the reviewer noted, which we think is of high value. Both reviewers #1 and #2 summarized the high value of items in the study.

Major Comments:

Referee #3 comment: 1) A fairly convincing argument is made that the reference presented here accurately represents the two haplotypes of the individual being studied. However, no results are given where this assembly is used and no recommendation is given to the reader on how to use it. Does this mean that the authors do not recommend the usage of the assembly presented? As far as I can tell the assembly is not being released.

Response: The final diploid assembly has been released, under accession numbers GCA_021951015.1 and GCA_021950905.1 for the maternal and paternal haplotypes, respectively. These accession numbers were mentioned in the Data Availability section of the paper. In case readers have difficulty finding the accession numbers, we now include them and a statement of public availability in the main text of the results. Yes, we do recommend that this diploid assembly be used by the public. The HPRC is using it for follow up studies. We used this assembly to make new discoveries in heterozygosity across chromosome partitions, gene losses and gains between haplotypes, and structural variants between haplotypes. We made this message clearer in the revised manuscript.

Referee #3 comment: 2) The assembly is performed on an extensively studied individual, it is understandable that the authors would like to test their methods on this individual as a curated set of true variants exist for the individual. This however makes the study design susceptible to overfitting. First much of the same data was used in the construction of the truth set, but more importantly the algorithms have been trained to fit biases in these dataset. A better study design would have been to use separate test and training sets, i.e. the algorithms would be trained on one set of data and then evaluated on another set. At minimum it is clear that the number of assembly errors estimated using this approach is a downward biased estimate of the numbers of errors one would expect to get for additional assemblies.

Response: We agree with the reviewer's points in principle. However, in practice, the starting assembly approaches and the final chosen assembly approach of trio hifiasm were not developed only on HG002 data, or on HG002 data at all. Conversely, the contig approach we developed has now been applied to over 45 human genomes obtaining similar values in our metrics. Likewise, the contig and scaffolding approaches have now been successfully applied to over 20 vertebrate species in the VGP. These subsequent assemblies and considerations are now noted more clearly in the discussion. Additional details will be reported in separate studies, but they were generated based on the findings in this study.

Referee #3 comment: 3) The study design is to evaluate a number of algorithms, compare that to an assumed truth and then the results for this comparison is used to develop a final super-algorithm by correcting errors made by other algorithms. This even further biases the results.

Response: We consider the "assumed truth" as not one given a "true assembly", but rather the consensus of multiple types of evidence. This includes reference-free consistency between all raw data types (HiFi, ONT, Illumina, Bionano) and the assembly, orthogonal data (Strand-Seq), assembly quality metrics, and reference analysis to a complete and accurate T2T-CHM13 assembly of another individual, although haploid. In case other readers have the same concern, we mention the above clarification in the methods.

Referee #3 comment: 4) The HG002 benchmarks mainly target relatively easy parts of the genome and mostly those that are already in GRCh38. However, the main problems in assembly are in those regions not reached by GRCh38, giving a further downward bias of the estimated error of the method.

Response: We assume that the reviewer is referring to the HG002 variant benchmarks (not GRCh38), generated by the GIAB consortium? This is one of the reasons we used HG002. We found that our final HG002 assemblies exceeded these benchmarks in both the easy and hard to assemble regions, including centromeres, telomeres, and gene duplications. For this reason, our new assembly has become the new benchmark. Further, we also compared our assemblies to variants from GRCh38 and CHM13 assemblies, the later of which had the hard to assemble regions complete. Thus, we do not see how this predicts a downward bias of our new assembly.

Referee #3 comment: 5) The authors don't explicitly state how they plan to use this assembly or others in the construction of the pangenome, but I infer from the text that the plan is to construct an assembly of each one of the individuals and then merge the assemblies into a graph. I am very much concerned about this study design if this is the case. It would be prudent of the authors to reconsider this design given the results of this first phase study, in particular as the results suggest that this is not likely to be the best possible design. In particular the proposed focus in future work on improving diploid assembly seems to be misplaced faith on part of the authors and a poor design decision.

Response: The focus of the current paper is not on the design of pangenome graphs, but on how to generate the most complete and accurate diploid assembly possible with as little manual curation as possible. The pangenome graph is the subject of our next study, which is in preparation. We now more clearly point the reader to a more detailed preliminary discussion of what new developments we are working on for a pangenome graph in an HPRC perspectives paper published in *Nature* since the submission of this first primary research article (<https://www.nature.com/articles/s41586-022-04601-8>). Yet, we believe that generating complete, haplotype phased, and accurate genome assemblies is critical for later generating accurate pangenome graphs. In our forthcoming paper we will show that this design indeed works well for building a robust pangenome, but we prefer to discuss and argue this point in that paper where there is space and scope for the evidence.

Referee #3 comment: a) The problem of merging assemblies into a graph is not well solved. In particular it is clear that graphs allow for multiple equivalent representations of the same sets of variants and that different merging strategies could lead to different graphs. This means that once they have constructed the assembly of all these individuals they still have to solve the problem of merging. The merging process is likely to propagate errors from each of the individual assemblies to the combined graph assembly and could also introduce further errors into the final graph assembly

Response: The reviewer is correct in that merging haplotypes of an assembly into a pangenome graph is difficult, and can have errors in part due to misalignments made by alignment algorithms. This is especially the case in the diverse centromeres we found in current study. These issues are successfully being dealt with in our pangenome study, and we wish to not steal the thunder of that study from the current study. We just simply note here that high-quality assemblies are necessary for high-quality pangenome graphs.

Referee #3 comment: b) Although the number of assembly errors appear to be much smaller than in GRCh38, the merging of 900 assemblies (at least naively) requires that the errors per assembly be 900 times lower so that the errors in the combined assembly are fewer than GRCh38, a threshold not reached in the current study.

Response: Again, this paper is not about the pangenome graph. However, we note here for the reviewer's benefit, that the final reference HG002 assemblies we already generated at most 100-200 fold improvements over GRCh38 in some metrics, with those improvements leading to nearly T2T chromosomes. So 900-fold better may not be mathematically possible nor necessary.

Referee #3 comment: c) The results presented here show that the best algorithm is to use sequence reads from parents along with the sequence read of the individual. In addition they show that algorithms can benefit from a reference, although caution needs to be taken in which inferences to draw from the reference. In light of this, why would the authors not consider a joint assembly of the multiple genomes for their future work

Response: A joint merged assembly of multiple individuals would amplify the errors many-fold, because haplotypes are one giant repeat. The need to separate haplotypes to prevent the assembly errors has been part of some of the conclusions of the VGP and T2T-CHM13 studies, and new methods for phasing assemblies, such as the trio binning and hifiasm Hi-C. We wrote the following statement in the introduction of the original submission: *"It is also now clear that merging diverse haplotypes in a single haploid assembly, even from the same individual, introduces many errors with standard assembly tools"*. This is the reason to not further consider a joint assembly within the same and between multiple individuals. We have tried to make this point clearer in the revised paper.

Referee #3 comment: 6) The authors of Giraffe (<https://www.science.org/doi/full/10.1126/science.abg8871>) found that there was a saturation in alignment accuracy when combining more than 64 haplotypes in an assembly graph. Do the authors have evidence to refute these claims?

Response: Fortunately, the lead authors of Giraffe are authors on the current study. They have improved upon the principles of Giraffe in the pangenome study the HPRC is working on. Giraffe is a short read mapping tool to characterize a pangenome, and is thus relevant to the pangenome study in progress.

Referee #3 comment: 7) Whole genome assembly algorithms have been in development for over 20 years. They have advanced with different sequencing methods, which have and continue to evolve rapidly. Do the authors have reason to believe that this will not continue to be the case during the course of the Human Pangenome Project?

Response: Yes, as mentioned in the discussion, we expect that algorithms and data quality will continue to improve, including due to the progress we made in the current study. The HPRC's goal and their relationship with the VGP and EBP is to continue to work on improvements to ultimately achieve T2T complete and accurate assemblies, and at scale. The results and methods developed here set an important diploid standard and benchmark milestones for these future studies. We further clarified this goal in the discussion.

Referee #3 comment: 8) The authors seem to implicitly make the assumption that all variants in the proposed final graph will be from the individual assemblies. Given the large number of whole genome sequenced individuals it seems a waste to not use some of the information already present in those.

Response: We believe the reviewer is referring to variants discovered in more draft quality genomes and whole genome resequencing efforts? If so, yes, variants from the available large numbers of sequenced individuals could be theoretically included in the reference assembly graph. However, we caution that a proportion of those variants will be assembly errors, as shown here in this study. Nevertheless, this is a question being addressed in our next study on generating a pangenome reference.

Referee #3 comment: 9) Most of the paper focuses on comparing algorithms that went into the bakeoff competition. I understand that this was important work but it is not clear to me if this is of general interest to the readers of the journal. Also it is not clear to me why the results for all the algorithms need to be presented as some are clearly inferior to others.

Response: A number of highly cited studies have been published in *Nature* and other high profile journals on methods and algorithms for producing high quality assemblies and data (Zhou et al 2022 Graph pangenome; Rhei et al 2021 VGP; Miga et al 2020 Complex X chromosome; Matthews et al 2018 mosquito all recent in *Nature*). We think this study will be of similar interest. We also think it is

important to present all of our results. It is not easy to predict which algorithms will be clearly inferior to others without empirical experimental testing. Some of the tools we demonstrated to be inferior in this study are indeed still being widely used. In terms of general interest beyond methods, we think it is important to clearly get the point across of the difficulty and the need to generate diploid assemblies, as well as examples of new biological discoveries learned from more complete diploid assemblies.

Referee #3 comment: 10) The authors should rephrase their discussion of the mitochondria assembly and perhaps also reconsider their modeling assumptions. The mitochondria is not expected to have a single haplotype. Individual cells may carry multiple mitochondria, which may differ from each other.

Response: Yes, we noted somatic heteroplasmy in mitochondrial genomes from the same individual as part of our mitogenome analyses in the VGP (Formenti et al 2021 *Genome Biology*). While our primary target in this study is the nuclear genome, following the reviewer's comment, we have now included an analysis of mitochondrial heteroplasmy in HG002 (**new Supplementary Figure 9**). We mapped all HiFi reads available for HG002 back to both maternal and paternal haplotypes combined with the reconstructed mitochondrial reference sequence. A total of 11,938 HiFi reads aligned to the mitochondrial reference. The alignment shows evidence of 6 SNPs at >1% frequency, which could be interpreted as mitochondrial heteroplasmy in the cell line. We chose the 1% threshold since lower frequency variants would be confounded with read errors. In one case, the major allele (T) is supported by 8033 reads (97%, 4186+, 3847-), while the minor allele (C) is supported by 202 reads (2%, 94+, 108-), but the minor allele was in the reference mitochondrial genome submitted. We have included this new analysis on the mitochondrial genome in the paper.

Supplementary Fig. 9. Mitochondrial genome heteroplasmy in HG002 cell line. Shown are raw HiFi reads mapped back to the reference HG002 mitochondrial genome assembly. Red vertical lines indicate SNPs at a frequency of >1%. The one red line prominent in the raw reads indicates that the minor allele was in the reference. This heteroplasmy could have been presented in the original blood cell plasma isolated from HG002 and/or in the subsequent cell line.

Minor comments:

Referee #3 comment: 1) “Our findings serve as a foundation for assembling near-complete diploid human genomes at the scale required for constructing a human pangenome reference that captures all genetic variation from single nucleotides to large structural rearrangements”. - A pangenome that captures all genetic variation is likely an unattainable goal. The authors state later that the work is to be the foundation for a graph with most variants > 1% frequency, a more reasonable goal.

Response: We now made these two statements consistent with each other, by stating our goal is to capture human genetic variation with a population frequency of at least 1%. In the abstract, we change “all genetic variation” to “global genetic variation”.

Referee #3 comment: 2) “Its current build, GRCh38, reflects two decades of additional effort by the Genome Reference Consortium and others to correct the initial draft.” - A lot of work on creating alternate assemblies has contributed to GRCh38.

Response: We assume that the reviewer is referring to the alternate haplotype fragments that have been generated (and thus variants) as well as alternate paths through the assembly graph (<https://www.ncbi.nlm.nih.gov/grc/help/definitions/#ALTERNATE> and <https://www.ncbi.nlm.nih.gov/grc/human>)? If so, these are still highly fragmented and not part of the primary assembly. To make our point clearer, we revised the sentence from “to correct the initial draft” to “to correct the initial draft of the primary assembly”.

Referee #3 comment: 3) “and does not adequately capture the full spectrum of humankind genomic variation” - capturing “the full spectrum” of humankind genomic variation is likely an unattainable goal, considering that de novo mutations occur in most meiosis and mitosis.

Response: We changed “full spectrum” to “global spectrum”.

Referee #3 comment: 4) “While resequencing efforts using less expensive short reads have contributed to revealing more single-nucleotide variation, structural variation is not fully captured” - Neither SNPs nor SVs are fully captured. Long read efforts have also contributed extensively to the characterization of SVs.

Response: We modified this sentence to indicate that SNVs are also more fully recovered with long reads.

Referee #3 comment: 5) Figure 1: The color coding (blue, light blue and turquoise) is confusing. A more diverse palette of color would be helpful. What does “more complete” mean in the caption?

Response: The assemblies are color-coded according to type of haplotype phasing: different shades of blue are diploid assemblies; different shades of green are haploid assemblies. More specifically: blue, trio-based; light blue, non-trio based and comparable haplotypes; turquoise, non-trio based and a more complete one pseudohaplotype; dark green, merged haploid contigs and scaffolds; light green, merged haploid contigs only. We have provided a more detailed description of the color coding in legend of Table 1. We will work with the journal editors to use a recommended distinct color palette that distinguishes these categories and sub-categories. We have revised “more complete for one haplotype” with “more complete assembly representing one pseudohaplotype”.

Referee #3 comment: 6) I think it would be useful to have somewhere in the paper a quick list of what data were available/used to perform the assemblies and more importantly at what coverage. Additionally, I would have liked to see a quick reminder of what distinguishes the different types of phased assemblies such as in <http://lh3.github.io/2021/04/17/concepts-in-phased-assemblies>.

Response: The information on data types and coverage levels used for each assembly were provided in Table S1 and Table S2. The data available was presented in the github link provided in the paper, once in the methods and again in the Data Availability section https://github.com/human-pangenomics/HG002_Data_Freeze_v1.0 We added a reminder description of the different types of phasing in the methods. The link the reviewer supplied above does not work.

Referee #3 comment: 7) The HPRC Trio pipeline as illustrated in Figure 10b is very useful but also shows that a lot of different sequencing technologies (most likely at very high coverage) are required to build such a type of assembly. I would have appreciated to see a summary of what each data type brings to the final assembly such as “UL ONT enables (mostly) gap filling”. I think this would be extremely useful for many labs who cannot afford all these sequencing technologies but instead, would need to prioritize some type of sequencing technology that fits the needs of their project.

Response: We added a summary of the contribution of each data type in the new github link (https://github.com/human-pangenomics/hpp_production_workflows/wiki/Assembly-Best-Practices).

Referee #3 comment: 8) The authors hint at the fact that the variant calls which differ between the assembly and the Genome In a Bottle truth set are most likely correct in the assembly but incorrect in the GIAB truth set. This is a major finding in my opinion given that many methods nowadays are optimizing their results or creating models based on this truth set. A list of these differences should be provided and eventually a new truth set without these variants should be released.

Response: The GIAB coauthors of this manuscript agree that the accuracy of this diploid assembly is quite promising. In fact, they recently published a benchmark for challenging medically relevant genes based on an earlier version of this assembly, and are currently working on refining and evaluating benchmarks based on the assembly presented here for as much of the genome as possible. As suggested by the reviewer, we now point to the list of differences at [https://ftp-trace.ncbi.nlm.nih.gov/ReferenceSamples/giab/data/AshkenazimTrio/analysis/HPRC-HG002.cur.20211005/..](https://ftp-trace.ncbi.nlm.nih.gov/ReferenceSamples/giab/data/AshkenazimTrio/analysis/HPRC-HG002.cur.20211005/)

Nitpicking:

Referee #3 comment: Can “latter” be used for the last one of three?

Response: We removed the word “latter”, and repeated what the last of three was.

Reviewer Reports on the First Revision:

Referees' comments:

Referee #1 (Remarks to the Author):

The authors have addressed all my questions and I am satisfied with their responses. A few additional comments are listed below, but I leave it up to the authors to decide if they would like to make any further changes.

- The use of lymphoid cell lines (LCLs) is discussed in the revised manuscript and karyotyping results for HG002 has been included. I think these additions were important. In addition, the authors performed sequence analysis of cell line vs blood samples from a separate individual (HG06807) as described in Supplementary Note 4. As far as I can see, the analysis of HG06807 is mainly focused on SNV differences between blood and the cell line. It could be interesting to also see a more detailed comparison of SVs between the samples, since such events will likely be missed both by SNV calling and karyotyping. However, I do understand if the authors consider such analyses to fall outside of the scope of the current paper, especially since a more detailed investigation of HG06807 has limited relevance for the diploid HG002 assembly.
- Regarding automation and availability, the authors have now set up a best practices website and the tools are hosted on the Galaxy website. These links are very useful. However, the Galaxy page (<https://assembly.usegalaxy.eu/>) mainly contains information about biodiversity-related projects, such as VGP and ERGA. Maybe that the Human PanGenome Reference project could be explicitly mentioned on the Galaxy site? I don't know though whether the authors have possibility to make such changes.
- The authors have proposed a new manuscript title, but maybe that the word "semi-automated" could be removed to make it more concise. However, this might just be my opinion.
- I really enjoyed reading the results on mosaicism. Obviously, it would be interesting to dig further into this and investigate the mosaic patterns on parental chromosomes in more detail and across individuals. But again, this is probably outside the scope of the present study.

Adam Ameur

Referee #2 (Remarks to the Author):

I am satisfied with the revisions made in response to my comments as well as those of the other reviewers. The result is a clearer, more accurate report of the work carried out and the conclusions drawn.

Referee #3 (Remarks to the Author):

In the revised manuscript the authors have considered some of the comments from the previous round of review. I continue to believe that the presentation could be improved upon. The manuscript is quite long, but also needs to be put better into context for the general reader.

Major comments:

1. As per my previous comment 9, the very long description of comparisons of the merits of 23 different algorithms could be shortened substantially, by either discussing fewer algorithms or summarizing the various metrics being discussed.
2. The authors have misinterpreted some of the comments that I made in the previous review. In particular comment 5, which had three subpoints a, b and c. I appreciate that the work is part of a larger project, but putting the work into context would be useful for the reader and could be done without substantially lengthening the paper. Is it a correct assumption that the design of the HPRC is to build a diploid assembly for each individual and then merge these assemblies to build a pangenome assembly? If so, this could be easily stated in the manuscript.
3. Putting the work in context also allows for questioning the initial study design. This is especially important for this manuscript as the authors state that this is the first scientific report of HPRC. A simple reflection on how the work presented, as the first subproject, reflects on the overall project and whether and how the initial study design can be improved upon is prudent. This is especially important as it seems that the HPRC may take a long time to finish and sticking to the initial design will only lengthen the project, if the original study design made assumptions that later proved to be incorrect. I have no knowledge of whether such imperfect assumptions were made in the original study design, but in my experience all study designs can be improved with more data.
4. The authors have misinterpreted my original comment 5c). Of course it is well known that merging the sequence reads from multiple individuals and building a single haplotype assembly is not a good study design. However, the final goal of the project is to build a graph that represents the haplotypes of all 450 individuals. As a starting point to building the graph the authors seem to suggest building the haplotypes of all individuals. To build the haplotypes one could, as it seems you are suggesting, start by building the two haplotypes of each individual. Alternatively, one could build all 900 haplotypes simultaneously. In light of the results of the paper, which suggest that using trios and a reference is beneficial in building a haplotypes, it would seem that the second option of building all the 900 haplotypes simultaneously would be sensible. This would allow one to use a) multiple other assemblies as a reference when building a given haplotype assembly b) long range sharing of haplotype of unrelated individuals to build assemblies in regions where they share haplotypes and c) use linkage disequilibrium (LD) information.
5. I have two comments regarding the claim: "One also needs a diploid assembly to: separately assemble the X and Y sex chromosomes; determine maternal and paternal imprinting of gene expression, with gene variants that lead to haplotype-specific diseases and determine functional consequences of allele combinations that co-segregate on the same haplotype. Thus, there are many reasons for needing to accurately separate out haplotypes of individuals during assembly."
 - a. The applications (imprinting of gene expression and determining functional consequences of haplotype-specific diseases) could also be solved using a single linear reference followed by haplotype phasing, as has been done in a number of publications already.
 - b. These types of relationships are generally only determined from a large set of individuals, which

would generally carry more than two haplotypes across a region, hence it is unlikely that a diploid assembly would make much progress towards this goal. The authors might argue that a diploid assembly could be performed per individual and then results somehow merged across individuals, but how to perform this step is very much unclear.

6. It would be desirable if the authors could offer support for this claim “needed to assemble diploid genomes at high quality and at scale, which are critical for clinically relevant samples and understanding human genetic variation.” If not, the authors might replace “are” with “we believe to be” or similar.

7. “Assessing against GIAB HG002 benchmarks further, this diploid assembly produced highly accurate SNV concordance (F1 score) of 99.7% and small indel concordance of 98.6% when aligning HPRC-HG002.mat and HPRC-HG002.pat haplotypes to GRCh38.” - These estimates are likely upwardly biased, as noted in my previous review and the authors seemed to agree in their response. It might be helpful for the reader to note this bias. In my previous review, I presented a number of examples of where the study design might be biased. I believe it would be helpful for the reader if such biases were pointed out to them.

8. The authors have added an analysis of mosaicism within haplotypes. I suggest that this analysis be removed. Large scale whole genome sequence analyses have studied this in great detail (studying tens of thousands of WGS genomes) and the current study is very much underpowered to draw meaningful conclusions. The previous studies have shown that estimates of mosaicism can be very much affected by sequencing artifacts, but also by age and a number of diseases and traits, such as Clonal hematopoiesis and various cancers in stem cells or the tissue being studied. It is hence not clear if the difference in mosaicism rate between this human sample and the marmoset sample is due to species level differences or within species sample level differences. The authors might argue that the haplotype assembly is adding some value to the analysis, but haplotypes can also be inferred using computational techniques.

9. As pointed out in my previous review, human cells do not necessarily carry a single mitochondrial haplotype. Each cell may have multiple mitochondria, each of which has an haplotype associated with it. These haplotypes may or may not be the same. The authors have therefore not provided a “haplotype assembly” of a mitochondria but rather a consensus assembly of the multiple haplotypes carried by the cells that were sequenced.

10. It would be helpful for the reader if an example could be given where the reader might make use of the references presented. I.e. apart from getting a better estimate of the sequence of the individual being studied, is there some application where the reference, as such, could be of use to the reader? Is there some use case you can present where the reader might benefit from using these assemblies instead of previously published assemblies (e.g. GRCh38 or CHM13) or in addition to those assemblies?

Minor comments:

1. I have difficulty parsing “a hydatidiform mole with nearly two identical paternal haploid complements inherited from the maternal line”. Are the paternal haploid complements from the father of the mother representing the maternal line? If so, it might be clearer to refer to grandpaternal.

2. “Assemblies that did not have microbial contamination were because: 1) they did not match the GRCh38 reference for the reference-based assemblies; 2) they were removed in assemblies that

filtered out scaffolds below a specific size; or 3) they were all moved to the alternate assembly.” - It is difficult for me to parse this sentence. Are you listing the three strategies that were used for removing microbial contamination?

3. Table 1 and Table S2 don't have the same headings for assembly categories, Figure 1 uses mostly the same headings as Table S2.

4. “The scaffolded assemblies still had quite a range of missing sequence (Ns), from ~40 kb to 50 Mb, either in the gaps between contigs or trailing Ns at scaffold ends (Extended Data Fig. 2c; Supplementary Table 2f). In comparison, GRCh38 has ~151 Mb of gaps.” - In the first sentence, missing sequence (Ns) are split into gaps and trailing (Ns), but then in the second sentence “gaps” are used to represent the number of Ns (counting both gaps and trailing Ns).

5. “which could also explain why some were bigger than expected” - above you state that only the diploid pair asm19a/asm19b was bigger than expected.

6. “There were about a dozen genes present in GRCh38 and asm17 that used it as a reference, but not in any of the other HG002 assemblies or CHM13, illustrating a bias of gene presence for reference-based assembly methods” - Why is this bias and not true positives?

7. Heatmap, the lighter the blue (closer to 1) the more similarity between pairs of assemblies - I don't understand what “closer to 1” refers to.

8. “Of the 3,690, 2,466 genes had exclusively non-synonymous differences, and were significantly enriched (FDR < 0.01) for metabolism, smell, taste, and HSV1 viral infection functions (Supplementary Table 12)” - The fact that these categories of genes have a higher polymorphism rates has been well documented in numerous other studies (better powered to do such an analysis).

9. “suggesting differential inherited differences from the parents.” - I also have a trouble parsing this comment.; Obviously since the haplotypes are different there were differences in what was inherited from the parents, is that what you mean by “differentially inherited differences” or something else?

10. “Future efforts will be necessary to develop a phasing method that does not require parental sequence data and works as well as a trio method.” - In addition to the methods mentioned, computational methods for determining haplotypes in a large set of individuals could be used. Instead of using parents more distantly related individuals could be used, long range sharing of haplotypes between these individuals could serve as “surrogate parents” . A large set of haploid references will also be helpful.

Author Rebuttals to First Revision:

Referee #1:

Reviewer #1 comment: The authors have addressed all my questions and I am satisfied with their responses. A few additional comments are listed below, but I leave it up to the authors to decide if they would like to make any further changes.

Reviewer #1 comment: The use of lymphoid cell lines (LCLs) is discussed in the revised manuscript and karyotyping results for HG002 has been included. I think these additions were important. In addition, the authors performed sequence analysis of cell line vs blood samples from a separate individual (HG06807) as described in Supplementary Note 4. As far as I can see, the analysis of HG06807 is mainly focused on SNV differences between blood and the cell line. It could be interesting to also see a more detailed comparison of SVs between the samples, since such events will likely be missed both by SNV calling and karyotyping. However, I do understand if the authors consider such analyses to fall outside of the scope of the current paper, especially since a more detailed investigation of HG06807 has limited relevance for the diploid HG002 assembly.

Response: We have now included SV results of an alignment of HG06807 assemblies from blood-derived and LCL-derived cells. We found three small inversions in the maternal LCL genome, indicating a low level SV change, but not major changes in genome structure. These results have been added to Supplementary Note 4.

Reviewer #1 comment: Regarding automation and availability, the authors have now set up a best practices website and the tools are hosted on the Galaxy website. These links are very useful. However, the Galaxy page (<https://assembly.usegalaxy.eu/>) mainly contains information about biodiversity-related projects, such as VGP and ERGA. Maybe that the Human PanGenome Reference project could be explicitly mentioned on the Galaxy site? I don't know though whether the authors have possibility to make such changes.

Response: Following the reviewers request, we now explicitly described the Human Pangenome Reference Consortium on the Galaxy website (<https://assembly.usegalaxy.eu/>).

Reviewer #1 comment: The authors have proposed a new manuscript title, but maybe that the word "semi-automated" could be removed to make it more concise. However, this might just be my opinion.

Response: In the previous round, the reviewer (e.g. Reviewer #1) asked that we add "Semi-automated" to the title, and the editor confers that we would keep it. We also agree, since the approach is semi-automated. We are not sure what caused the reviewer to suggest the opposite, but

for the original reasons, we have kept the reviewer's first suggestion of semi-automated in the revised title.

Reviewer #1 comment: I really enjoyed reading the results on mosaicism. Obviously, it would be interesting to dig further into this and investigate the mosaic patterns on parental chromosomes in more detail and across individuals. But again, this is probably outside the scope of the present study.

Response: We thank the reviewer for the original suggestion that led to our mosaicism analyses. As the reviewer suspected, more detailed analyses across individuals are outside the scope of the present study but we plan to perform them in future studies when we analyze genomes of multiple individuals being generated for the HPRC.

Referee #2

Reviewer #2 comment: I am satisfied with the revisions made in response to my comments as well as those of the other reviewers. The result is a clearer, more accurate report of the work carried out and the conclusions drawn.

Response: We are glad that the reviewer is satisfied with the revisions. Thank you for the positive feedback.

Referee #3

Reviewer #3 comment: In the revised manuscript the authors have considered some of the comments from the previous round of review. I continue to believe that the presentation could be improved upon. The manuscript is quite long, but also needs to be put better into context for the general reader.

Response: We tried to balance further requests for increased content, with requests to reduce the length of the manuscript by the editor and implied by the reviewer. We shortened the manuscript some, but we have been given lead way to publish a longer manuscript than usual, as part of a special series of manuscripts on the panhuman genome effort being considered in *Nature*.

Major comments:

Reviewer #3 comment: 1. As per my previous comment 9, the very long description of comparisons of the merits of 23 different algorithms could be shortened substantially, by either discussing fewer algorithms or summarizing the various metrics being discussed.

Response: The other reviewers were satisfied with our current length, including presenting results of all the algorithms tested. Further reviewer #2 mentioned being satisfied with our response to the other reviewer's comments. Thus, we prefer to compare and contrast the merits of all algorithms tested. A summary of our previous justification to comment 9 were: a) that there are highly cited studies published in *Nature* and other journals on methods and algorithms for producing high quality assemblies and data, like this study; b) it is important to present all relevant results; c) it is not easy to predict which algorithms will be clearly inferior to others without empirical experimental testing; and d) tools that one might categorize as inferior in this study are still being widely used, and thus our results need to be presented in a transparent manner to the community. Nevertheless, we have reduced the manuscript length, without compromising description of the results for all algorithms tested.

Reviewer #3 comment: 2. The authors have misinterpreted some of the comments that I made in the previous review. In particular comment 5, which had three subpoints a, b and c. I appreciate that the work is part of a larger project, but putting the work into context would be useful for the reader and could be done without substantially lengthening the paper. Is it a correct assumption that the design of the HPRC is to build a diploid assembly for each individual and then merge these assemblies to build a pangenome assembly? If so, this could be easily stated in the manuscript.

Response: Yes, it is "*a correct assumption that the design of the HPRC is to build a diploid assembly for each individual and then merge these assemblies to build a pangenome assembly.*" To improve clarity, we have now used nearly this exact sentence in the revised introduction. However, we again did not add as much description about pangenome construction requested by the reviewer in the previous review, as it is the subject of our subsequent paper, which has since been submitted and posted in bioRxiv (Liao et al <https://www.biorxiv.org/content/10.1101/2022.07.09.499321v1.full.pdf>). All the reviewer's questions on comment 5 has been answered in that paper. We hope that this is a satisfactory solution.

Reviewer #3 comment: 3. Putting the work in context also allows for questioning the initial study design. This is especially important for this manuscript as the authors state that this is the first scientific report of HPRC. A simple reflection on how the work presented, as the first subproject, reflects on the overall project and whether and how the initial study design can be improved upon is prudent. This is especially important as it seems that the HPRC may take a long time to finish and sticking to the initial design will only lengthen the project, if the original study design made

assumptions that later proved to be incorrect. I have no knowledge of whether such imperfect assumptions were made in the original study design, but in my experience all study designs can be improved with more data.

Response: We agree with the reviewer, and we have now emphasized these points further. Our overall description of the study design was reported in our perspectives paper, published in *Nature* earlier this year (Wang et al 2021: *The Human Pangenome Project: a global resource to map genomic diversity*; <https://www.nature.com/articles/s41586-022-04601-8>). We further point the reader to this perspectives paper in the introduction, to place the paper in context. Yes, we do expect methods to change and improve, and the consortium is designed to be flexible.

Reviewer #3 comment: 4. The authors have misinterpreted my original comment 5c). Of course it is well known that merging the sequence reads from multiple individuals and building a single haplotype assembly is not a good study design. However, the final goal of the project is to build a graph that represents the haplotypes of all 450 individuals. As a starting point to building the graph the authors seem to suggest building the haplotypes of all individuals. To build the haplotypes one could, as it seems you are suggesting, start by building the two haplotypes of each individual. Alternatively, one could build all 900 haplotypes simultaneously. In light of the results of the paper, which suggest that using trios and a reference is beneficial in building a haplotypes, it would seem that the second option of building all the 900 haplotypes simultaneously would be sensible. This would allow one to use a) multiple other assemblies as a reference when building a given haplotype assembly b) long range sharing of haplotype of unrelated individuals to build assemblies in regions where they share haplotypes and c) use linkage disequilibrium (LD) information.

Response: The reviewer's first alternative is correct about what the consortium is doing, in part as an outcome of this paper. We are building high-quality haplotype assemblies first, and then using them to create a pangenome graph that displays variation among all haplotypes. The second alternative that the reviewer proposes is not what the consortium is doing. The reason is that the reference-based approaches tested in our study produced assemblies of lower quality, in part because they had errors biased towards the reference. For the other suggestion, building 900 haplotype assemblies simultaneously would be very messy. As shown in the paper and other studies cited, even trying to build two haplotypes simultaneously leads to many haplotype-related genome assembly errors. So we do not recommend this approach. We have now made these points clearer in the revised manuscript.

Reviewer #3 comment: 5. I have two comments regarding the claim: "One also needs a diploid assembly to: separately assemble the X and Y sex chromosomes; determine maternal and paternal imprinting of gene expression, with gene variants that lead to haplotype-specific diseases and determine functional consequences of allele combinations that co-segregate on the same haplotype. Thus, there are many reasons for needing to accurately separate out haplotypes of individuals during assembly."

a. The applications (imprinting of gene expression and determining functional consequences of haplotype-specific diseases) could also be solved using a single linear reference followed by haplotype phasing, as has been done in a number of publications already.

b. These types of relationships are generally only determined from a large set of individuals, which would generally carry more than two haplotypes across a region, hence it is unlikely that a diploid assembly would make much progress towards this goal. The authors might argue that a diploid assembly could be performed per individual and then results somehow merged across individuals, but how to perform this step is very much unclear.

Response: For comment “a”, the proposed alternative of phasing the haplotypes from a linear reference will always lead to reference bias, and any complex variation will often be missed in this approach. This is shown also in the current study, where the algorithms that make a linear reference first and then phase haplotypes are inferior to those that phase before assembly. The most accurate assembly is phasing during the assembly process in the graph. We have tried to make this point clearer in the revised paper. For comment “b”, we agree that rigorous discovery of allelic imprinting of gene expression and other allelic relationships needs a sample size of more than $n = 1$ diploid genome assembly. We did not intend to imply this. We changed the beginning of the sentence to be plural. From “One also needs a diploid assembly:...” TO “One also needs diploid assemblies:...”. We hope that will prevent others thinking that an $n = 1$ assembly is sufficient for addressing these questions, and the HPRC is about generating many high-quality diploid assemblies.

Reviewer #3 comment: 6. It would be desirable if the authors could offer support for this claim “needed to assemble diploid genomes at high quality and at scale, which are critical for clinically relevant samples and understanding human genetic variation.” If not, the authors might replace “are” with “we believe to be” or similar.

Response: We changed the text to “we believe to be” as suggested by the reviewer.

Reviewer #3 comment: 7. “Assessing against GIAB HG002 benchmarks further, this diploid assembly produced highly accurate SNV concordance (F1 score) of 99.7% and small indel concordance of 98.6% when aligning HPRC-HG002.mat and HPRC-HG002.pat haplotypes to GRCh38.” - These estimates are likely upwardly biased, as noted in my previous review and the authors seemed to agree in their response. It might be helpful for the reader to note this bias. In my previous review, I presented a number of examples of where the study design might be biased. I believe it would be helpful for the reader if such biases were pointed out to them.

Response: In our prior response, we said we agree with the reviewer’s points in principle - of what could cause bias - but then stated that for all three of the reviewer’s comments (2, 3 and 4) our evidence does not reveal such a bias. The reason is that unlike what the reviewer thought, we were

comparing algorithm tools and haplotype variation on more than one individual (more than HG002). The reviewer mentioned “downward” bias (as opposed to upward) in the previous review. We argue our same explanation applies to possible upwards biases.

Reviewer #3 comment: 8. The authors have added an analysis of mosaicism within haplotypes. I suggest that this analysis be removed. Large scale whole genome sequence analyses have studied this in great detail (studying tens of thousands of WGS genomes) and the current study is very much underpowered to draw meaningful conclusions. The previous studies have shown that estimates of mosaicism can be very much affected by sequencing artifacts, but also by age and a number of diseases and traits, such as Clonal hematopoiesis and various cancers in stem cells or the tissue being studied. It is hence not clear if the difference in mosaicism rate between this human sample and the marmoset sample is due to species level differences or within species sample level differences. The authors might argue that the haplotype assembly is adding some value to the analysis, but haplotypes can also be inferred using computational techniques.

Response: Reviewer #1 asked for us to include the mosaicism analyses and was satisfied with it. We think it has increased the value of the study. Therefore we prefer to keep it. We agree that having a higher sample size for humans and marmoset would help determine individual versus species differences, but we do not think that this preliminary mosaicism analysis on the most complete human diploid genome to date should not be done because it is an $n = 1$ individual. This study sets up a hypothesis that can then be tested with multiple more complete genomes in the future. By haplotypes inferred computationally, we believe the reviewer may be referring to tools that take high-throughput raw read data, align them against a high-quality reference, and infer haplotypes without a de-novo assembly? If so, our results imply that directly de-novo assembly of the haplotypes is more accurate than computational inference of haplotypes.

Reviewer #3 comment: 9. As pointed out in my previous review, human cells do not necessarily carry a single mitochondrial haplotype. Each cell may have multiple mitochondria, each of which has a haplotype associated with it. These haplotypes may or may not be the same. The authors have therefore not provided a “haplotype assembly” of a mitochondria but rather a consensus assembly of the multiple haplotype carried by the cells that were sequenced.

Response: We didn't mention “mitochondrial haplotype”. The reviewer could be referring to our term of “mitochondrial heteroplasmy”, which could be interpreted as the analog of a haplotype. The reviewer makes a valid point here of a “consensus mitochondrial assembly”, between different genome variants within an individual. We added a sentence to explain this: “We note that our mitochondrial assembly represents a consensus of reads with this mosaicism”

Reviewer #3 comment: 10. It would be helpful for the reader if an example could be given where the reader might make use of the references presented. I.e. apart from getting a better estimate of the

sequence of the individual being studied, is there some application where the reference, as such, could be of use to the reader? Is there some use case you can present where the reader might benefit from using these assemblies instead previously published assemblies (e.g. GRCh38 or CHM13) or in addition to those assemblies?

Response: The assembly is relatively new. Since submission of this study, there have been 10 published papers or submissions in bioRxiv that have used the HG002 reference assembly we generated. They are mostly genome assembly development papers. We now cite several of these new studies in our paper, where relevant to the topic discussed. But, we have been asked not to go beyond a specific citation limit. For biological findings beyond this paper, those studies are in progress.

Minor comments:

Reviewer #3 comment: 1. I have difficulty parsing “a hydatidiform mole with nearly two identical paternal haploid complements inherited from the maternal line”. Are the paternal haploid complements from the father of the mother representing the maternal line? If so, it might be clearer to refer to grandpaternal.

Response: We revised the sentence to make the mechanism of the cell line origin clearer. “The CHM13 cell line originated from a hydatidiform mole, whereby an ovum without maternal chromosomes was fertilized by one sperm which then duplicated its DNA leading to two nearly identical paternal haploid complements with an X chromosome (46, XX), eliminating the need to separate haplotypes and the associated diploid assembly errors.”

Reviewer #3 comment: 2. “Assemblies that did not have microbial contamination were because: 1) they did not match theGRCh38 reference for the reference-based assemblies; 2) they were removed in assemblies that filtered out scaffolds below a specific size; or 3) they were all moved to the alternate assembly.” - It is difficult for me to parse this sentence. Are you listing the three strategies that were used for removing microbial contamination?

Response: These were three methods of inadvertent removal of contamination. We have revised the sentence: “For the other assemblies, microbial contamination was inadvertently removed before submission by: 1) not matching the GRCh38 reference for the reference-based assemblies; 2) filtering out scaffolds below a specific size; or 3) all moved from the primary to the alternate assembly.”

Reviewer #3 comment: 3. Table 1 and Table S2 don’t have the same headings for assembly categories, Figure 1 uses mostly the same headings as Table S2.

Response: Thank you for this catch. We forgot to remove one of the headings in Table 1, which we have now done.

Reviewer #3 comment: 4. “The scaffolded assemblies still had quite a range of missing sequence (Ns), from ~40 kb to 50 Mb, either in the gaps between contigs or trailing Ns at scaffold ends (Extended Data Fig. 2c; Supplementary Table 2f). In comparison, GRCh38 has ~151 Mb of gaps.” - In the first sentence, missing sequence (Ns) are split into gaps and trailing (Ns), but then in the second sentence “gaps” are used to represent the number of Ns (counting both gaps and trailing Ns).

Response: Thanks for catching the ambiguity, we revised the second sentence to refer to total N bases for gaps and trailing Ns.

Reviewer #3 comment: 5. “which could also explain why some were bigger than expected” - above you state that only the diploid pair asm19a/asm19b was bigger than expected.

Response: Good catch. asm19a and asm19b had the highest false duplications. We now refer to them specifically.

Reviewer #3 comment: 6. “There were about a dozen genes present in GRCh38 and asm17 that used it as a reference, but not in any of the other HG002 assemblies or CHM13, illustrating a bias of gene presence for reference-based assembly methods” - Why is this bias and not true positives?

Response: This is a bias, because the evidence suggests that these genes are not in the HG002 genome, and instead were inserted in the assembly from filling of gaps with GRCh38 data. This is one of the fallacies of this type of reference-based method. We added to the term “false” gene presence to make it clearer to the reader that these genes were not in HG002 and are from GRCh38.

Reviewer #3 comment: 7. Heatmap, the lighter the blue (closer to 1) the more similarity between pairs of assemblies - I don't understand what “closer to 1” refers to.

Response: This is the Jaccard similarity index between pairs of assemblies in the alignment; the closer to 1 the more similar the two assemblies compared, with 1 = identical assemblies. We have now made this clear in the figure 2 legend.

Reviewer #3 comment: 8. “Of the 3,690, 2,466 genes had exclusively non-synonymous differences, and were significantly enriched (FDR < 0.01) for metabolism, smell, taste, and HSV1 viral infection

functions (Supplementary Table 12)” - The fact that these categories of genes have a higher polymorphism rates has been well documented in numerous other studies (better powered to do such an analysis).

Response: We are aware that olfactory receptors undergo more rapid evolution with higher polymorphism rate than other gene families in some species; and we found one study that make a similar claim for taste receptors in fruit flies. But, we could not find studies showing a higher rate than average for metabolism genes or genes related to HSV1 viral infections. In the revised paper, we wrote: “These findings are consistent with more rapid evolution of smell and taste receptor genes relative to the average gene family in some species (McBride et al 2007).”

Reviewer #3 comment: 9. “suggesting differential inherited differences from the parents.” - I also have a trouble parsing this comment.; Obviously since the haplotypes are different there were differences in what was inherited from the parents, is that what you mean by “differentially inherited differences” or something else?

Response: Yes, this is what we meant. But since it is trivial, we deleted the statement.

Reviewer #3 comment: 10. “Future efforts will be necessary to develop a phasing method that does not require parental sequence data and works as well as a trio method.” - In addition to the methods mentioned, computational methods for determining haplotypes in a large set of individuals could be used. Instead of using parents more distantly related individuals could be used, long range sharing of haplotypes between these individuals could serve as “surrogate parents' ". A large set of haploid references will also be helpful.

Response: As mentioned above, we believe computational inference of haplotypes from a large set of individuals won't work as well as separating and assemble haplotypes individually. Instead of solving two puzzles at once, one would need to solve many puzzles with mixed pieces at once. Surrogate data from other close relatives will be helpful for separating haplotypes, but it still will not be as complete separation as parental data. For the sake of trying to shorten the paper, we decided not to include these other future alternatives.